# MMR-Life: Piecing Together Real-life Scenes for Multimodal Multi-image Reasoning

**Jiachun Li[1,2]\*, Shaoping Huang[2,3]\*, Zhuoran Jin[1,2]†, Chenlong Zhang[1,2], Pengfei Cao[1,2], Yubo Chen[1,2], Kang Liu[1,2], Jun Zhao[1,2]†**

[1]School of Artificial Intelligence, University of Chinese Academy of Sciences
[2]The Key Laboratory of Cognition and Decision Intelligence for Complex Systems, Institute of Automation, Chinese Academy of Sciences
[3]School of Advanced Interdisciplinary Sciences, University of Chinese Academy of Sciences
{jiachun.li, pengfei.cao, zhuoran.jin}@nlpr.ia.ac.cn
**https://mmr-life-bench.github.io/**

## Abstract

Recent progress in the reasoning capabilities of multimodal large language models (MLLMs) has empowered them to address more complex tasks such as scientific analysis and mathematical reasoning. Despite their promise, MLLMs' reasoning abilities across different scenarios in real life remain largely unexplored and lack standardized benchmarks for evaluation. To address this gap, we introduce MMR-Life, a comprehensive benchmark designed to evaluate the diverse multimodal multi-image reasoning capabilities of MLLMs across real-life scenarios. MMR-Life consists of 2,646 multiple-choice questions based on 19,108 images primarily sourced from real-world contexts, comprehensively covering seven reasoning types: abductive, analogical, causal, deductive, inductive, spatial, and temporal. Unlike existing reasoning benchmarks, MMR-Life does not rely on domain-specific expertise but instead requires models to integrate information across multiple images and apply diverse reasoning abilities. The evaluation of 37 advanced models highlights the substantial challenge posed by MMR-Life. Even top models like GPT-5 achieve only 58% accuracy and display considerable variance in performance across reasoning types. Moreover, we analyze the reasoning paradigms of existing MLLMs, exploring how factors such as thinking length, reasoning method, and reasoning type affect their performance. In summary, MMR-Life establishes a comprehensive foundation for evaluating, analyzing, and improving the next generation of multimodal reasoning systems.

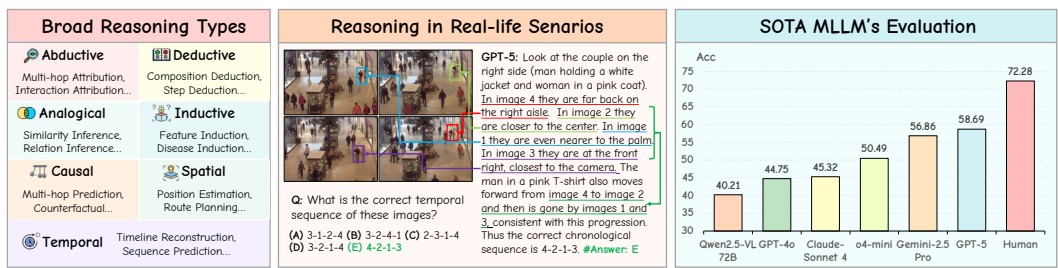

Figure 1: Overview of the **MMR-Life**. Left: 7 reasoning types and 21 tasks. Middle: A typical example of multi-image reasoning in real-life scenarios. Right: Extensive evaluation reveals a gap between humans and SOTA MLLMs on some real-life reasoning tasks.

## 1 Introduction

Reasoning is the process of generalizing from known premises to new conclusions, and it is considered a key capability for AI systems on the path to artificial general intelligence (AGI) (Sun

---

\*Equal contribution.
†Corresponding authors.

et al., 2024; Wang et al., 2025a; Li et al., 2024; 2025c; Jin et al., 2025). Recently, with the great success of reasoning large language models (RLLMs) in tasks such as mathematical reasoning (DeepSeek-AI et al., 2025; Muennighoff et al., 2025; Li et al., 2025d), there has been a widespread exploration of transferring this reasoning-enhanced paradigm to multimodal large language models (MLLMs). Representative models such as Gemini-2.5-Pro (Comanici et al., 2025), Claude-Sonnet-4 (Anthropic, 2025b), and GPT-5 (OpenAI, 2025b) leverage long Chain-of-Thought (CoT) (Wei et al., 2022) style reasoning to capture key visual information, decompose complex problems, thereby achieving or even surpassing human-level performance in diverse reasoning scenarios.

With the advancement of MLLM reasoning capabilities, there has been an increasing demand for more challenging and realistic multimodal reasoning benchmarks. Recent work mainly evaluates the reasoning ability of MLLMs through two approaches: One line of research collects expert-level domain-specific problems to assess the model's reasoning based on knowledge in areas such as scientific knowledge answering (Tie et al., 2025; Xi et al., 2025; Yue et al., 2024) and math problem solving (Wang et al., 2025c; He et al., 2024). The other line of research attempts to separate knowledge from reasoning by using synthetic problems like symbolic puzzles to assess reasoning capabilities across different difficulty levels (Song et al., 2025; Yuan et al., 2025; Chia et al., 2024).

Despite significant progress, current benchmarks still exhibit a considerable deviation from real-life reasoning scenarios. **(1) From the task design perspective, the tasks in existing benchmarks are not commonly encountered in everyday reasoning.** Both knowledge-intensive tasks and synthesized puzzle-based tasks remain misaligned with the authentic reasoning demands that arise in everyday situations. For the former, daily reasoning seldom relies on expert-level knowledge, whereas for the latter, the symbolic input images differ substantially from those encountered in real-world scenarios. **(2) From the perspective of input images, current benchmarks fail to include multi-image inputs that span a diverse range of reasoning types.** A large portion of multimodal general reasoning benchmarks focus exclusively on single-image inputs (Yue et al., 2024; 2025a; Song et al., 2025), which contrasts with real-world conditions where we perceive visual information as a sequence of images rather than a single one. For multi-image benchmarks, existing work either incorporates non-reasoning tasks or focuses on a limited reasoning type (Cheng et al., 2025; Kil et al., 2024; Liu et al., 2024; Meng et al., 2025b), making it difficult to support further comprehensive evaluation of MLLM reasoning performance.

To address these issues, we introduce **MMR-Life**, a comprehensive benchmark designed to evaluate the multimodal multi-image reasoning capability of MLLMs across real-life scenarios. MMR-Life contains **2,646** carefully curated questions, covering 7 distinct reasoning types (see Figure 1, 2), which broadly encompass the reasoning abilities necessary for everyday situations. In MMR-Life, each question is associated with a set of images, primarily taken in real-world scenarios. The answers do not require domain-specific expertise but instead ask models to extract key information from multiple real-life images and derive new conclusions. This design aligns MMR-Life more closely with the reasoning types found in everyday life. Figure 1 shows an example from MMR-Life. To address the temporal ordering problem, the model needs to detect individuals recurring across different surveillance images and track their movements, selecting the correct order.

Extensive evaluations on 37 advanced MLLMs demonstrate that the real-world reasoning scenarios in MMR-Life remain highly challenging. As illustrated in Figure 1, even the most advanced models, including GPT-5 and Gemini-2.5-Pro, reach only 58.69% and 56.86% accuracy on MMR-Life, falling short of human performance by 14%. Besides, the evaluation results demonstrate substantial performance disparities across reasoning types. Existing MLLMs perform relatively well on analogical, deductive, and inductive reasoning, but encounter notable bottlenecks in causal, spatial, and temporal reasoning. Based on MMR-Life, we conduct an analysis of MLLM reasoning paradigms and obtain several key findings, including that long thinking benefits only limited reasoning types, RL's weaker generalization in small models, and the clustering of reasoning types into patterns.

In summary, our contributions include: (1) We propose MMR-Life, the first comprehensive benchmark for evaluating multimodal multi-image reasoning in real-life scenarios across seven reasoning types. (2) Through an extensive evaluation of 37 state-of-the-art MLLMs on MMR-Life, we find that existing models struggle considerably in real-life reasoning, especially in causal, spatial, and temporal tasks. (3) Based on MMR-Life, we conduct an in-depth analysis of current MLLM reasoning paradigms, revealing key findings such as the limited effectiveness of long thinking to certain

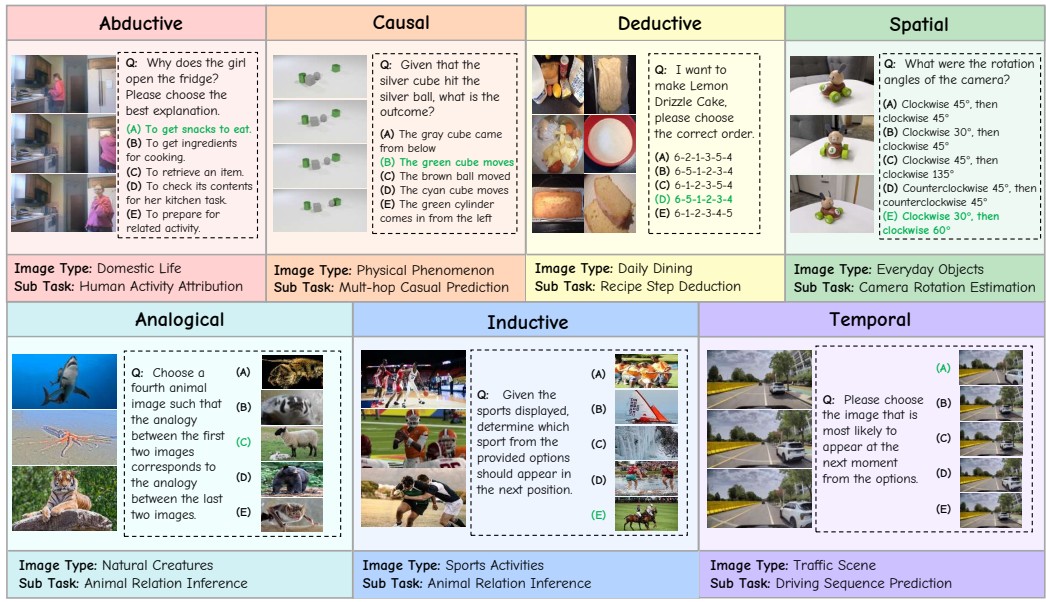

Figure 2: MMR-Life examples from each reasoning type.

reasoning types, the weaker generalization of RL on small models, and the presence of pattern clustering across reasoning types.

## 2 THE MMR-LIFE BENCHMARK

### 2.1 OVERVIEW

We introduce the **M**ultimodal **M**ulti-image **R**easoning benchmark under real-**Life** scenarios (MMR-Life), a novel benchmark meticulously curated to evaluate the ability of MLLMs to perform diverse types of reasoning in everyday situations. MMR-Life consists of 2,646 multiple-choice questions based on 19,108 images, comprehensively covering 7 reasoning types (i.e., abductive, analogical, causal, deductive, inductive, spatial, and temporal) and 21 tasks. Each task is based on a set of multi-images, predominantly sourced from real-life contexts, such as domestic life, daily dining, and sports activities. See Figure 2 for examples in MMR-Life and Table 1 for dataset statistics. We further discuss the key concepts (e.g., real-life scenarios) of our benchmark in Appendix B.

### 2.2 DATA CURATION PIPELINE

**Data Collection.** We initiate our pipeline by collecting real-life images from a variety of sources, including: **(1) Public image datasets:** We select high-resolution real-world image datasets from Kaggle (Kaggle, 2025), ensuring that the images within each dataset are related (e.g., temporal relationships), to facilitate the construction of multi-image inputs for our questions. **(2) Open web resources:** We take screenshots from publicly available web resources to collect real-world multi-image data. For example, we obtain bird distribution density images from the eBird website (eBird, 2025). **(3) Public video sources:** Given the inherent correlation between frames in a video, they are ideal for multi-image data. We extract frames from publicly available video datasets to create images, while ensuring the clarity of each frame. **(4) Other existing benchmarks:** Finally, we collect data from existing multi-image or video reasoning bench-

Table 1: Key statistics of MMR-Life.

| Statistics | Number |
|---|---|
| Total Questions | 2,646 |
| Total Reasoning Types/Tasks | 7/21 |
| Image Types | 15 |
| Reasoning Type | |
| - Abductive Reasoning | 307 (11.60%) |
| - Analogical Reasoning | 568 (21.47%) |
| - Causal Reasoning | 263 (9.94%) |
| - Deductive Reasoning | 282 (10.66%) |
| - Inductive Reasoning | 429 (16.21%) |
| - Spatial Reasoning | 255 (9.64%) |
| - Temporal Reasoning | 542 (20.48%) |
| Text Options | 1454 (54.95%) |
| Image Options | 1192 (45.05%) |
| Average Image Counts | 7.22 |
| Average Question Length | 283 |

marks, extract frames from the videos, and remove images with low quality. The detailed collection protocol and data sources for each task are reported in Appendix C.1.

**Task Design.** To make our benchmark more aligned with real-life scenarios, we aim to cover a broader range of reasoning types, reflecting diverse everyday situations. Specifically, based on the collected images, we design 7 distinct reasoning types (see Figure 2 for examples): **(1) Abductive Reasoning (Abd)**: Given the observed event, inferring the most plausible explanation for why the event occurred. **(2) Analogical Reasoning (Ana)**: Inferring conclusions about a new situation by identifying similarities with a known case. **(3) Causal Reasoning (Cau)**: In contrast to abductive reasoning, based on the cause, inferring the effect. **(4) Deductive Reasoning (Ded)**: Based on general rules or premises, drawing logically certain conclusions about specific cases. **(5) Inductive Reasoning (Ind)**: Generalizing rules or patterns from specific observations. **(6) Spatial Reasoning (Spa)**: Understanding and reasoning about the locations, movement, and spatial relations of objects. **(7) Temporal Reasoning (Tem)**: Reasoning about the order, duration, and timing of events.

**Question-Answer Generation.** We generate question-answer pairs using either automatic synthesis or manual annotation, depending on the task type. In some cases, the explicit information contained within the multi-image set we collect is already sufficient to fulfill the task's requirements. For example, in the temporal reasoning example of Figure 2, the images themselves contain sequential information, which is sufficient for the sequence prediction task. In these cases, we can define heuristic rules and use code to automate the synthesis of question-answer pairs using the information. However, some tasks require reasoning over implicit information in images. For instance, in the abductive reasoning example of Figure 2, we need to identify causal event pairs within the scene to construct the questions. In these cases, we manually design question-answer pairs according to the reasoning type to ensure the quality of the data. This process leads to the creation of a diverse set of 3.2K questions from multiple sources. See Appendix C.2 for detailed annotation guidelines and Appendix E for task details.

**Negative Option Generation.** Given that many reasoning tasks do not have a single correct answer (e.g., providing a plausible explanation in abductive reasoning), we design all questions in a multiple-choice format, where the model must choose the most appropriate answer from five options. Each option is presented as either an image or text (with the distribution provided in Table 1). For image options, we use heuristic rules to sample incorrect candidates. As an example, in the temporal reasoning example in Figure 2, we construct negative options by choosing frames that either precede the input images or occur at much later time steps. For text options, we invoke GPT-5-mini (OpenAI, 2025b), GPT-4o (OpenAI, 2024), and Qwen2.5-VL-32B (Bai et al., 2025) to generate responses (see prompts in Appendix C.3). From all generated incorrect responses, we manually choose the four highest-quality erroneous options to serve as the final incorrect choices.

**Data Quality Control.** To further control the quality of our data, we perform three steps of data filtering. **(1) Difficulty filtering:** We employ three smaller models, Qwen2.5-VL-7B (Bai et al., 2025), Gemma3-4B (Kamath et al., 2025), and InternVL3.5-8B (Wang et al., 2025e), to generate answers for each question. If all models answer correctly, this suggests that the questions are too easy for existing MLLMs, and they are therefore filtered out. **(2) Format filtering:** The model-generated incorrect options may have significant format differences (e.g., length) compared to the human-constructed correct answers, which may result in the model relying on shortcuts. To mitigate this effect, we manually revise the options with substantial format differences. **(3) Quality filtering:** Finally, we distribute the problems among different co-authors, filtering out questions that exhibit semantic ambiguity, have multiple correct answers, or require domain-specific expertise.

## 2.3 COMPARISONS WITH EXISTING BENCHMARKS

To further distinguish the difference between MMR-Life and other existing ones, we provide detailed comparisons in Table 2. From the image type perspective, most existing datasets include a large proportion of symbolic images such as charts and puzzles, which creates a gap from the natural images encountered in daily life. Our benchmark excludes such images, making the evaluation more closely aligned with real-life scenarios. From the source perspective, all questions in our dataset are newly annotated rather than sampled directly from existing datasets, textbooks, or the web, which reduces the risk of data contamination.

Table 2: The comparison between MMR-Life and other existing benchmarks. **W** (Web), **T** (Textbook), **A** (Annotated), **E** (Existing datasets), and **Avg Img.#** (average image counts each question).

| Dataset | Size | Images | Reason | Source | Knowledge | Avg Img.# |
|---------|------|--------|--------|--------|-----------|-----------|
| MME-Reasoning (Yuan et al., 2025) | 1.2K | Symbolic | 3 Types | W+T+A+E | Low | 1 |
| VisualPuzzles (Song et al., 2025) | 1.1K | Symbolic | 5 Types | W+T+A | Low | 1 |
| MMMU (Yue et al., 2024) | 11.5K | Hybrid | - | W+T+A | High | 1.05 |
| MMLU-Reason (Tie et al., 2025) | 1.1K | Hybrid | 6 Types | W+T+E | Medium | 1.85 |
| MEGA-Bench (Chen et al., 2025) | 8K | Hybrid | - | A | Medium | 2 |
| MME-COT (Jiang et al., 2025) | 1.1K | Hybrid | 6 Types | E | Medium | 2.10 |
| MV-MATH (Wang et al., 2025c) | 2K | Symbolic | 1 Type | W+A | High | 3.02 |
| MMRB (Cheng et al., 2025) | 4.8K | Hybrid | 3 Types | E | Medium | 6.17 |
| MMIU (Meng et al., 2025b) | 11.6K | Hybrid | - | A | Medium | 6.64 |
| **MMR-Life (Ours)** | **2.7K** | **Natural** | **7 Types** | **A** | **Low** | **7.22** |

## 3 MAIN EXPERIMENT

### 3.1 EXPERIMENTAL SETTINGS

**Multi-modal Language Models without Thinking.** We first evaluate the performance of SOTA non-thinking MLLMs on our benchmark. These models have not undergone additional reasoning-enhancement training and lack long CoT capabilities. Open-source models include Qwen2.5-7/32/70B (Bai et al., 2025), Gemma3-12/27B (Kamath et al., 2025), InternVL3.5-8B/30B-A3B (Wang et al., 2025e). Closed-source models include GPT-4.1-mini, GPT-4.1 (OpenAI, 2025a), GPT-4o (OpenAI, 2024), Claude-3.7-Sonnet (without thinking) (Anthropic, 2025a) and Doubao-1.5-vision (ByteDance Seed Team, 2025).

**Multi-modal Language Models with Thinking.** To study the effect of long CoT patterns on the reasoning abilities of MLLMs, we introduce several advanced thinking models into the evaluation. Open-source models include VL-Rethinker-7/72B(Wang et al., 2025b), MM-Eureka-Qwen-32B (Meng et al., 2025a), MiMo-VL-7B-RL (Yue et al., 2025c), Keye-VL-1.5-8B (Team et al., 2025), QVQ-72B-Preview (Qwen Team, 2024). Closed-source models include o4-mini (OpenAI, 2025c), Claude-Sonnet-4-Thinking (Anthropic, 2025b), Gemini-2.5-Flash (Comanici et al., 2025), Gemini-2.5-Pro (Comanici et al., 2025), GPT-5-mini and GPT-5 (OpenAI, 2025b). We provide complete experiments and results for a total of 37 models in Appendix F.2.

**Human Level Performance.** We employ 12 students with varying degrees and academic backgrounds. Then, we extract 10 questions from each task to form a mini test set of 210 unique questions. From this pool, we repeatedly sample 50 questions at a time and assign them to one of 12 students, yielding a total of 600 valid human answers. These students are instructed not to use external knowledge sources such as the internet or books. We report the experimental results on this tiny set in Appendix F.3.

**Implementation Details.** We employ the same zero-shot CoT prompt as input for all models in the main experiments to perform reasoning. To minimize random variation, we conduct five runs for every open-source model and use the average performance as the final outcome. All experiments are performed using 8 NVIDIA A100 GPUs. The detailed experimental parameters and prompts are provided in Appendix F.1.

### 3.2 MAIN RESULTS

Table 3 presents MLLMs' performance on MMR-Life, from which we draw several critical insights:

**Our MMR-Life benchmark poses significant challenges for MLLMs.** Despite achieving nearly 90% accuracy (OpenAI, 2025b) on complex multimodal reasoning tasks like GPQA (Rein et al., 2023) and MMMU (Yue et al., 2024), GPT-5 only achieved an accuracy of **58.69%** on MMR-Life, with a 14% gap compared to human performance. Moreover, almost all open-source models have an accuracy rate below **40%**, with some of the most recent models, such as Skywork-R1V-38B and InternVL3.5-8B, performing worse than random guessing (20%). **This suggests that, although MMR-Life does not include complex knowledge requirements, our real-life reasoning**

Table 3: Performance comparison of SOTA MLLMs on MMR-Life. The highest and lowest scores for each model type across reasoning types are highlighted in green and red, respectively. The highest performance achieved by the model in each type is indicated in **bold**.

| Model | Abd | Ana | Cau | Ded | Ind | Spa | Tem | Avg |
|---|---|---|---|---|---|---|---|---|
| Human | 79.76 | 57.65 | 75.00 | 70.59 | 63.41 | 79.76 | 79.76 | 72.28 |
| *Closed-source & Thinking* | | | | | | | | |
| GPT-5 | 53.75 | **78.87** | 41.06 | **80.14** | **78.32** | 17.25 | **41.70** | 58.69 |
| Gemini-2.5-Pro | **54.40** | 73.77 | 36.99 | 79.43 | 73.66 | **25.10** | 35.79 | 56.86 |
| Gemini-2.5-Flash | 46.25 | 75.18 | 34.22 | 71.63 | 73.66 | 23.92 | 30.81 | 53.10 |
| o4-mini | 41.37 | 73.59 | 27.38 | 71.28 | 68.07 | 19.22 | 32.66 | 50.49 |
| GPT-5-mini | 44.95 | 69.72 | 32.32 | 75.18 | 68.76 | 12.16 | 29.52 | 49.77 |
| Claude-Sonnet-4 | 36.96 | 60.92 | **44.11** | 67.02 | 56.64 | 15.69 | 28.23 | 45.32 |
| *Closed-source & No Thinking* | | | | | | | | |
| GPT-4.1 | 44.30 | 71.30 | 22.43 | 67.38 | 70.16 | 13.73 | 27.31 | 48.15 |
| Claude-3.7-Sonnet | 33.55 | 66.55 | 35.36 | 59.93 | 59.67 | 20.78 | 26.01 | 45.09 |
| GPT-4o | 46.91 | 65.67 | 25.86 | 51.42 | 66.20 | 11.37 | 26.01 | 44.75 |
| GPT-4.1-mini | 32.90 | 61.62 | 30.80 | 52.13 | 65.27 | 16.47 | 30.63 | 44.10 |
| Doubao-1.5-vision | 37.13 | 53.70 | 31.18 | 59.57 | 54.31 | 12.16 | 23.06 | 39.98 |
| *Open-source & Thinking* | | | | | | | | |
| VL-Rethinker-72B | 36.48 | 50.88 | 33.08 | 56.03 | 57.58 | 15.69 | 21.59 | 39.68 |
| QVQ-72B-Preview | 31.27 | 41.20 | 38.02 | 47.87 | 31.24 | 14.12 | 16.42 | 31.14 |
| MM-Eureka-Qwen-32B | 26.06 | 41.02 | 25.10 | 47.52 | 27.97 | 16.08 | 17.34 | 29.02 |
| MiMo-VL-7B-RL | 38.76 | 25.88 | 28.14 | 60.99 | 24.94 | 14.12 | 19.19 | 28.68 |
| Keye-VL-1.5-8B | 19.87 | 21.30 | 23.95 | 14.18 | 20.28 | 13.73 | 23.62 | 20.22 |
| Skywork-R1V-38B | 22.15 | 10.39 | 16.73 | 23.76 | 11.89 | 9.80 | 11.07 | 14.13 |
| *Open-source & No Thinking* | | | | | | | | |
| Qwen2.5-VL-72B | 35.50 | 55.46 | 35.36 | 52.13 | 55.48 | 12.94 | 23.80 | 40.21 |
| Gemma3-27B | 35.18 | 57.92 | 36.88 | 31.21 | 60.61 | 12.94 | 18.27 | 38.32 |
| Gemma3-12B | 25.08 | 50.70 | 17.11 | 27.30 | 42.42 | 10.20 | 15.87 | 29.52 |
| Qwen2.5-VL-32B | 23.45 | 42.78 | 21.29 | 50.00 | 27.27 | 15.69 | 16.24 | 28.61 |
| Qwen2.5-VL-7B | 26.06 | 35.74 | 20.53 | 20.92 | 38.93 | 9.41 | 12.18 | 24.68 |
| InternVL3.5-30B-A3B | 45.60 | 19.19 | 33.46 | 36.52 | 14.45 | 12.16 | 14.39 | 23.09 |
| InternVL3.5-8B | 35.18 | 11.44 | 18.63 | 34.04 | 11.19 | 14.90 | 16.61 | 18.67 |

**scenarios still present a significant challenge for current MLLMs.** Future model training and optimization should focus more on these real-world situations.

**MLLMs exhibit large disparities across different types of reasoning.** While current models perform well in analogical, deductive, and inductive reasoning tasks, they still have substantial room for improvement in causal, spatial, and temporal reasoning tasks. We observe that all models perform poorly in spatial reasoning, with the highest accuracy being only **25.10%**, compared to the human accuracy of **79.76%**. In contrast, for tasks like analogical reasoning, most closed-source models outperform human performance. Current models can easily acquire abilities such as analogy and deductive reasoning through feature associations or by memorizing explicit reasoning paths. However, **they struggle to learn more abstract world representations, such as spatial and temporal reasoning**. This bias is one that future model training should seek to correct.

**Current open-source thinking models bring limited improvement.** When evaluating the effect of adding a thinking mode to MLLMs, we find that closed-source thinking models generally outperform closed-source no-thinking models. However, **for open-source models, the thinking mode does not show improved reasoning capabilities.** In Table 3, the open-source no-thinking model achieves an average accuracy of 29.01%, whereas the thinking model achieves only **27.15%** on average. This implies that there is substantial potential for improving the reasoning abilities of current open-source thinking models, particularly in their ability to generalize to real-world contexts.

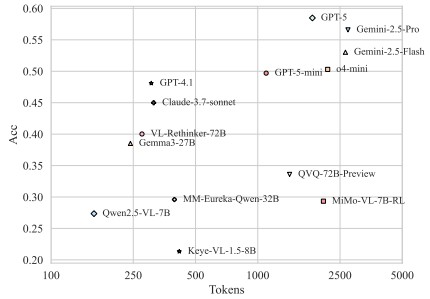 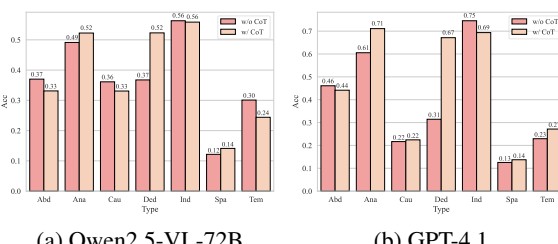

(a) Qwen2.5-VL-72B          (b) GPT-4.1

Figure 3: Response tokens vs. Accuracy.          Figure 4: Performance: without CoT vs. with CoT.

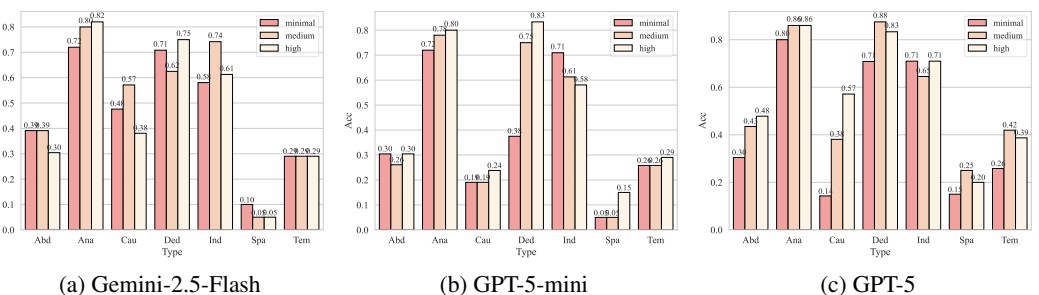

(a) Gemini-2.5-Flash          (b) GPT-5-mini          (c) GPT-5

Figure 5: Performance comparison under different thinking budgets.

# 4 THINKING PATTERN ANALYSIS

## 4.1 IS LONGER THINKING ALWAYS BETTER?

From Table 3, we find that closed-source thinking models perform best on MMR-Life. An important question then arises: Is this superior performance associated with the longer reasoning processes?

**Reasoning Performance Scales Logarithmically With Thinking Length.** To investigate the question, we first present the semi-log plot of average response token count versus average accuracy over 14 models (see Figure 3). The overall trend shows that models with longer outputs tend to achieve higher scores, indicating that reasoning capabilities scale roughly in proportion to the logarithm of the reasoning length. However, there are notable exceptions. Certain open-source thinking models, including MiMo-VL-7B-RL and QVQ-72B-Preview, are located in the lower-right region of Figure 3, demonstrating that balancing reasoning efficiency and model effectiveness remains a major challenge for future open-source MLLMs.

**Longer Thinking Is Not All You Need.** We conduct a more fine-grained analysis to investigate the relationship between model performance and thinking length across distinct reasoning types. Specifically, for no-thinking models, we follow prior work by comparing their reasoning performance with and without CoT (Li et al., 2025e; Sprague et al., 2025) (see Figure 4). For thinking models, we select those with a controllable reasoning budget and vary the budget (minimal, medium, and high) to gradually increase CoT length, thereby comparing performance across different thinking lengths (see Figure 5). From both figures, it is evident that longer thoughts do not lead to better performance for all reasoning types. For reasoning types like inductive reasoning, the performance with CoT is worse in no-thinking models (see Figure 4) and using more reasoning budget does not lead to better performance in thinking models (see Figure 5). Conversely, for reasoning types such as analogical reasoning, the incorporation of CoT or longer CoT results in a noticeable performance improvement. We hypothesize that this is because longer CoT may only be suitable for tasks requiring step-by-step reasoning, while types like inductive reasoning may benefit more from faster thinking (Li et al., 2025e).

Table 4: Performance across different methods. Scores higher and lower than the base model's CoT performance are marked in green and red . The highest score in each column is in **bold**.

| Model | Method | Abd | Ana | Cau | Ded | Ind | Spa | Tem | Avg (Δ) |
|---|---|---|---|---|---|---|---|---|---|
| **Qwen2.5-VL-7B** | CoT | 26.06 | 35.74 | 20.53 | 20.92 | 38.93 | 9.41 | 12.18 | 24.68 |
| | SC@8 | 28.01 | 39.44 | **23.57** | 25.18 | 45.45 | 10.98 | **13.10** | 27.85 (+3.17) |
| | BoN@8 | 27.64 | **44.72** | 22.81 | 25.53 | **48.02** | 13.33 | **13.10** | **29.54** (+4.86) |
| | GRPO | **30.62** | 40.49 | 21.29 | **28.72** | 43.59 | **13.73** | 11.81 | 28.23 (+3.55) |
| **Qwen2.5-VL-32B** | CoT | 23.45 | 42.78 | 21.29 | 50.00 | 27.27 | 15.69 | 16.24 | 28.61 |
| | SC@8 | **26.06** | **45.42** | 23.95 | 51.77 | 28.67 | **16.47** | 17.90 | 30.57 (+1.96) |
| | BoN@8 | 25.78 | 44.89 | 19.39 | **55.32** | 30.54 | **16.47** | 19.56 | **30.97** (+2.36) |
| | GRPO | 22.98 | 42.96 | **28.14** | 49.65 | **30.77** | 14.90 | 19.19 | 30.29 (+1.68) |
| **Qwen2.5-VL-72B** | CoT | 35.50 | 55.46 | **35.36** | 52.13 | 55.48 | 12.94 | 23.80 | 40.21 |
| | SC@8 | 35.18 | **56.16** | **35.36** | 52.13 | 54.78 | 12.94 | 24.35 | **40.33** (+0.12) |
| | BoN@8 | 34.20 | 53.35 | 32.70 | 51.77 | 56.88 | 13.73 | **24.72** | 39.80 (-0.41) |
| | GRPO | **36.48** | 50.88 | 33.08 | **56.03** | **57.58** | 15.69 | 21.59 | 39.68 (-0.53) |

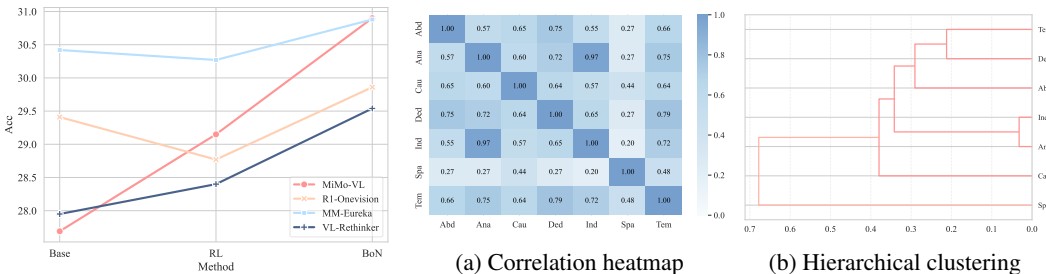

Figure 6: Comparison of BoN and RL performance on different models.

Figure 7: Analysis of correlations across different reasoning types (averaged across all models we evaluate).

(a) Correlation heatmap      (b) Hierarchical clustering

## 4.2 Do Generalizable Reasoning Enhancement Methods Exist?

From the inception of CoT (Wei et al., 2022) to the broad application of GRPO (DeepSeek-AI et al., 2025), the reasoning-enhancement techniques have undergone substantial evolution. In this section, we analyze and compare the generalizability of these approaches.

**Failure of Enhancement Methods in Larger Models.** We select four representative reasoning-enhancement methods for comparison: CoT, Self-Consistency (SC), Best-of-N (BoN), and GRPO. To evaluate the generalizability of these methods, we directly use previously trained models for inference without any training on MMR-Life. Specifically, we adopt the Skywork-VL Reward (Wang et al., 2025f) as the reward model for BoN and the VL-Rethinker series (Wang et al., 2025b) as the GRPO-trained models. As shown in Table 4, the results demonstrate that: Across model scales from 7B to 72B, the average performance difference between other methods and CoT consistently decreases, while an increasing number of subtypes transition from performance gains to performance drops (from green to red). Strikingly, on Qwen-2.5-VL-72B, the performance of BoN and GRPO falls short of simply applying CoT. According to previous works (Yue et al., 2025b), we hypothesize that this is because these methods primarily improve sampling efficiency towards correct reasoning paths. For larger models, the likelihood of sampling correct paths is naturally higher, which diminishes the gains from reasoning-enhancement methods.

**RL Generalizes Worse than BoN on Small Models.** Reinforcement learning methods, exemplified by GRPO, have gained wide adoption for their strong reasoning generalization (DeepSeek-AI et al., 2025). Nevertheless, our results in Table 4 reveal that on all of the three models, GRPO exhibits weaker generalization compared to BoN. To further validate this finding, we conduct experiments on additional small MLLMs (see Appendix G for details), comparing the performance of BoN@8 applied to base models with that of RL-trained models. The results in Figure 6 show that across different model architectures and training datasets, RL-trained models consistently underperform compared to BoN inference on the corresponding base models. In some cases, RL models even perform worse than the base models using CoT. This calls for a reconsideration of RL tech-

niques: Do RL methods on small models merely lead to overfitting on specific datasets? We leave this question open for further exploration in future work.

### 4.3 DO DIFFERENT REASONING TYPES CORRELATE?

Former findings demonstrate significant differences in model performance across types. In this section, we aim to capture the underlying relationships among these categories.

**Correlations Between Reasoning Types.** We compute the accuracy of all models across reasoning types, calculate the Pearson correlation coefficients between them, and present the results in Figure 6a. It demonstrates substantial differences in correlations across these types. Some categories, such as inductive and analogical reasoning, exhibit very high correlations (0.97), while some others, such as spatial and inductive reasoning, show low correlations (0.40).

**Uncovering Pattern Clusters in Reasoning.** Furthermore, we normalize the negative of the correlation coefficient as the distance between categories and perform hierarchical clustering. In Figure 6b, we observe clusters formed by similar reasoning types (e.g., Ana–Ind), suggesting the existence of higher-order reasoning patterns in MLLMs. For example, both analogical and inductive reasoning rely on a shared pattern of abstracting general rules from concrete features. Conversely, reasoning types with greater distances suggest that they involve relatively disjoint patterns. As an example, spatial reasoning is distant from all other categories, suggesting that the capabilities it requires (e.g., location, distance estimation) are difficult to learn from non-spatial tasks. In conclusion, MMR-Life enables us to uncover a higher-level hierarchy of reasoning patterns, facilitating a deeper understanding of reasoning generalization across diverse tasks.

## 5 ERROR ANALYSIS

This section focuses on the errors made by GPT-5 and Gemini-2.5-Pro, the two strongest models on MMR-Life. For each model, we randomly select 20 incorrect examples from each reasoning type and identify the root causes of the model's erroneous responses. The distribution of these errors is shown in Figure 8, with a selection of notable 42 cases and detailed analyses provided in the Appendix H. The results reveal that reasoning errors dominate at 32%, with the model frequently making basic logical mistakes such as causal inversion (24%), tem-

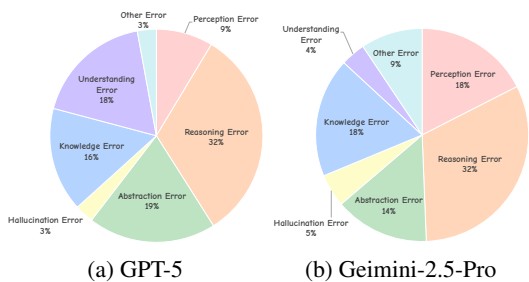

(a) GPT-5      (b) Geimini-2.5-Pro

Figure 8: Error distribution over 140 errors for each model on MMR-Life.

poral confusion (42%), and missing key steps (24%) during reasoning. In addition, abstraction errors (17%), which reflect the model's short-term thinking capabilities, such as the ability to make associations, are also notable. Knowledge errors (17%) and perception errors (12%) constitute substantial portions of failures, indicating challenges in recalling the correct knowledge for reasoning, as well as difficulties in identifying static attributes of objects (e.g., color, shape) and dynamic changes (e.g., movement). By systematically examining these failures, we not only expose critical shortcomings in current MLLMs but also derive actionable insights that can inform the next generation of MLLMs.

## 6 RELATED WORK

**Multimodal Reasoning Enhancement Methods.** The development of methods in multimodal reasoning closely follows the approaches established in pure language processing. Inspired by its success in text-only settings, CoT has recently been extended to MLLMs, leading to the development of prompt-guided multimodal reasoning. Studies such as IPVR (Chen et al., 2023), CCoT(Mitra et al., 2024), and VisualSketchpad (Hu et al., 2024) combine reasoning with perception, enhancing the reliability of the reasoning process. After that, the search-based inference method brings reward models into the multimodal reasoning process, training a scoring model to evaluate and select the best reasoning path (Wang et al., 2025d; Zang et al., 2025; Wang et al., 2025f). Recently, following the success of Deepseek-R1 GRPO (DeepSeek-AI et al., 2025), a group of thinking MLLMs like

VL-Rethinker (Wang et al., 2025b), MM-Eureka (Meng et al., 2025a), and MiMo-VL (Yue et al., 2025c) have emerged. Our benchmark comprehensively evaluates different methods and MLLMs, aiming to guide their further optimization.

**Multimodal Reasoning Benchmarks.**   There exists a number of multimodal benchmarks testing MLLMs' reasoning abilities. Several studies combine world knowledge with reasoning and assess the reasoning capabilities of MLLMs across various STEM fields, such as GPQA (Rein et al., 2023), OlympiadBench (He et al., 2024), MME-CoT (Jiang et al., 2025), MMR-V (Zhu et al., 2025) and MMLU-Reason (Tie et al., 2025). Other works argue that reasoning should be decoupled from knowledge, using symbolic patterns to evaluate the model's logical reasoning abilities, such as Puz-zleVQA (Chia et al., 2024), VisualPuzzles (Song et al., 2025), and MME-Reasoning (Yuan et al., 2025). However, both types of benchmarks exhibit deviations from real-life reasoning scenarios due to the expert-level knowledge and symbolic images. Although recent work on spatial reasoning meets real-life requirements (Yang et al., 2025a; Li et al., 2025b; Yang et al., 2025b), it covers only a limited set of reasoning types. Our MMR-Life benchmark covers seven different reasoning types and introduces real-life multi-image input, addressing former gaps.

## 7 CONCLUSION

We present MMR-Life, a novel benchmark designed to evaluate the multimodal reasoning abilities of current MLLMs across seven distinct reasoning types using multiple real-life images as inputs. Through careful and diverse data curation, our dataset provides a comprehensive evaluation of MLLMs' reasoning performance across various real-life scenarios, which shows that existing MLLMs still face significant challenges and exhibit notable performance imbalances across different reasoning types. We conduct a further analysis of the reasoning paradigms of these models, uncovering the relationship between the thinking length, enhancement methods, and reasoning abilities of MLLMs, which lays the foundation for the development of more generalizable AI systems.

## ETHICS STATEMENT

In constructing our benchmark, we ensure strict adherence to copyright and licensing regulations, explicitly avoiding data from sources that prohibit copying or redistribution. Besides, we avoid the images that contain any private information or harmful content. The data in our MMR-Life are not intended to replace, nor are they capable of replacing, the original data source. Therefore, we assert that their inclusion does not affect the market value or utility of the original materials. We did not employ external crowdsourcing or paid annotation platforms. All participants volunteered, with a complete understanding of the research goals, procedures, and the intended use of the data.

## REPRODUCIBILITY STATEMENT

We have taken several steps to improve the reproducibility of our research. Regarding the data, we provide a thorough description of the data sources for each task, along with links, in Appendix C.1. A subset of 210 items, including the questions and their corresponding images, is also uploaded in the supplementary materials. Additionally, we describe the dataset construction process and the prompts used in both §2.2 and Appendix C. On the experimental side, we offer a detailed account of the model versions, parameter settings, and prompts used in the experiments, which are outlined in Appendix F.1. The full experimental code is also uploaded in the supplementary materials. We commit to making all data and code open source if the paper is accepted.

## ACKNOWLEDGEMENT

This work is supported by the National Natural Science Foundation of China (No. U24A20335, No. 62406321). This work is also supported by Beijing Natural Science Foundation (L243006).

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

## A  THE USE OF LARGE LANGUAGE MODELS

In this study, a large language model (LLM) was employed as a tool to assist in the refinement and enhancement of the manuscript's language. The specific usages of the LLM include:

- **Grammar and Syntax Improvement:** The LLM helped to correct grammatical errors and improve sentence structures, contributing to greater clarity and fluency in the writing.

- **Conciseness and Precision:** It provided suggestions for more concise and precise wording, aiding in the refinement of certain sections without altering their meaning.

It is important to note that while the LLM contributed to the refinement of the manuscript's language, the research ideas, data analysis, and conclusions were independently conceived and developed by the authors. The LLM's contributions were exclusively related to text refinement and did not extend to the conceptual aspects of the study.

## B  KEY CONCEPTS IN MMR-LIFE

We begin by discussing key concepts in the benchmark to clearly define the core problems and design principles that our work addresses.

### B.1  REASONING IN REAL-LIFE SCENARIO

As real-life reasoning is a fundamental design principle of our benchmark, we provide a brief definition of it:

**Definition 1** (**Reasoning in Real-life Scenarios**). *Reasoning in real-life scenarios refers to the process of applying diverse reasoning capabilities to solve problems from everyday situations, which are defined by a set of images and textual descriptions that satisfy the following conditions:*

- *(i)* *Multiple natural images: The input must contain multiple images, each depicting objects or events that either objectively exist in real life or are realistically simulated to resemble real-world conditions. Purely abstract diagrams or symbolic renderings are excluded.*

- *(ii)* *Commonsense solvability: The answer to the problem must not rely on complex domain-specific knowledge. Instead, it should be solvable using only basic human commonsense reasoning and general logic.*

As mentioned in §1, the two existing benchmark types do not fully adhere to the above definition, as they often incorporate unnatural images, such as charts and synthetic puzzles, and may require specialized domain knowledge. In contrast, MMR-Life is constructed in strict accordance with the above definition, emphasizing the evaluation of reasoning in real-life scenarios from the outset. It should be noted that this definition is not intended to be broadly applicable but serves as the guiding principle for the design of this study.

### B.2  MULTI-IMAGE VS. VIDEO

In §1, we noted that real-life images are continuous, which led us to adopt multi-image input. However, a natural question arises: why not use continuous videos instead? In the following, we compare and discuss this choice. Overall, we opt not to use video as our input format for the following reasons:

- *Low Reasoning Types Coverage:* The relationship between multiple images in a video is typically limited to temporal sequencing. In this context, it is difficult to design reasoning tasks, such as analogy or inductive reasoning, since these tasks often require a parallel relationship between the images, which cannot be fully captured by a video input.

- *Low Data Diversity:* From a data perspective, as discussed in §2.2, real-world videos are only a subset of our image sources. If all inputs were required to be videos, we would lose a significant variety of data sources, such as natural photographs, thereby reducing data diversity.

Table 5: Data sources and image types of different tasks in MMR-Life

| Reasoning Type | Task Type | Image Type | Data Source |
|---|---|---|---|
| Abductive | Human Activity Attribution
Character Interaction Attribution
Multi-Hop Collision Attribution | Domestic Life
Human Animation
Physical Phenomenon | TVbench (Cores et al., 2025)
Tom & Jerry Cartoon (Parmar et al., 2024)
CLEVRER-Humans (Mao et al., 2022) |
| Analogical | Animal Relation Inference
Product Similarity Inference
Artwork Style Inference | Natural Creatures
Product Shots
Human Artwork | Kaggle (sanadalali, 2025)
Kaggle (Gégénava, 2025)
Kaggle (ikarus777, 2019) |
| Causal | Character Interaction Prediction
Multi-Hop Collision Prediction
Counterfactual Fluid Prediction | Human Animation
Physical Phenomenon
Physical Phenomenon | Tom & Jerry Cartoon (Parmar et al., 2024)
CLEVRER-Humans (Mao et al., 2022)
ContPhy (Zheng et al., 2024) |
| Deductive | Material Composition Deduction
Card Winner Deduction
Recipe Step Deduction | Everyday Objects
Game Symbols
Daily Dining | MathVisa (Lu et al., 2024)
Kaggle (patricklford, 2025)
RecipeQA (Yagcioglu et al., 2018) |
| Inductive | Bird Migration Induction
Plant Disease Induction
Sport Feature Induction | Migration Map
Pathology Photos
Sports Activities | eBird (eBird, 2025)
Kaggle (vipoooool, 2024)
Kaggle (Piosenka, 2022) |
| Spatial | Relative Position Estimation
Camera Rotation Estimation
Navigation Route Planning | Interior Views
Everyday Objects
Interior Views | ViewSpatial-Bench (Li et al., 2025a)
NAVI (Jampani et al., 2023)
ViewSpatial-Bench (Li et al., 2025a) |
| Temporal | Crowd Timeline Reconstruction
Driving Sequence Prediction
Human Activity Localization | Crowd Surveillance
Traffic Scene
Domestic Life | Kaggle (fmena14, 2025)
Drivingdojo (Wang et al., 2024)
TVBench (Cores et al., 2025) |

- **High Noise in Input:** In video-based benchmarks, frames are typically sampled from videos and input into the model, which can introduce many irrelevant frames that interfere with reasoning. While this setup is closer to real-world scenarios, our benchmark aims to directly assess the model's reasoning abilities, minimizing interference from other factors.

## C DETAILS OF ANNOTATION PROTOCOLS

This section presents additional details of our task annotation pipeline and protocols, providing complete details for §2.2 of the main paper.

### C.1 DATA SOURCES OF DIFFERENT TASKS

Table 5 presents the data sources for all the tasks included in MMR-Life. During the data collection phase, all annotators strictly adhere to copyright and licensing regulations on the source sites or datasets. Moreover, following Definition 1, we limit the dataset strictly to natural images, explicitly excluding symbolic diagrams and other non-photographic forms.

### C.2 ANNOTATION GUIDELINES

During the annotation of questions and golden answers, all annotators were given the following guidelines:

- All questions must contain multiple images (at least two images).
- All questions should be written in English.
- All questions should be solvable without complex domain-specific knowledge.
- The question should not be ambiguous and can be answered with one option.
- The questions should adhere to the definitions of the respective reasoning types (see §2.2), ensuring clear differentiation between tasks of different reasoning types.

### C.3 PROMPTS FOR NEGATIVE OPTION GENERATION

We list our negative option generation prompts from Figure 11 to Figure 17.

## D    DATA DIVERSITY OF MMR-LIFE

We demonstrate the diversity of data in MMR-Life in this section, where Figure 9 visualizes the variety of image types and Figure 10 presents the distribution of input image counts. The various types of tasks included in our study are illustrated in Appendix E.

## E    TASK DETAILS

In this section, we give a detailed description of each task presented in MMR-Life.

### E.1    ABDUCTIVE REASONING

### E.1.1    HUMAN ACTIVITY ATTRIBUTION

**Task Description.**    This task tests a model's reasoning about human behavior motivations. By observing people's behavior in a given context, the model must analyze environmental clues and behavior cues to select the most plausible motivation among candidate explanations.

**Examples.**    See Figure 19, 20.

### E.1.2    CHARACTER INTERACTION ATTRIBUTION

**Task Description.**    This task requires the model to understand causal relationships between characters (e.g., in Tom & Jerry). Given a scene of interaction, the model must analyze character behaviors and situational factors to infer the most reasonable cause for a specific event or outcome.

**Examples.**    See Figure 21, 22.

### E.1.3    MULTI-HOP COLLISION ATTRIBUTION

**Task Description.**    This task assesses a model's causal reasoning in complex physical collision chains. In scenes involving multiple objects colliding, the model must trace the collision chain and identify the root cause or triggering event for a given outcome.

**Examples.**    See Figure 23, 24.

### E.2    ANALOGICAL REASONING

### E.2.1    ANIMAL RELATION INFERENCE

**Task Description.**    This task requires models to understand visual analogical relationships between animals. Given three animal images, the model must recognize the relational pattern between the first two animals and then select a fourth animal from the options so that the relation between the third and fourth animals matches the original pattern.

**Examples.**    See Figure 25, 26.

### E.2.2    PRODUCT SIMILARITY INFERENCE

**Task Description.**    This task assesses a model's reasoning about product style preference. Based on a person's owned or disliked product samples, the model must analyze design features and style preferences to recommend, from candidate options, a product that best suits their intentions or tastes.

**Examples.**    See Figure 27, 28.

### E.2.3 ARTWORK STYLE INFERENCE

**Task Description.** This task evaluates a model's understanding and recognition of artistic style. Given multiple sample works from the same artist, the model must learn the distinctive stylistic features and then identify which candidate option is most likely also created by that artist.

**Examples.** See Figure 29, 30.

## E.3 CAUSAL REASONING

### E.3.1 CHARACTER INTERACTION PREDICTION

**Task Description.** This task tests a model's ability to predict outcomes of interactions between animated characters. Given a specific behavior or event by a character, the model must use contextual understanding and character relations to predict the most likely follow-up reaction or result.

**Examples.** See Figure 31, 32.

### E.3.2 MULTI-HOP COLLISION PREDICTION

**Task Description.** Given a sequence of consecutive images capturing object motion from initial to current time, the model must reason about the underlying physics and simulate possible multi-stage collision propagation, ultimately predicting the most likely next collision event or chain reaction.

**Examples.** See Figure 33, 34.

### E.3.3 COUNTERFACTUAL FLUID PREDICTION

**Task Description.** This task examines a model's counterfactual reasoning ability in fluid dynamics. The model must analyze how a fluid flows and, if a barrier is removed, predict how the flow would change (i.e., determine the altered flow paths) and final positions under the new condition.

**Examples.** See Figure 35, 36.

## E.4 DEDUCTIVE REASONING

### E.4.1 MATERIAL COMPOSITION DEDUCTION

**Task Description.** This task requires complex combinatorial reasoning about material composition. Given different types and quantities of material components and the material requirements for certain products, the model must calculate how many products can be produced under the current material constraints.

**Examples.** See Figure 37, 38.

### E.4.2 CARD WINNER DEDUCTION

**Task Description.** This task examines a model's understanding of Texas Hold 'em poker rules and logical reasoning. In a multiplayer poker game, each player has hole cards and there are community cards on the board; based on these, the model must analyze the best possible hand for each player and determine the winner.

**Examples.** See Figure 39, 40.

### E.4.3 RECIPE STEP DEDUCTION

**Task Description.** This task requires understanding the logical order of cooking processes. Given a dish name and a set of unordered images depicting stages of preparation, the model must deduce the correct cooking sequence based on ingredient states, tool usage, and causal relationships.

**Examples.** See Figure 41, 42.

## E.5 INDUCTIVE REASONING

### E.5.1 BIRD MIGRATION INDUCTION

**Task Description.** This task requires the model to analyze temporal distribution changes of birds. By observing how bird distributions change over past years, the model must infer migration patterns and predict the likely distribution in the upcoming year.

**Examples.** See Figure 43, 44.

### E.5.2 PLANT DISEASE INDUCTION

**Task Description.** This task evaluates a model's ability to learn disease patterns in plants. Given samples of leaves afflicted with a particular disease, the model must learn the visual features and then identify which candidate leaves also suffer from the same disease.

**Examples.** See Figure 45, 46.

### E.5.3 SPORT FEATURE INDUCTION

**Task Description.** This task tests the model's ability to induce patterns in sports characteristics. Given a series of images depicting sports with certain patterns or rules, the model must understand the characteristic relationships and choose the next sport that best matches the pattern.

**Examples.** See Figure 47, 48.

## E.6 SPATIAL REASONING

### E.6.1 RELATIVE POSITION ESTIMATION

**Task Description.** This task tests a model's spatial relationship reasoning. Given the relative positions of some objects in an indoor scene, the model must infer the relative positions of others and judge directional relationships (e.g. east, west, north, south).

**Examples.** See Figure 49, 50.

### E.6.2 CAMERA ROTATION ESTIMATION

**Task Description.** This task requires the model to analyze viewpoint changes between consecutive images. By comparing the same scene from different angles in the image sequence, the model must accurately estimate the camera's rotation angles and directions at each step.

**Examples.** See Figure 51, 52.

### E.6.3 NAVIGATION ROUTE PLANNING

**Task Description.** This task tests a model's spatial reasoning and path planning capability. A robot must navigate in a given indoor environment from a start point to a goal point. Only 90° or 180° turns and forward moves are allowed, and obstacles must be avoided. The model must plan the correct sequence of moves.

**Examples.** See Figure 53, 54.

### E.7 TEMPORAL REASONING

#### E.7.1 CROWD TIMELINE RECONSTRUCTION

**Task Description.** This task assesses a model's understanding of temporal sequences in complex scenes. Given a set of unordered images of crowd activities, the model must use cues from people's positions, actions, and environmental changes to infer the correct chronological order.

**Examples.** See Figure 55, 56.

#### E.7.2 DRIVING SEQUENCE PREDICTION

**Task Description.** This task evaluates a model's ability to predict time-varying driving scenes. Given a sequence of images from a front-facing cockpit (driver's perspective) view, the model must integrate road geometry, vehicle motions, traffic participants, and environmental cues to predict the most likely next frame.

**Examples.** See Figure 57, 58.

#### E.7.3 HUMAN ACTIVITY LOCALIZATION

**Task Description.** This task asks the model to locate when in a video sequence a particular human activity occurs. Given a video and a description of an activity, the model must precisely predict which time segment (start, middle, end, or throughout) the activity takes place.

**Examples.** See Figure 59, 60.

## F DETAILS OF MAIN EXPERIMENT

### F.1 DETAILED EXPERIMENTAL SETUP

**Multimodal Language Models.** Here, we list all the models used in our experiment and provide the corresponding version (if available): *gpt-5-2025-08-07* (OpenAI, 2025b), *gpt-5-mini-2025-08-07* (OpenAI, 2025b), *gpt-4.1-2025-04-14* (OpenAI, 2025a), *gpt-4.1-mini-2025-04-14* (OpenAI, 2025a), *gpt-4o-2024-11-20* (OpenAI, 2024), *gpt-4o-mini-2024-07-18* (OpenAI, 2024), *o4-mini-2025-04-16* (OpenAI, 2025c), *claude-sonnet-4-20250514* (Anthropic, 2025b), *claude-3-7-sonnet-20250219* (Anthropic, 2025a), *gemini-2.5-flash* (Comanici et al., 2025), *gemini-2.5-pro* (Comanici et al., 2025), *doubao-1-5-vision-pro-32k* (ByteDance Seed Team, 2025), *Kimi-VL-A3B-Thinking-2506* (Du et al., 2025), *Keye-VL-1.5-8B* (Team et al., 2025), *MiMo-VL-7B-RL-2508* (Yue et al., 2025c), *MiMo-VL-7B-SFT-2508* (Yue et al., 2025c), *MM-Eureka-Qwen-7B* (Meng et al., 2025a), *MM-Eureka-Qwen-32B* (Meng et al., 2025a), *OpenVLThinker-7B-v1.2* (Deng et al., 2025), *OpenVLThinker-7B-v1.2-sft-iter3* (Deng et al., 2025), *Qwen2.5-VL-7B-Instruct* (Bai et al., 2025), *Qwen2.5-VL-32B-Instruct* (Bai et al., 2025), *Qwen2.5-VL-72B-Instruct* (Bai et al., 2025), *R1-Onevision-7B* (Yang et al., 2025c), *R1-Onevision-7B-RL* (Yang et al., 2025c), *Skywork-R1V-38B* (Peng et al., 2025), *VL-Rethinker-7B* (Wang et al., 2025b), *VL-Rethinker-32B* (Wang et al., 2025b), *VL-Rethinker-72B* (Wang et al., 2025b), *InternVL3.5-8B* (Wang et al., 2025e), *InternVL3.5-30B-A3B* (Wang et al., 2025e), *InternVL3.5-38B* (Wang et al., 2025e), *gemma-3-4b-it* (Kamath et al., 2025), *gemma-3-12b-it* (Kamath et al., 2025), *gemma-3-27b-it* (Kamath et al., 2025), *QVQ-72B-Preview* (Qwen Team, 2024).

**Parameters.** For parameters during the model's inference. We set the temperature to 0.5, top p to 0.5, and seed to 17.

**Prompts.** The prompt used in the main experiments are illustrated in Figure 18.

### F.2 FULL EXPERIMENTAL RESULTS

We demonstrate full evaluation results on 37 MLLMs in Table 6.

### F.3 Experimental Results on Tiny Set

We present the model performance comparison on the mini test set in Table 7.

## G Details of Thinking Pattern Analysis

For the base setting, we use MiMo-VL-7B-SFT, RL-Onevision, Qwen-2.5-VL-32B and Qwen-2.5-VL-7B (with CoT prompting). For the RL setup, we use the model corresponding to the RL training version for CoT: MiMo-VL-7B-RL, RL-Onevision-RL, MM-Eureka-32B, and VL-Rethinker-7B. These models are trained on various datasets to illustrate the generalizability of our conclusions.

## H Case Study

We further provide additional case studies as shown from Figure 19 to Figure 60, showing both correct and incorrect responses by GPT-5 and Gemini-2.5-Pro.

Table 6: Full performance comparison of SOTA MLLMs on MMR-Life.

| Model | Abd | Ana | Cau | Ded | Ind | Spa | Tem | Avg |
|---|---|---|---|---|---|---|---|---|
| gpt-5 | 53.75 | 78.87 | 41.06 | 80.14 | 78.32 | 17.25 | 41.70 | 58.69 |
| gemini-2.5-pro | 54.40 | 73.77 | 36.99 | 79.43 | 73.66 | 25.10 | 35.79 | 56.86 |
| gemini-2.5-flash | 46.25 | 75.18 | 34.22 | 71.63 | 73.66 | 23.92 | 30.81 | 53.10 |
| o4-mini | 41.37 | 73.59 | 27.38 | 71.28 | 68.07 | 19.22 | 32.66 | 50.49 |
| gpt-5-mini | 44.95 | 69.72 | 32.32 | 75.18 | 68.76 | 12.16 | 29.52 | 49.77 |
| gpt-4.1 | 44.30 | 71.30 | 22.43 | 67.38 | 70.16 | 13.73 | 27.31 | 48.15 |
| claude-sonnet-4-thinking | 36.96 | 60.92 | 44.11 | 67.02 | 56.64 | 15.69 | 28.23 | 45.32 |
| claude-3.7-sonnet | 33.55 | 66.55 | 35.36 | 59.93 | 59.67 | 20.78 | 26.01 | 45.09 |
| gpt-4o | 46.91 | 65.67 | 25.86 | 51.42 | 66.20 | 11.37 | 26.01 | 44.75 |
| gpt-4.1-mini | 32.90 | 61.62 | 30.80 | 52.13 | 65.27 | 16.47 | 30.63 | 44.10 |
| claude-sonnet-4 | 35.50 | 57.22 | 38.02 | 64.89 | 55.71 | 14.51 | 25.83 | 42.82 |
| Qwen2.5-VL-72B | 35.50 | 55.46 | 35.36 | 52.13 | 55.48 | 12.94 | 23.80 | 40.21 |
| doubao-1.5-vision | 37.13 | 53.70 | 31.18 | 59.57 | 54.31 | 12.16 | 23.06 | 39.98 |
| VL-Rethinker-72B | 36.48 | 50.88 | 33.08 | 56.03 | 57.58 | 15.69 | 21.59 | 39.68 |
| Gemma3-27B | 35.18 | 57.92 | 36.88 | 31.21 | 60.61 | 12.94 | 18.27 | 38.32 |
| gpt-4o-mini | 23.45 | 55.28 | 20.15 | 31.21 | 62.00 | 12.55 | 16.42 | 34.54 |
| QVQ-72B-Preview | 31.27 | 41.20 | 38.02 | 47.87 | 31.24 | 14.12 | 16.42 | 31.14 |
| VL-Rethinker-32B | 22.98 | 42.96 | 28.14 | 49.65 | 30.77 | 14.90 | 19.19 | 30.29 |
| MM-Eureka-Qwen-7B | 31.27 | 42.25 | 20.91 | 36.17 | 37.53 | 13.73 | 17.34 | 29.59 |
| Gemma3-12B | 25.08 | 50.70 | 17.11 | 27.30 | 42.42 | 10.20 | 15.87 | 29.52 |
| MM-Eureka-Qwen-32B | 26.06 | 41.02 | 25.10 | 47.52 | 27.97 | 16.08 | 17.34 | 29.02 |
| R1-Onevision-7B | 28.34 | 37.85 | 23.57 | 25.89 | 42.66 | 14.51 | 19.37 | 28.80 |
| MiMo-VL-7B-RL | 38.76 | 25.88 | 28.14 | 60.99 | 24.94 | 14.12 | 19.19 | 28.68 |
| Qwen2.5-VL-32B | 23.45 | 42.78 | 21.29 | 50.00 | 27.27 | 15.69 | 16.24 | 28.61 |
| VL-Rethinker-7B | 30.62 | 40.49 | 21.29 | 28.72 | 43.59 | 13.73 | 11.81 | 28.23 |
| MiMo-VL-7B-SFT | 36.81 | 23.24 | 27.00 | 63.48 | 23.31 | 12.16 | 18.08 | 27.36 |
| Qwen2.5-VL-7B | 26.06 | 35.74 | 20.53 | 20.92 | 38.93 | 9.41 | 12.18 | 24.68 |
| R1-Onevision-7B-RL | 24.76 | 35.04 | 22.05 | 25.53 | 30.30 | 12.16 | 12.73 | 24.00 |
| InternVL3_5-30B-A3B | 45.60 | 19.19 | 33.46 | 36.52 | 14.45 | 12.16 | 14.39 | 23.09 |
| InternVL3_5-38B | 46.25 | 15.67 | 26.24 | 41.13 | 5.59 | 14.51 | 18.27 | 21.77 |
| Gemma3-4B | 14.66 | 29.75 | 20.91 | 26.60 | 23.31 | 12.16 | 17.34 | 21.50 |
| Keye-VL-1.5-8B | 19.87 | 21.30 | 23.95 | 14.18 | 20.28 | 13.73 | 23.62 | 20.22 |
| InternVL3_5-8B | 35.18 | 11.44 | 18.63 | 34.04 | 11.19 | 14.90 | 16.61 | 18.67 |
| Kimi-VL-A3B-Thinking-2506 | 24.10 | 12.85 | 18.25 | 37.23 | 10.96 | 12.94 | 18.08 | 18.07 |
| OpenVLThinker-7B-v1.2 | 16.94 | 19.37 | 20.91 | 12.06 | 18.18 | 17.25 | 18.27 | 17.84 |
| OpenVLThinker-7B-v1.2-sft | 16.29 | 19.19 | 21.29 | 13.83 | 18.41 | 17.65 | 17.16 | 17.80 |
| Skywork-R1V-38B | 22.15 | 10.39 | 16.73 | 23.76 | 11.89 | 9.80 | 11.07 | 14.13 |

Table 7: Performance comparison of SOTA MLLMs on MMR-Life mini set.

| Model | Abd | Ana | Cau | Ded | Ind | Spa | Tem | Avg |
|---|---|---|---|---|---|---|---|---|
| Human | 79.76 | 57.65 | 75.00 | 70.59 | 63.41 | 79.76 | 79.76 | 72.28 |
| gpt-5 | 63.33 | 66.67 | 53.33 | 73.33 | 66.67 | 30.00 | 50.00 | 57.62 |
| gemini-2.5-pro | 63.33 | 80.00 | 60.00 | 76.67 | 50.00 | 30.00 | 36.67 | 56.67 |
| gemini-2.5-flash | 53.33 | 63.33 | 50.00 | 53.33 | 50.00 | 50.00 | 53.33 | 53.33 |
| o4-mini | 46.67 | 73.33 | 43.33 | 73.33 | 50.00 | 23.33 | 43.33 | 50.48 |
| gpt-5-mini | 50.00 | 56.67 | 40.00 | 83.33 | 50.00 | 6.67 | 33.33 | 45.71 |
| claude-sonnet-4 | 40.00 | 66.67 | 60.00 | 66.67 | 30.00 | 16.67 | 56.67 | 48.10 |
| gpt-4.1 | 50.00 | 56.67 | 36.67 | 16.67 | 63.33 | 6.67 | 26.67 | 36.67 |
| claude-3.7-sonnet | 30.00 | 73.33 | 46.67 | 70.00 | 40.00 | 13.33 | 43.33 | 45.24 |
| gpt-4o | 43.33 | 50.00 | 40.00 | 16.67 | 46.67 | 20.00 | 26.67 | 34.76 |
| gpt-4.1-mini | 33.33 | 63.33 | 36.67 | 53.33 | 50.00 | 10.00 | 26.67 | 39.05 |
| doubao-1.5-vision | 50.00 | 50.00 | 36.67 | 63.33 | 40.00 | 13.33 | 23.33 | 39.52 |
| VL-Rethinker-72B | 26.67 | 50.00 | 40.00 | 63.33 | 33.33 | 13.33 | 23.33 | 35.71 |
| QVQ-72B-Preview | 26.67 | 53.33 | 50.00 | 36.67 | 26.67 | 3.33 | 16.67 | 30.48 |
| MM-Eureka-Qwen-32B | 20.00 | 46.67 | 43.33 | 50.00 | 33.33 | 16.67 | 36.67 | 35.24 |
| MiMo-VL-7B-RL | 36.67 | 16.67 | 40.00 | 70.00 | 16.67 | 13.33 | 26.67 | 31.43 |
| VL-Rethinker-7B | 23.33 | 36.67 | 13.33 | 13.33 | 33.33 | 0.00 | 16.67 | 22.78 |
| Keye-VL-1.5-8B | 13.33 | 23.33 | 33.33 | 20.00 | 26.67 | 13.33 | 26.67 | 22.38 |
| Skywork-R1V-38B | 30.00 | 6.67 | 33.33 | 16.67 | 30.00 | 16.67 | 13.33 | 20.95 |
| Qwen2.5-VL-72B | 36.67 | 40.00 | 40.00 | 20.00 | 36.67 | 10.00 | 30.00 | 30.48 |
| Gemma3-27B | 20.00 | 36.67 | 33.33 | 30.00 | 40.00 | 6.67 | 30.00 | 28.10 |
| Gemma3-12B | 30.00 | 50.00 | 16.67 | 33.33 | 20.00 | 6.67 | 23.33 | 25.71 |
| Qwen2.5-VL-32B | 23.33 | 30.00 | 30.00 | 20.00 | 20.00 | 6.67 | 23.33 | 21.90 |
| Qwen2.5-VL-7B | 13.33 | 0.00 | 23.33 | 20.00 | 0.00 | 6.67 | 10.00 | 14.67 |
| InternVL3.5-30B-A3B | 33.33 | 13.33 | 40.00 | 26.67 | 16.67 | 10.00 | 16.67 | 22.38 |
| InternVL3.5-8B | 30.00 | 6.67 | 20.00 | 16.67 | 20.00 | 13.33 | 10.00 | 16.67 |

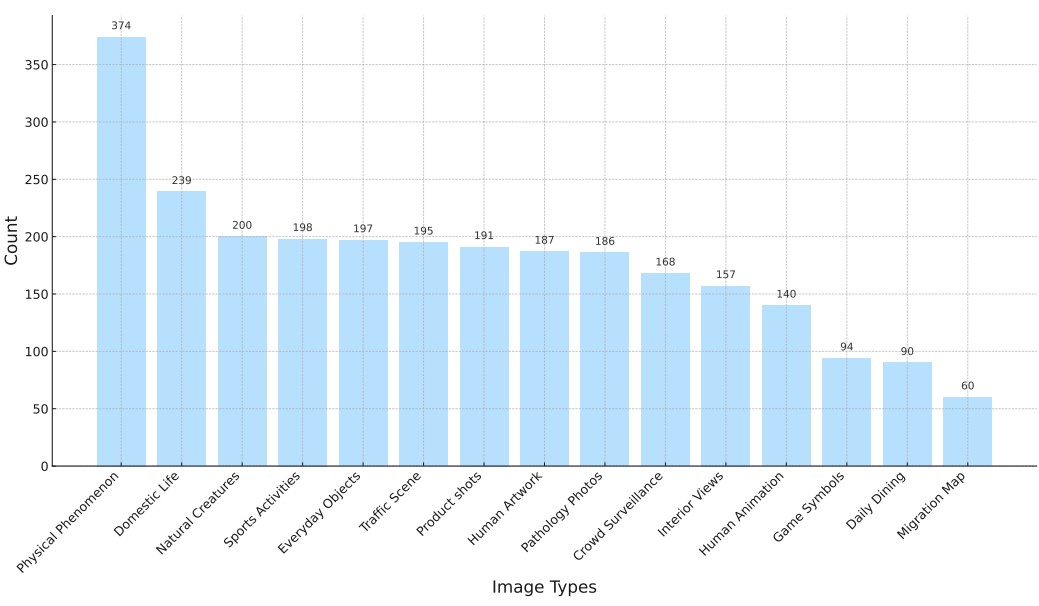

Figure 9: Image type distribution in MMR-Life.

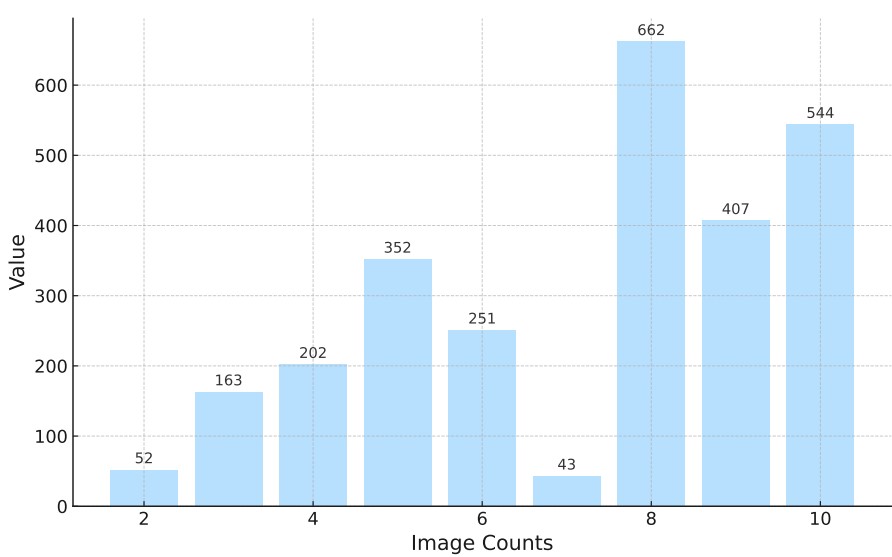

Figure 10: Image counts distributions in MMR-Life.

**Negative Option Generation Prompt I**

[Task]

Please act as a reasoning expert to answer the given multimodal reasoning questions based on multiple images. The last line of your response should be of the following format:

[Output]

Answer:

Let's think step by step before answering.

Figure 11: Negative option generation prompt.

**Negative Option Generation Prompt II**

[Task]

Please act as a reasoning expert to answer the given multimodal reasoning questions based on multiple images. The last line of your response should be of the following format:

[Output]

Answer: (Event 1, Event 2)

Let's think step by step before answering.

Figure 12: Negative option generation prompt.

**Negative Option Generation Prompt III**

[Task]

Please act as a reasoning expert to answer the given multimodal reasoning questions based on multiple images. The last line of your response should be of the following format:

[Output]

Answer: x-x-x-x-x..., where 'x' is the number corresponding to the image.

Let's think step by step before answering.

Figure 13: Negative option generation prompt.

**Negative Option Generation Prompt IV**

[Task]

Please act as a reasoning expert to answer the given multimodal reasoning questions based on multiple images. The last line of your response should be of the following format:

[Output]

Answer: Rotate clockwise/counterclockwise about x degrees(, then clockwise/counterclockwise about x degrees)(, and finally counterclockwise about x degrees), where x represents the degrees.

Let's think step by step before answering.

Figure 14: Negative option generation prompt.

**Negative Option Generation Prompt V**

[Task]

Please act as a reasoning expert to answer the given multimodal reasoning questions based on multiple images. The last line of your response should be of the following format:

[Output]

Answer: DIRECTION, where DIRECTION represents the eight common directions.

Let's think step by step before answering.

Figure 15: Negative option generation prompt.

**Negative Option Generation Prompt VI**

[Task]

Please act as a reasoning expert to answer the given multimodal reasoning questions based on multiple images. The last line of your response should be of the following format:

[Output]

Answer: 1. ACTION 2. ACTION ..., where ACTION is limited to the following actions: Turn Left, Turn Right, Turn Back, Go forward until the xxx. You should add numbering (e.g., 1., 2., 3. ...) before each action in the answer. Answer: 1. ACTION 2. ACTION ..., where ACTION is limited to the following actions: Turn Left, Turn Right, Turn Back, Go forward until the xxx. You should add numbering (e.g., 1., 2., 3. ...) before each action in the answer.

Let's think step by step before answering.

Figure 16: Negative option generation prompt.

**Negative Option Generation Prompt VII**

[Task]

Please act as a reasoning expert to answer the given multimodal reasoning questions based on multiple images. The last line of your response should be of the following format:

[Output]

Answer: COLOR stick/container(, COLOR stick/container)(, COLOR stick/container), where COLOR represents different colors.

Let's think step by step before answering.

Figure 17: Negative option generation prompt.

**Question Answer With CoT Prompt**

[Task]

Please act as a reasoning expert to choose the correct option of the given multimodal reasoning questions based on multiple images. The last line of your response should be of the following format:

[Output]

Answer: A/B/C/D/E.

Let's think step by step before answering.

Figure 18: Prompt used in the main experiment.

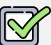 **Abductive: Human Activity Attribution**

**Question:** Why does the man look to the left after shaving? Please choose the best explanation based on the given images.
**Options:**
(A) To check his shave.
(B) He looks to the left after shaving to ensure thorough grooming.
(C) He likely looks to the left after shaving because he is preparing to leave the area.
(D) He looks to the left to grab his shirt to put on after shaving.
(E) He likely looks to respond to something in his environment.

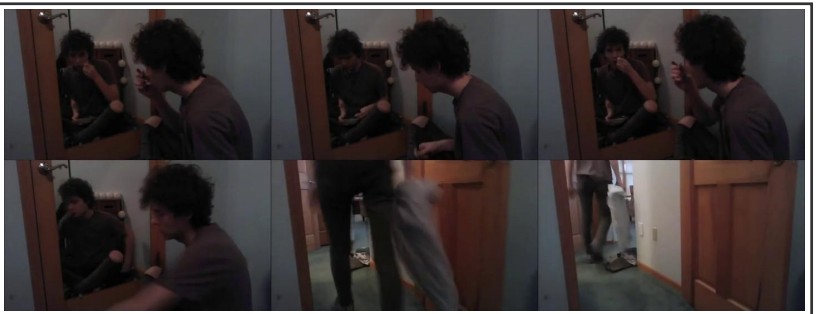

<input images>

**Gemini-2.5-Pro:** Step 1: Analyze the initial action. The first four images show a person sitting in front of a mirror, engaged in what appears to be a grooming activity, possibly shaving or applying something to their face. They are focused on their reflection. Step 2: Identify the key change in action. In the fourth image, the person turns their head to the left, away from the mirror. The question asks for the reason for this specific action. Step 3: Observe the subsequent actions. The fifth and sixth images show what happens immediately after the person looks to the left. They get up, and in the fifth image, they are seen grabbing a light-colored shirt or piece of clothing that was located to their left. The sixth image shows them walking away with this item of clothing. Step 4: Connect the look to the subsequent action. The person's look to the left directly precedes the action of getting up and grabbing the shirt from that same direction. This indicates a clear cause-and-effect relationship: they looked towards the object they intended to interact with next. Step 5: Evaluate the given options based on the evidence. A. To check his shave: This is unlikely, as he turned his head away from the mirror, which is necessary for checking one's shave. B. He looks to the left after shaving to ensure thorough grooming: Similar to A, looking away from the mirror would not help with grooming. C. He likely looks to the left after shaving because he is preparing to leave the area: This is partially true, but it's not the most specific reason. His immediate action is not just leaving, but grabbing something first. D. He looks to the left to grab his shirt to put on after shaving: This is the most precise explanation. The images clearly show him looking left and then immediately grabbing a shirt from that location. This action is a direct follow-up to his look. E. He likely looks to respond to something in his environment: While possible, there is no evidence in the images of an external stimulus (like a sound or another person). The most direct evidence points to his own planned action. Step 6: Conclude the best option. Option D provides the most accurate and specific explanation for why the man looked to the left, as it is directly supported by the visual sequence of events shown in the images.# Answer: D

**Ground Truth: (D) He looks to the left to grab his shirt to put on after shaving.**

Figure 19: A correct example of Human Activity Attribution task.

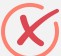 **Abductive: Human Activity Attribution**

**Question:** Why does the man reach for the table? Please choose the best explanation based on the given images.
**Options:**
(A) To set his drink down on the table.
(B) To use it as a surface to hold or manage items.
(C) To either obtain a detergent for mopping
(D) To pick up the cup with milk.
(E) To gain balance while handling the broom.

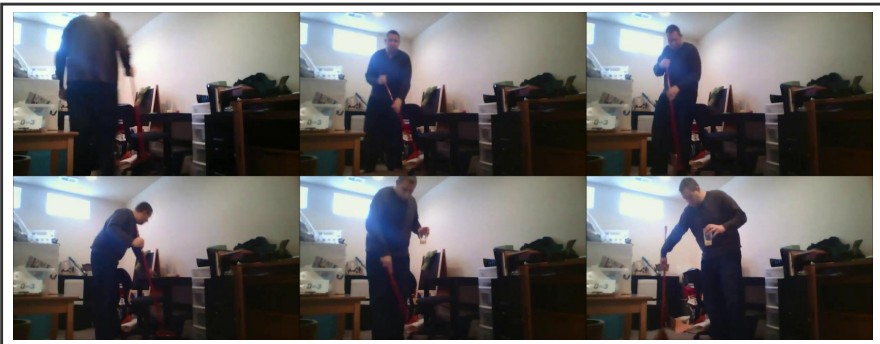

<input images>

**GPT-5:** The sequence shows the man sweeping with a broom.- Then he appears holding a drink in one hand while still handling the broom.- He leans toward the small table with the drink in his hand, consistent with placing it down so he can continue sweeping.- There's no sign of detergent, balancing, or milk specifically.# Answer: A"

**Error Category: Perception Error (Motion and State Confusion)**
**Error Reason:** Instead of perceiving the man picking up the cup (the correct action), GPT-5 incorrectly perceives him as placing the cup down.

**Ground Truth: (D) To pick up the cup with milk.**
**Explanation:** From the next three frames we can see that the man reaches into the table because he wants to pick up a glass of milk, not put the drink down.

Figure 20: An error example of Human Activity Attribution task.

### ☑ Abductive: Character Interaction Attribution

**Question:** Why did Tom cover its eyes and start smoking in the end? Please choose the best explanation based on the given images.
**Options:**
(A) Tom covered his eyes and started smoking simply to imitate a cowboy stereotype, not because of the bull's attack.
(B) Tom covered his eyes because he got dust and sand in them during the scuffle, and smoking was just an unrelated gag added afterward.
(C) Because he was upset and embarrassed (defeated by Jerry), so he covered his eyes in shame and lit a cigarette to calm himself.
(D) Because Tom had no way out and wanted to calmly wait for the cow's impact./Tom covered his eyes and started smoking due to being overwhelmed and stressed after narrowly escaping the charging bull.
(E) Tom covered his eyes and smoked because he thought the show was over and wanted to make a dramatic, theatrical exit like an old Western movie character.

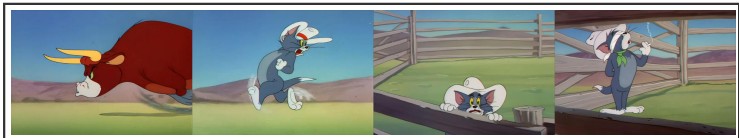

<input images>

**Gemini-2.5-Pro:** Step 1: Analyze the sequence of events in the images. Image 1 shows a very angry bull charging. This establishes a clear and immediate danger. Image 2 shows Tom running away in extreme fear. His expression and posture indicate he is being chased by the bull from the first image. Image 3 shows Tom hiding behind a wooden fence, peeking over with a look of continued fear and anxiety. He has found temporary refuge, but the threat is likely still present. Image 4 shows a shift in Tom's behavior. He is now dressed as a cowboy, smoking a cigarette in a feigned cool and nonchalant manner. However, he is simultaneously covering his eyes with his other hand. Step 2: Interpret Tom's final action in Image 4.  The act of smoking, especially in a cowboy outfit, is a trope for appearing cool and unfazed. The act of covering one's eyes is a classic gesture of fear, indicating an inability or unwillingness to watch a terrifying event unfold. Combining these two contradictory actions creates a comedic effect. Tom is trying to pretend he is calm and in control, but his underlying terror from the bull chase is so overwhelming that he can't even bear to look. He has narrowly escaped the immediate danger by getting behind the fence, but the stress and the lingering threat (the bull is likely about to smash through the fence) have pushed him to this state. =Step 3: Evaluate the given options based on the analysis. A:  This option is incomplete. While he is imitating a cowboy stereotype, it's a direct reaction to the bull's attack, which is the primary motivator for his actions.  B:  There is no visual evidence to suggest dust or sand in his eyes. The gesture is a clear \"I can't look\" pose, not one of irritation.  C:  The images provided do not include Jerry. The conflict shown is solely between Tom and the bull. Attributing his actions to Jerry is an assumption not supported by the evidence.  D:  This option accurately captures the duality of Tom's state. He has narrowly escaped but is still in immense danger or at least suffering from the extreme stress of the chase. He is \"overwhelmed and stressed.\" The smoking is an attempt at a calm facade, while covering his eyes reveals his true terror, as he anticipates the bull's impact or is simply unable to process the stressful situation. This aligns perfectly with the visual narrative.  E:  This is a meta-interpretation that breaks the fourth wall. The character's actions are driven by the in-story events (the bull), not by an awareness of being in a cartoon. Step 4: Conclude the best explanation. Option D provides the most logical and comprehensive explanation for Tom's behavior. It correctly identifies the cause (the charging bull), the emotional state (overwhelmed and stressed), and the meaning behind his contradictory actions (feigning calm while being terrified).  # Answer: D"

**Ground Truth:** (D) Because Tom had no way out and wanted to calmly wait for the cow's impact./Tom covered his eyes and started smoking due to being overwhelmed and stressed after narrowly escaping the charging bull.

Figure 21: A correct example of Character Interaction Attribution task.

**⊗ Abductive: Character Interaction Attribution**

**Question:** Why did the appearance of the yellow cat change? Please choose the best explanation based on the given images.
**Options:**
(A) The yellow cat's appearance changed because it got tangled in the toy plane's propeller, twisting its body into a distorted shape.
(B) The yellow cat's appearance changed because paint or slime from the toy splattered onto it, making it look different and exaggerated.
(C) Because it was squashed/hit by the falling toy (plane/balloon), which flattened and deformed its appearance.
(D) The yellow cat's appearance changed because it got wrapped inside the toy plane's parachute, puffing it up in a cloth-like covering.
(E) It swallowed the watermelon that was used to attack them, and its body transformed into the shape of a watermelon.

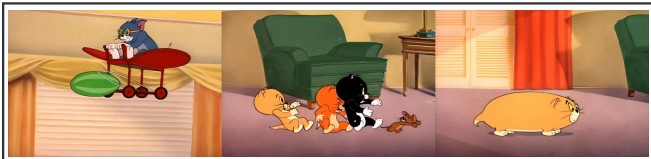

<input images>

**Gemini-2.5-Pro:** Step 1: Analyze the first image. The first image shows Tom the cat in a red toy airplane, flying indoors. A large green watermelon is attached to the bottom of the plane, positioned as if it's a bomb to be dropped. Tom is wearing pilot goggles and has a mischievous look, suggesting he is on a mission. Step 2: Analyze the second image. The second image shows three kittens and Jerry the mouse on the floor, looking up in alarm and scrambling to get away. Their expressions and postures indicate that something is coming down towards them from above. This links directly to the first image, where Tom is flying above. Step 3: Analyze the third image. The third image shows one of the yellow/orange kittens from the second image. Its body has been drastically altered; it is now extremely flat and elongated, almost like a pancake. It has a dazed expression. The background is the same room, confirming it's the same scene. Step 4: Synthesize the information and evaluate the options. We can infer a sequence of events: Tom, in his plane (Image 1), drops the watermelon on the kittens below. The kittens see it coming and try to flee (Image 2). One of the kittens is hit by the falling object, resulting in it being squashed flat (Image 3). Option A: The cat is flattened, not twisted as if by a propeller. This is incorrect. Option B: The cat's shape is changed, it's not just covered in paint or slime. This is incorrect. Option C: This option states the cat was squashed/hit by a falling toy (the watermelon from the plane), which flattened and deformed its appearance. This perfectly matches the logical sequence of events inferred from the images. The cause (falling object) and the effect (flattened cat) are consistent with the visual evidence. Option D: The cat is flattened, not puffed up or wrapped in a parachute. This is incorrect. Option E: The cat is flat, not round like a watermelon. It clearly didn't swallow the watermelon. This is incorrect. Step 5: Conclude the best explanation. Based on the step-by-step analysis, the most logical and accurate explanation for the yellow cat's change in appearance is that it was hit and flattened by the watermelon dropped from Tom's toy plane #Answer: C

**Error Category: Perceptual Error (Color and Visual Confusion)**
**Error Reason:** The cat ends up being round like a watermelon, not flat. However, the image information that Gemini-2.5-Pro didn't analyze says the exact opposite.

**Ground Truth: (E) It swallowed the watermelon that was used to attack them, and its body transformed into the shape of a watermelon.**
**Explanation:** We can see in Figure 1 that Tom wants to attack Jerry with a watermelon, and then in Figure 3 Jerry's body becomes an oval (the shape of a watermelon).

Figure 22: An error example of Character Interaction Attribution task.

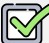 **Abductive: Multi-Hop Collision Attribution**

**Question:** What event is responsible for the silver cylinder moves to the left?
**Options:**
(A) The green ball collides with the cyan cylinder
(B) The gray cylinder bumps into the brown ball
(C) The brown ball was pushed to the west
(D) The green ball came from the right
(E) The blue ball collides with the silver cylinder

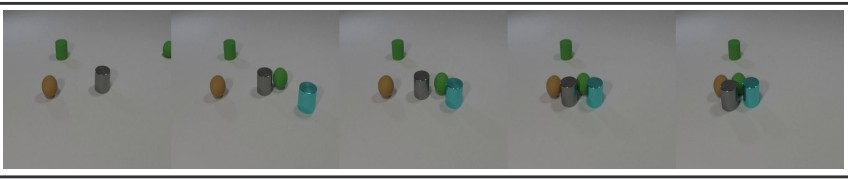

\<input images\>

**GPT-5:** Initially, the silver cylinder is in the center, the brown ball is on its left, and the green ball is on the far right.- A cyan cylinder appears to the right and pushes the green ball leftward.- The green ball approaches from the right and presses against the silver cylinder.- This contact causes the silver cylinder to shift left, later nearing the brown ball.Therefore, the event responsible is the green ball coming from the right and pushing the silver cylinder leftward. # Answer: D

**Ground Truth: (D) The green ball came from the right**

Figure 23: A correct example of Multi-Hop Collision Attribution task.

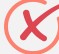 **Abductive: Multi-Hop Collision Attribution**

**Question:** What event was the catalyst for the green cylinder moves?
**Options:**
(A) The red cylinder bumps into the gray ball
(B) The purple sphere slows down a bit
(C) The red cylinder collides with the purple ball
(D) The red cylinder moves away.
(E) The purple cylinder bumps into the gray ball.

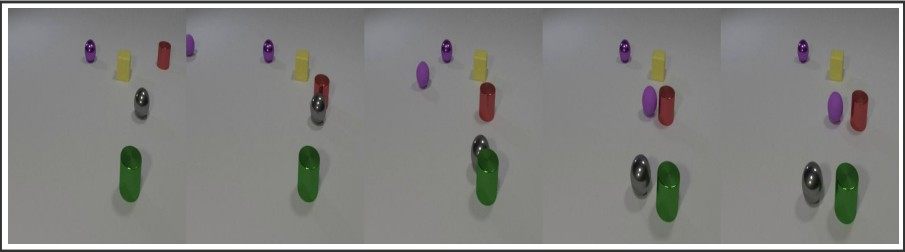

<input images>

**GPT-5:** Initially, the objects are static along a diagonal: purple sphere (matte) at left, then the red cylinder, gray ball, and finally the green cylinder. The red cylinder and gray ball are already near each other, but the green cylinder doesn't move. When the purple sphere reaches and collides with the red cylinder, it pushes the red cylinder, which in turn pushes the gray ball into the green cylinder. This chain reaction causes the green cylinder to start moving. # Answer: C"

**Error Category: Reasoning Error (Flawed Causal Reasoning)**
**Error Reason:** When the purple sphere arrives and collides with the red cylinder, the green cylinder has already started moving, confusing cause and effect.

**Ground Truth: (A) The red cylinder bumps into the gray ball**
**Explanation:** As can be seen from the picture, after the red cylinder comes out from the upper right corner, it first collides with the gray ball, and then the gray ball collides with the green cylinder, causing the green cylinder to start moving.

Figure 24: An error example of Multi-Hop Collision Attribution task.

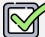 **Analogical: Animal Relation Inference**

**Question:** You are presented with three animal pictures. Select the most appropriate fourth animal so that the relationship between the first pair is analogous to the relationship between the second pair.
**Options:**

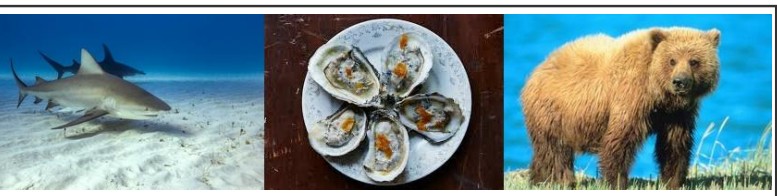

\<input images\>

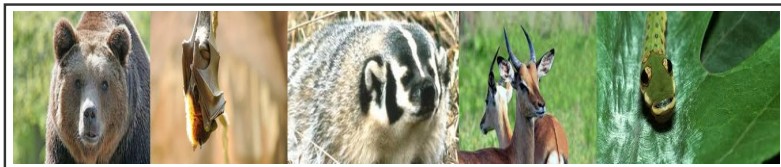

\<options images（A-E）\>

**Gemini-2.5-Pro:** Step 1: Analyze the relationship between the first pair of images. The first image shows sharks, which are large, predatory, vertebrate animals living in a marine environment. They are high on the food chain. The second image shows oysters, which are small, invertebrate (mollusk) animals. They are filter feeders, low on the food chain, and often consumed as food. They also live in a marine environment. The core relationship is between a large, predatory vertebrate and a small, invertebrate food source from the same general habitat (marine). Step 2: Analyze the third image.   The third image shows a brown bear, which is a large, omnivorous (often predatory), vertebrate animal living in a terrestrial environment. It is high on the food chain. Step 3: Apply the relationship from Step 1 to the third image to find the fourth image.   We are looking for an animal that has a similar relationship to the bear as the oyster has to the shark. Following the analogy, we need a small, invertebrate food source from a terrestrial environment. Step 4: Evaluate the given options.  Option A (Caterpillar):   A caterpillar is a small, terrestrial invertebrate (insect larva). Bears are omnivores and are known to eat insects, including caterpillars and grubs. This fits the analogy perfectly: large terrestrial vertebrate (bear) vs. small terrestrial invertebrate food source (caterpillar).   Option B (Impala):   An impala is a large herbivorous vertebrate. While it could be prey for some predators, it does not fit the invertebrate or small size criteria established by the first pair.   Option C (Badger):   A badger is a medium-sized omnivorous vertebrate. It is not an invertebrate and does not fit the \"small food source\" role relative to a bear. Option D (Bat):   A bat is a small flying vertebrate. It is not an invertebrate.   Option E (Bear):   This is another bear. The relationship in the first pair is between two very different types of animals, not the same type. Step 5: Conclude the best fit.   The relationship is: Large Vertebrate Predator : Small Invertebrate Food Source. Pair 1: Shark (Large Marine Vertebrate) : Oyster (Small Marine Invertebrate). Pair 2: Bear (Large Terrestrial Vertebrate) : Caterpillar (Small Terrestrial Invertebrate). Option A is the only one that maintains this analogous relationship.# Answer: A"

**Ground Truth: (A)**

Figure 25: A correct example of Animal Relation Inference task.

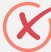 **Analogical: Animal Relation Inference**

**Question:** Consider a sequence of three animal images. Identify which animal should appear as the fourth image to maintain the same relational pattern observed between the first two and the last two images.
**Options:**

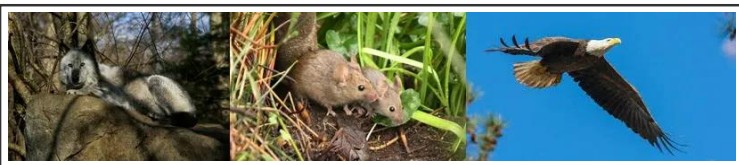

<input images>

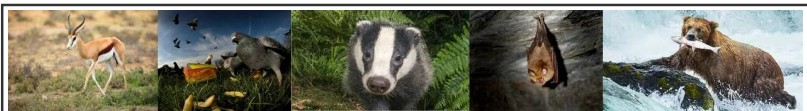

<options images (A-E) >

**Gemini-2.5-Pro:** Step 1: Analyze the relationship between the first two images. The first image shows a wolf. A wolf is a carnivorous predator. The second image shows mice. Mice are small mammals that are often prey for various predators, including wolves (though not their primary food source, the predator-prey relationship exists). Therefore, the relationship between the first and second images is Predator : Prey. Step 2: Apply this relationship to the third and fourth images. The third image shows a bald eagle. A bald eagle is a bird of prey, a predator. Following the established pattern, the fourth image should depict a common prey animal for a bald eagle. Step 3: Evaluate the options as potential prey for a bald eagle. Bald eagles are known for their diet, which primarily consists of fish. They also eat other birds, small mammals, and carrion. Option A (Springbok): This is a medium-sized antelope found in Africa. It is not prey for a bald eagle, which is native to North America. Option B (Pigeons): While eagles might occasionally hunt pigeons, they are not their primary food source. Option C (Badger): A badger is a formidable mammal and is not prey for a bald eagle. Option D (Bat): Bats can be prey for some raptors, but they are not a typical or primary food source for bald eagles. Option E (Bear with a fish): This image depicts a bear that has caught a fish (likely a salmon). Fish, especially salmon, are the primary and most iconic food source for bald eagles. Although the image also shows another predator (a bear), it is the only option that clearly features the main prey of a bald eagle. The logical connection is between the eagle (predator) and the fish (prey). Step 4: Conclude the best fit. The relationship is Predator (Wolf) : Prey (Mice). To maintain this relationship, we need Predator (Eagle) : Prey. The most fitting prey for an eagle among the options is the fish shown in image E.# Answer: E

**Error Category: Perceptual Error (Color and Visual Confusion)**
**Error Reason:** Gemini-2.5-Pro may focus only on local details (e.g., a fish) while ignoring broader image relationships

**Ground Truth: (B)**
**Explanation:** The relationship between wolf and mouse, hawk and dove is essentially one of predator and prey.

Figure 26: An error example of Animal Relation Inference task.

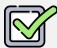 **Analogical: Product Similarity Inference**

**Question:** Since my friend already owns these shoes but wants a comparable pair, which of the following should I give him?
**Options:**

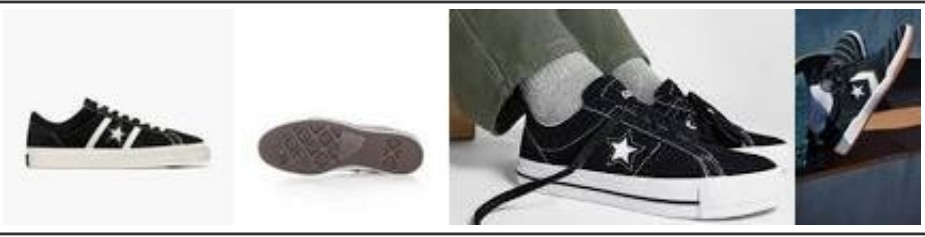

<input images>

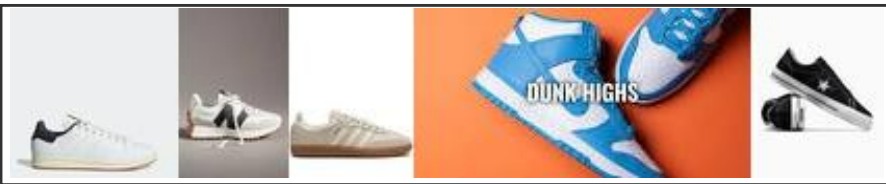

<options images (A-E) >

**Gemini-2.5-Pro:** Observe the original shoes: black, low-top skate style with a white star and two white side stripes — the Converse One Star/Academy style.- Compare options: - A: White tennis shoe (Stan Smith) — different silhouette and branding. - B: Retro runner — different style and sole. - C: Adidas Samba — soccer-inspired with three stripes, not a star. - D: High-top Nike Dunk — different cut and look. - E: Black low-top with the same star-and-bars design — essentially the same/contemporary comparable pair.Therefore, the most comparable choice is E.# Answer: E

**Ground Truth: (E)**

Figure 27: A correct example of Product Similarity Inference task.

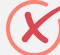 **Analogical: Product Similarity Inference**

**Question:** My friend already owns these pairs of shoes, but he prefers other styles. Which of the following shoes would be the best gift for him?
**Options:**

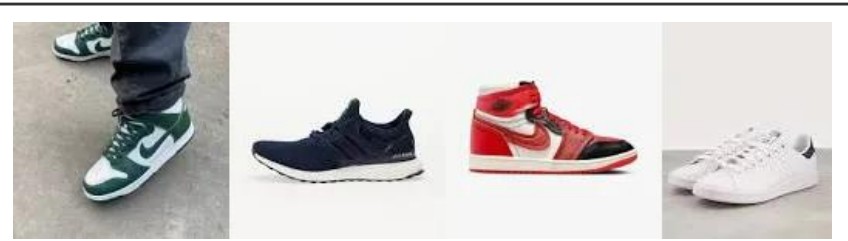

\<input images\>

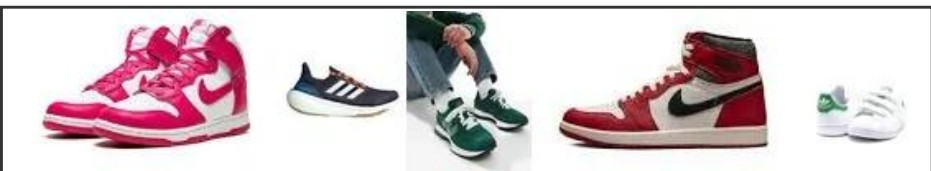

\<options images（A-E）\>

**GPT-5:** The first image shows the pair he already owns: a Nike Dunk-style low-top leather sneaker. - He prefers other styles, as suggested by the subsequent images (running shoe, high-top Jordan, minimalist Stan Smith). - Among the options:  - A is another Dunk (same style he already has) — not ideal.  - D is a Jordan 1 — silhouette is very similar to a Dunk.  - E is a white leather Stan Smith with velcro — closer in feel to his current low-top leather style and also a bit off due to velcro.  - B is an Adidas performance running shoe — clearly a different style he prefers.  - C (New Balance) isn't among the shown preferences.  So the best gift that matches a different preferred style is the Adidas running shoe.  # Answer: B

**Error Category: Understanding Error（Rule or Question Misunderstanding）**
**Error Reason:** GPT-5 limits the scope of interpretation by narrowing "other styles" to "preferences shown only in the figure," without considering that option C (New Balance) is actually the correct "other styles" given in the question.

**Ground Truth: (C)**
**Explanation:** We can see that the shoes in the first and fourth pictures in the options are similar to the friend's first and third pairs of shoes, while the second and fifth pairs of shoes in the options are similar to the friend's second and fourth pairs of shoes respectively, so only the third pair of shoes meets the different styles.

Figure 28: An error example of Product Similarity Inference task.

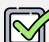 **Analogical: Artwork Style Inference**

**Question:** Here are some paintings from one artist. Choose the option that you think was painted by the same artist.
**Options:**

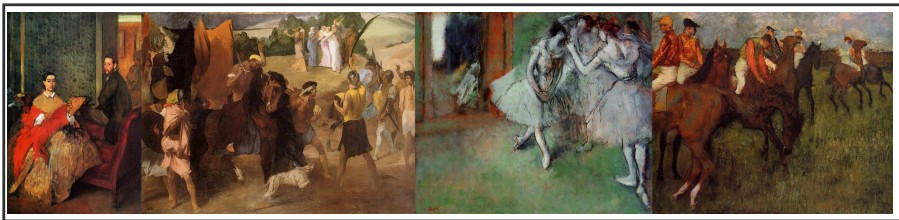

<input images>

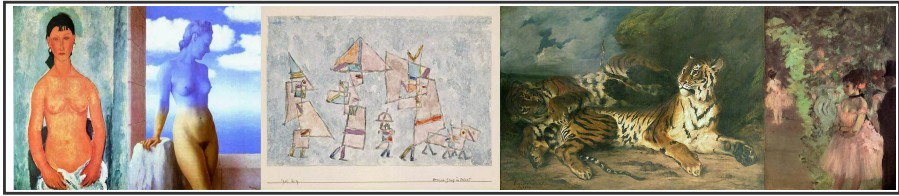

<options images (A-E) >

**GPT-5:** The given set includes ballet dancers and jockeys on horseback, both hallmark subjects of Edgar Degas. The loose, sketch-like brushwork, unusual cropping, and muted palette further point to Degas, who also painted occasional historical scenes.- Among the options, E shows dancers backstage with the same impressionist handling and composition typical of Degas.- The other choices match different artists: A (Modigliani), B (Magritte), C (Klee), D (Delacroix).#
Answer: E"

**Ground Truth: (E)**

Figure 29: A correct example of Artwork Style Inference task.

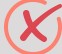 **Analogical: Artwork Style Inference**

**Question:** You are given a set of paintings from a particular artist. Based on the following choices, select the artwork that best matches this artist's style.
**Options:**

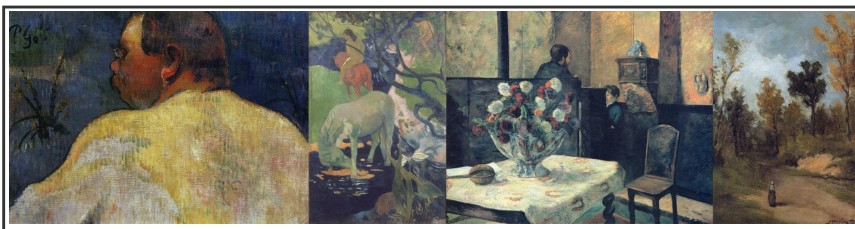

<input images>

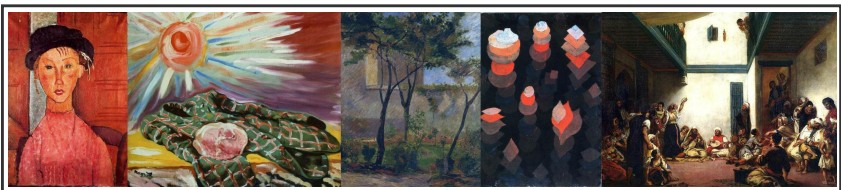

<options images (A-E) >

**GPT-5:** The set of reference images share characteristics of Paul Gauguin: post—Impressionist color, simplified forms with dark contours, Breton/Tahitian themes (e.g., riders and horses), and an early-period interior/landscape. - Among the options:  - A shows the elongated, stylized portrait style of Modigliani.  - B is a Surrealist still life linked to Magritte.  - C is geometric abstraction, unlike Gauguin.  - D is an Orientalist scene in the manner of Delacroix.  - E depicts a garden with impressionistic brushwork and palette, consistent with Gauguin's early landscapes. Therefore, the artwork that best matches the artist's style is E.  # Answer: E

**Error Category: Abstraction Error (Incorrect Abstraction and Analogy)**
**Error Reason:** GPT-5's misidentification of "E" as Gauguin likely stems from a style match based on superficial similarity: it's drawn to Impressionist-style natural landscapes and localized tones and textures, while overlooking Gauguin's key stylistic elements: Synthetism/Cloisonnism, which emphasizes the flattening of large blocks of color, heavy outlines, and a weakening of perspective and gradients.

**Ground Truth: (C)**
**Explanation:** Option C demonstrates Gauguin's core grammar: replacing volumetric modeling with flattened large blocks of color, separating color fields with clear (often dark) outlines, simplifying/decorating forms, and weakening perspective and light and shadow gradients—these are exactly the stylistic points he established in Synthetism/Cloisonnism.

Figure 30: An error example of Artwork Style Inference task.

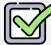 **Causal: Character Interaction Prediction**

**Question:** The yellow cat was attacked by the watermelon. Based on the given images, what is the most likely consequence of this event?
**Options:**
(A) The cat is likely to be involved in a scuffle with other cats.
(B) The result after the cat being attacked by the watermelon would be that the cat is very full and feeling a bit sluggish or satisfied.
(C) The orange cat is now tired, upset, or in a state of defeat.
(D) The result is that the cat is left alone and distressed or tired due to the commotion.
(E) The appearance of the yellow cat changed.

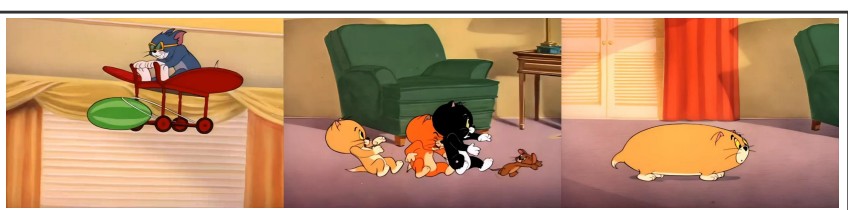

<input images>

**GPT-5:** tom is flying a toy plane carrying a green watermelon-like object.- the prompt says the yellow cat was attacked by the watermelon.- after the attack, the third image shows the yellow cat looking swollen and deformed compared to its normal shape.- therefore, the most direct consequence is a noticeable change in the yellow cat's appearance due to the impact.# answer: E

**Ground Truth: (E) The appearance of the yellow cat changed.**

Figure 31: A correct example of Character Interaction Prediction task.

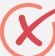 **Causal: Character Interaction Prediction**

**Question:** The little mouse has a higher emotional intelligence, but the big mouse knows how to survive. Based on the given images, what is the most likely consequence of this event?
**Options:**
(A) The big mouse ( Tom ) will maintain control and benefit from the situation.
(B) The cat and mouse form a truce or temporary friendship, with the mouse's emotional intelligence and the cat's survival instincts creating mutual respect.
(C) The little mouse tricks the cat into wearing the bell, so the mouse stays safe while the cat is foolishly burdened.
(D) The little mouse will likely outsmart and continue to manipulate the big mouse due to its higher Emotional Intelligence.
(E) Tom had completely different attitudes towards the two mice.

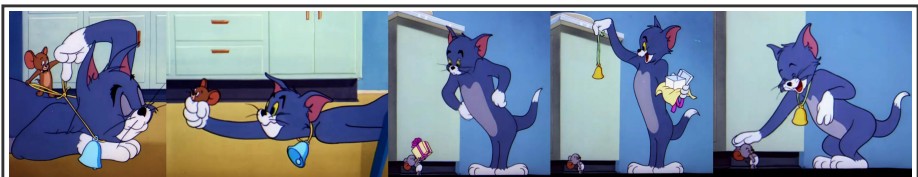

<input images>

**GPT-5:** In the first images, Tom is hostile toward Jerry and tries to catch him. - Then a smaller mouse (Nibbles) offers Tom a gift. Inside is a bell with a ribbon. - Using charm and innocence (emotional intelligence), the little mouse convinces Tom to put the bell on himself. - With the bell on Tom's neck, his movements will be noisy and predictable, keeping the mouse safe while Tom is handicapped. Therefore, the likely consequence is that the little mouse tricks the cat into wearing a bell so he stays safe.
# Answer: C

**Error Category: Hallucination Error ( Fabrication without Basis )**
**Error Reason:** GPT-5's answer fully constructs a causal chain of the "bell plot": the mouse coaxes → Tom wears the bell → the mouse is safe.

**Ground Truth: (E) Tom had completely different attitudes towards the two mice.**
**Explanation:** Because Jerry used the traditional confrontation method to get along with Tom, while the little mouse used gifts to show his friendship to Tom, Tom's attitude towards them was completely different.

Figure 32: An error example of Character Interaction Prediction task.

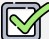 **Causal: Multi-Hop Collision Prediction**

**Question:** After the blue cylinder came from the right, what will be the result?
**Options:**
(A) The green cube collides with the yellow cube.
(B) The orange cylinder moved west.
(C) The orange cylinder collides with the green cube.
(D) The yellow cylinder moved west.
(E) The yellow cube comes from the side.

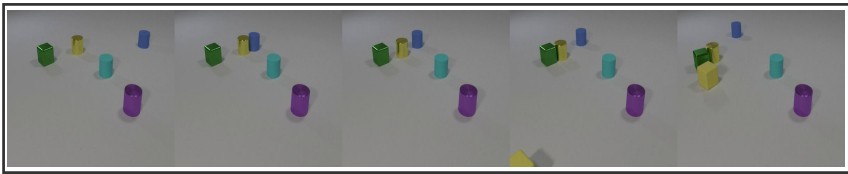

<input images>

**Gemini-2.5-Pro:** Step 1: Analyze the initial scene. The first image shows five objects on a grey surface: a green cube, a yellow cylinder, a cyan cylinder, a purple cylinder, and a blue cylinder. Step 2: Observe the sequence of events. The images show a progression of actions: In the second image, the blue cylinder, which was in the upper right, has moved closer to the yellow cylinder. In the third image, the blue cylinder collides with the yellow cylinder. In the fourth image, the impact has pushed the yellow cylinder to the left (which can be interpreted as \"west\" in this 2D projection), causing it to collide with the green cube. In the fifth image, a new object, a yellow cube, appears from the bottom left of the frame. This appears to be a separate event, not directly caused by the previous collisions. Step 3: Evaluate the question. The question asks for the \"result\" of the blue cylinder coming from the right. This implies we need to identify a consequence of the blue cylinder's movement and subsequent impact. The direct consequences are the movement of the yellow cylinder and its collision with the green cube. Step 4: Analyze the given options based on the observed events. A. The green cube collides with the yellow cube: This is incorrect. The green cube collides with the yellow cylinder, not a yellow cube. B. The orange cylinder moved west: This is incorrect. There is no orange cylinder in the scene; the cylinder in question is yellow. C. The orange cylinder collides with the green cube: This is incorrect for the same reason as B; the cylinder is yellow, not orange. D. The yellow cylinder moved west: This is correct. After being struck by the blue cylinder, the yellow cylinder is pushed to the left, which is a westward direction in this context. This is a direct result of the action described in the question. E. The yellow cube comes from the side: While a yellow cube does appear in the last image, it's not presented as a direct physical result of the blue cylinder's action. It seems to be an independent event. The question asks for the result of the collision, making the movement of the yellow cylinder a more direct and logical answer. Step 5: Conclude the best option. Based on the analysis, option D is the only statement that is both factually correct according to the images and describes a direct consequence of the action mentioned in the question. The blue cylinder's movement causes the yellow cylinder to move west.# Answer: D

**Ground Truth: (D) The yellow cylinder moved west.**

Figure 33: A correct example of Multi-Hop Collision Prediction task.

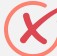 **Causal: Multi-Hop Collision Prediction**

**Question:** In the wake of the gray ball comes, what transpires?
**Options:**
(A) The cyan cylinder tagged the cyan cylinder.
(B) The green ball hit the cyan cylinder.
(C) The cyan ball hit the cyan cylinder.
(D) The green cylinder moved.
(E) The blue ball moved.

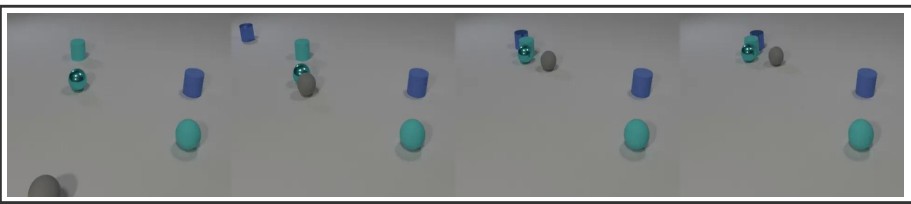

<input images>

**GPT-5:** The gray ball rolls forward and makes contact with the shiny green ball. This pushes the green ball toward the nearby cyan cylinder. In the last frames, the green ball reaches and hits the cyan cylinder, while the other objects (cyan ball, blue cylinders) do not move notably.# Answer: B

**Error Category: Perceptual Error (Color and Visual Confusion)**
**Error Reason:** GPT-5 mistook the cyan sphere for a green sphere.

**Ground Truth: (C) The cyan ball hit the cyan cylinder.**
**Explanation:** We can see that after the gray ball appears from below, it first hits the cyan metal ball, and then the cyan metal ball hits the cyan cylinder. There is no green ball in the whole scene.

Figure 34: An error example of Multi-Hop Collision Prediction task.

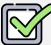 **Causal: Counterfactual Fluid Prediction**

**Question:** If the pink stick were removed, which container would pink fluid flow into?
**Options:**
(A) Orange container, Green container
(B) Green container, Red container
(C) Pink container
(D) Green container
(E) Green container, Pink container

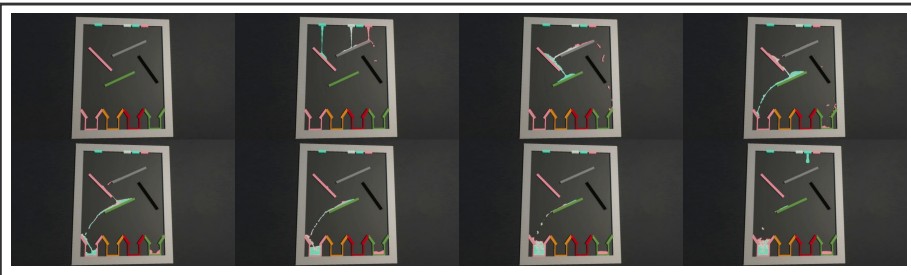

<input images>

**GPT-5:** The pink fluid drops from the right spout. Some of it always falls along the right side guided by the black bar and ends up in the right green container.- The rest runs left along the gray ramp. If the pink stick is removed, the stream from the gray ramp drops directly onto the green ramp below.- The green ramp launches this stream leftward into the left pink container.So, with the pink stick removed, pink fluid goes into both the green container and the pink container.# Answer: E

**Ground Truth: (E) Green container, Pink container**

Figure 35: A correct example of Counterfactual Fluid Prediction task.

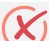 **Causal: Counterfactual Fluid Prediction**

**Question:** If the black stick were removed, which stick would green fluid pass?
**Options:**
(A) Pink stick, Blue stick
(B) Blue stick, Yellow stick
(C) Blue stick
(D) Pink stick
(E) Yellow stick

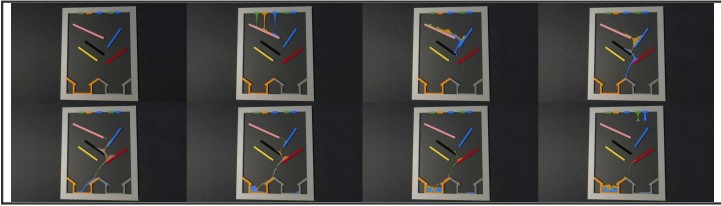
<input images>

**Gemini-2.5-Pro:** Step 1: Analyze the initial flow of the green fluid. The images show a sequence of fluids being poured. We need to trace the path of the green fluid The first green source is the leftmost one In the second image, the green fluid is released and lands on the pink stick It flows down the pink stick and, as seen in the third and fourth images, it then falls onto the blue stick From the blue stick, the fluid is directed towards and hits the black stick After the black stick, it hits the red stick and then falls into the containers below So, the original path for the green fluid is: Pink stick -> Blue stick -> Black stick -> Red stick. Step 2: Hypothesize the new flow with the black stick removed. The question asks what would happen if the black stick were removed The initial part of the path would remain the same. The green fluid would still fall from its source, land on the pink stick, and then flow onto the blue stick The change occurs after the fluid leaves the blue stick. In the original setup, it hits the black stick If the black stick is removed, the fluid will continue to fall from the end of the blue stick Looking at the positions of the other sticks, the yellow stick is located directly below the black stick, in the path of the fluid falling from the blue stick Therefore, without the black stick to intercept it, the fluid would fall from the blue stick directly onto the yellow stick Step 3: Determine the complete new path and evaluate the options. The new path for the green fluid would be: Pink stick -> Blue stick -> Yellow stick The question asks \"which stick would green fluid pass?\". We need to choose the best option from the list The fluid passes the pink, blue, and yellow sticks in this new scenario Let's look at the options: A. Pink stick, Blue stick: This is true, but incomplete. B. Blue stick, Yellow stick: This is also true, but incomplete. C. Blue stick: True, but incomplete. D. Pink stick: True, but incomplete. E. Yellow stick: This is also true Since multiple options are technically correct subsets of the full path, we must infer the question's intent. The question is about the consequence of removing the black stick. The interactions with the pink and blue sticks are unchanged. The crucial change is that the fluid, instead of hitting the black stick, now hits the yellow stick. Therefore, the most significant and defining part of the new path is the interaction with the yellow stick. This makes \"Yellow stick\" the most logical and informative answer.# Answer: E"

**Error Category: Hallucination Error（Misinformation and Confusion）**
**Error Reason:** even if the black stick is removed, the green liquid will not flow through the yellow stick, but the model's analysis is wrong, causing Gemini-2.5-Pro to justify itself and exclude the correct answer, even though the model's previous analysis was correct.

**Ground Truth: (A) Pink stick, Blue stick**
**Explanation:** We can see that after the gray ball appears from below, it first hits the cyan metal ball, and then the cyan metal ball hits the cyan cylinder. There is no green ball in the whole scene.

Figure 36: An error example of Counterfactual Fluid Prediction task.

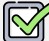 **Deductive: Material Composition Deduction**

**Question:** You are tasked with building toy robots. Each robot requires 2 spheres, 2 cubes, and 1 cylinder. You can source spheres from both Image 1 and Image 2, cubes only from Image 2, and cylinders only from Image 1. What is the maximum number of complete robots you can build?
**Options:**
(A) 1
(B) 2
(C) 3
(D) 4
(E) None of the above

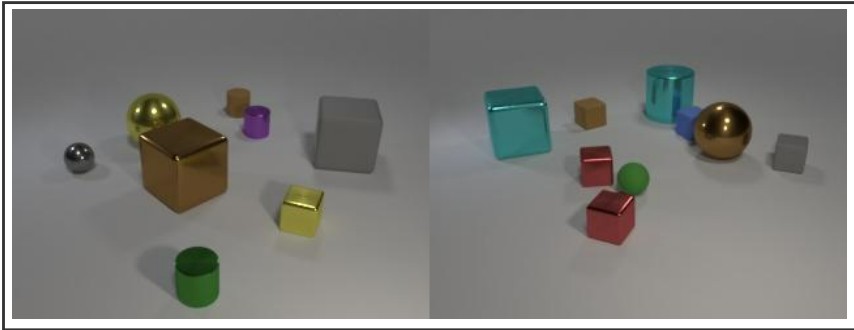

<input images>

**Gemini-2.5-Pro:** Here is a step-by-step breakdown of the reasoning:1. Identify the requirements for one robot: 2 spheres 2 cubes 1 cylinder2. Count the available parts according to the sourcing rules: Spheres (from Image 1 and Image 2): Image 1 has 2 spheres (one large gold, one small silver). Image 2 has 2 spheres (one large gold, one small green). Total available spheres = 2 + 2 = 4. Cubes (only from Image 2): Image 2 has 6 cubes (one large cyan, two small red, one small brown, one small blue, one small grey). Total available cubes = 6. Cylinders (only from Image 1): Image 1 has 3 cylinders (one green, one purple, one brown). Total available cylinders = 3.3. Calculate the maximum number of robots that can be built based on each part type (this identifies the limiting factor): Based on Spheres: You have 4 spheres, and each robot needs 2. So, 4 / 2 = 2 robots. Based on Cubes: You have 6 cubes, and each robot needs 2. So, 6 / 2 = 3 robots. Based on Cylinders: You have 3 cylinders, and each robot needs 1. So, 3 / 1 = 3 robots.4. Determine the overall maximum: The number of complete robots you can build is limited by the part that runs out first. In this case, you only have enough spheres to build 2 robots. Even though you have enough cubes and cylinders for 3 robots, you will run out of spheres after completing the second one.Therefore, the maximum number of complete robots you can build is 2.# Answer: B

**Ground Truth: (B) 2**

Figure 37: A correct example of Material Composition Deduction task.

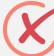 **Deductive: Material Composition Deduction**

**Question:** Each explorer-bot requires 1 sphere, 1 cube, and 2 cylinders. Spheres can be taken only from Figure 1, cubes only from Figure 2, and cylinders from either figure. After building as many complete explorer-bots as possible, how many cylinders are left unused?
**Options:**
(A) 1
(B) 2
(C) 3
(D) 4
(E) None of the above

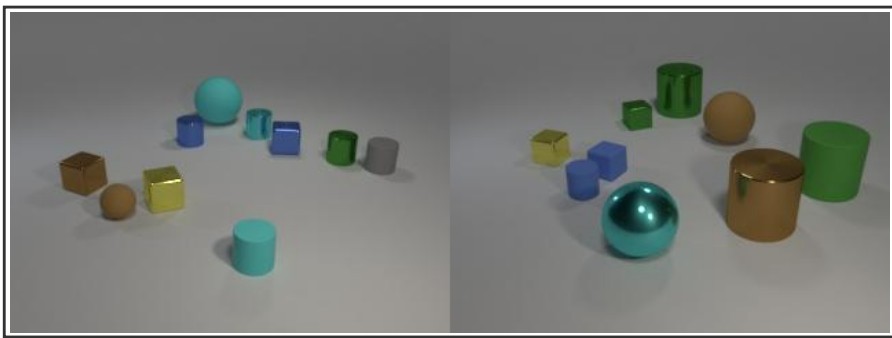

<input images>

**GPT-5:** from figure 1: spheres = 2 (teal and brown), cylinders = 5. - from figure 2: cubes = 4 (yellow, two blue, green), cylinders = 3. - total cylinders available = 5 + 3 = 8. each explorer-bot requires 1 sphere, 1 cube, and 2 cylinders. - max bots possible = min(2 spheres, 4 cubes, 8/2 cylinders) = min(2, 4, 4) = 2 bots. - cylinders used = 2 bots × 2 = 4. cylinders left unused = 8 - 4 = 4. # answer: d

**Error Category: Perceptual Error (Color and Visual Confusion)**
**Error Reason:** GPT-5 correctly identified that there were 5 cylinders in Figure 1, but it misidentified the number of cylinders in Figure 2. There were actually 4 cylinders, but the model only identified 3, resulting in an incorrect answer.

**Ground Truth: (E) None of the above**
**Explanation:** We can see that there are 5 and 4 cylinders in Figure 1 and Figure 2 respectively. Since at most 2 explorer-bots can be made, and each explorer-bot requires two 2 cylinders, there are 5 cylinders left at the end.

Figure 38: An error example of Material Composition Deduction task.

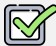 **Deductive: Card Winner Deduction**

**Question:** We have 5 players in a poker game. Each player has 2 hole cards, and there are 5 community cards on the table. Determine the winner based on the best 5-card hand formed using any combination of hole cards and community cards. Hand rankings (from highest to lowest): 1. Royal Flush: Five consecutive cards of the same suit, from 10 to Ace (10-J-Q-K-A). 2. Straight Flush: Five consecutive cards of the same suit. 3. Four of a Kind: Four cards of the same rank. 4. Full House: Three cards of one rank and two cards of another rank. 5. Flush: Five cards of the same suit that are not in sequential order. 6. Straight: Five consecutive cards of different suits. 7. Three of a Kind: Three cards of the same rank. 8. Two Pair: Two cards of one rank and two cards of another rank. 9. One Pair: Two cards of the same rank. 10. High Card: A hand that doesn't fit into any of the above categories. The input images are in the following order: the community cards, Player 1's hole cards, Player 2's hole cards, Player 3's hole cards, Player 4's hole cards, and Player 5's hole cards. Please evaluate each player's hand strength and determine the winner.

**Options:**

(A) Player 1
(B) Player 2
(C) Player 3
(D) Player 4
(E) Player 5

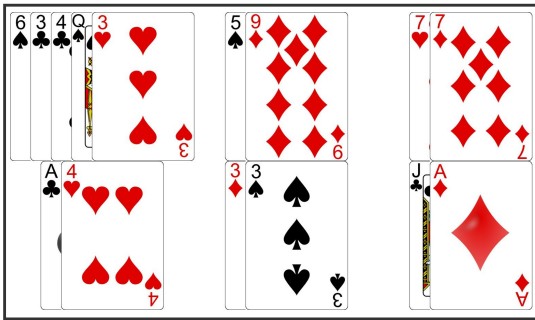

<input images>

**GPT-5:** community cards: q♠ , 6♠ , 4♣ , 3♣ , 3♥ .- player 1 (9♦  5♠ ): best hand is one pair — 3s with q, 9, 6 kickers.- player 2 (7♥  7♦ ): two pair — 7s and 3s with q kicker.- player 3 (a♣  4♥ ): two pair — 4s and 3s with a kicker.- player 4 (3♦  3♠ ): four of a kind — four 3s with q kicker.- player 5 (a♦  j♣ ): one pair — 3s with a, q, j kickers.the strongest hand is player 4's four of a kind (3s).  # answer: D

**Ground Truth: (D) Player 4**

Figure 39: A correct example of Card Winner Deduction task.

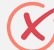 **Deductive: Card Winner Deduction**

**Question:** We have 5 players in a poker game. Each player has 2 hole cards, and there are 5 community cards on the table. Determine the winner based on the best 5-card hand formed using any combination of hole cards and community cards.Hand rankings (from highest to lowest):1. Royal Flush: Five consecutive cards of the same suit, from 10 to Ace (10-J-Q-K-A).2. Straight Flush: Five consecutive cards of the same suit.3. Four of a Kind: Four cards of the same rank.4. Full House: Three cards of one rank and two cards of another rank.5. Flush: Five cards of the same suit that are not in sequential order.6. Straight: Five consecutive cards of different suits.7. Three of a Kind: Three cards of the same rank.8. Two Pair: Two cards of one rank and two cards of another rank.9. One Pair: Two cards of the same rank.10. High Card: A hand that doesn't fit into any of the above categories.The input images are in the following order: the community cards, Player 1's hole cards, Player 2's hole cards, Player 3's hole cards, Player 4's hole cards, and Player 5's hole cards.Please evaluate each player's hand strength and determine the winner.

**Options:**
(A) Player 1
(B) Player 2
(C) Player 3
(D) Player 4
(E) Player 5

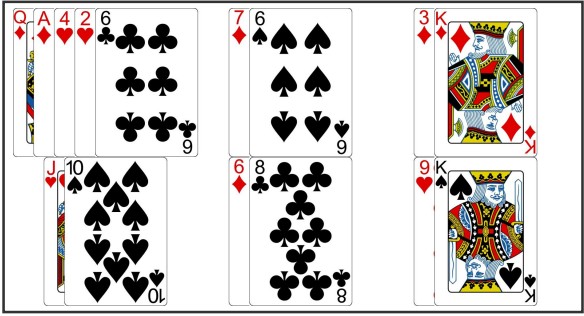

<input images>

**GPT-5:** "Let's read the cards:  Community cards: A♦ , Q♦ , 4♥ , 2♥ , 9♣ .  - Player 1: 7♦  9♠  → Makes pair of 9s using 9♣  + 9♠  with kickers A♦ , Q♦ , 7♦ . - Player 2: K♦  3♦  → No pair; best is high card A (A♦ , K♦ , Q♦ , 9♣ , 4♥ ). - Player 3: J♥  10♠  → No pair; best is high card A (A♦ , Q♦ , 10♠ , 9♣ , 4♥ ). - Player 4: 6♦  8♣  → No pair; best is high card A (A♦ , Q♦ , 9♣ , 8♣ , 6♦ ). - Player 5: 9♥  K♠  → Makes pair of 9s using 9♣  + 9♥  with higher kickers A♦ , K♠ , Q♦ .  Compare the best hands: Player 1 and Player 5 both have a pair of 9s, but Player 5's kickers (A, K, Q) beat Player 1's (A, Q, 7). Therefore, Player 5 wins.   # Answer: E"

**Error Category: Perceptual Error (Color and Visual Confusion)**
**Error Reason:** GPT-5 misidentified the suit and value of the playing cards.

**Ground Truth: (D) Player 4**
**Explanation:** Although Player 1 also has a pair of 6s, Player 4 has a stronger kicker (8) and therefore beats the other player (high card).

Figure 40: An error example of Card Winner Deduction task.

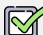 **Deductive: Recipe Step Deduction**

**Question:** I want to make Bundt Bread With Nuts And Chocolate, please generate the correct order of the images to make a complete recipe.
**Options:**
(A) 9-3-5-2-7-1-4-8
(B) 8-1-3-2-5-7-4-6
(C) 8-3-2-5-9-7-1-4
(D) 8-3-5-7-2-1-4-6
(E) 8-3-2-5-7-1-4-6

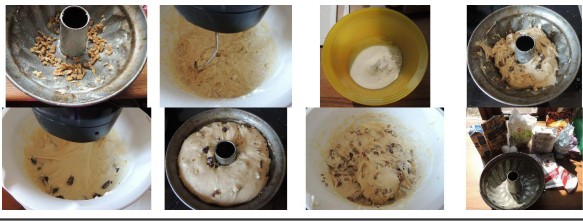
<input images>

**Gemini-2.5-Pro:** Here is a step-by-step analysis of the images to determine the correct order for making Bundt bread with nuts and chocolate:1. Image 8: Gather all ingredients. This is the classic \"mise en place\" step, where all necessary ingredients and tools are laid out before starting. We can see bags of flour, sugar, chocolate, nuts, eggs, and the Bundt pan itself. This is logically the first step. Therefore, the correct sequence must start with 8. This eliminates option A.2. Image 1: Prepare the baking pan. The Bundt pan is shown greased, floured, and with chopped walnuts sprinkled on the bottom. Preparing the pan is an early step, often done right after gathering the ingredients, to ensure it's ready when the batter/dough is mixed.3. Image 3: Measure dry ingredients. A yellow bowl contains a pile of white powder, which is likely flour and sugar, key dry ingredients for the bread. This is another preparatory step. The order of preparing the pan (1) and measuring ingredients (3) can sometimes be swapped, but both are done before mixing begins. The sequence 8-1-3 represents a very organized start.4. Image 2: Mix the initial dough. A stand mixer with a dough hook is kneading a plain, smooth dough in a white bowl. This is the base dough before any solid additions like nuts or chocolate are mixed in. This step must follow the measurement of ingredients (3).5. Image 5: Add chocolate to the dough. The same mixer and bowl are shown, but now dark chunks, clearly identifiable as chocolate, are being incorporated into the dough. This step logically follows the creation of the base dough (2).6. Image 7: The final mixed dough. The dough is shown in the mixing bowl (mixer arm removed). We can see both the dark chocolate chunks and lighter brown pieces of nuts distributed throughout the dough. This indicates that the nuts were added after the chocolate, and the mixing process is now complete. This step must follow step 5.7. Image 4: Transfer the dough to the pan. The fully mixed dough from step 7 is now being placed into the prepared Bundt pan from step 1. The dough does not yet fill the pan completely.8. Image 6: Let the dough rise. The final image shows the dough in the Bundt pan, but it has risen significantly, now filling the pan almost to the top. This is the final proofing stage before the bread goes into the oven. This step must follow the transfer of dough into the pan (4).Based on this logical progression, the correct sequence is: 8 → 1 → 3 → 2 → 5 → 7 → 4 → 6.Let's check this sequence against the given options: A. 9-3-5-2-7-1-4-8: Incorrect start and contains an invalid image number (9). B. 8-1-3-2-5-7-4-6: This matches our derived logical sequence perfectly. C. 8-3-2-5-9-7-1-4: Contains an invalid image number (9). D. 8-3-5-7-2-1-4-6: Incorrect mixing order (5 before 2). E. 8-3-2-5-7-1-4-6: This sequence suggests preparing the pan (1) after the dough is completely mixed (7). While possible, it's a less conventional and less efficient workflow than preparing the pan as part of the initial setup (as in option B). The \"mise en place\" approach shown in B is more standard.Therefore, option B provides the most logical and standard order for the baking process shown in the images.# Answer: B

**Ground Truth: (B) 8-1-3-2-5-7-4-6**

Figure 41: A correct example of Recipe Step Deduction task.

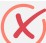 **Deductive: Recipe Step Deduction**

**Question:** I want to make Boeuf Bourguignon Modern Fusion, please generate the correct order of the images to make a complete recipe.
**Options:**
(A) 5-4-2-6-3-7-1-8
(B) 5-4-2-6-3-1-7-8
(C) 5-4-2-1-6-3-7-8
(D) 4-5-2-6-1-3-7-8
(E) 4-5-6-2-1-3-7-8

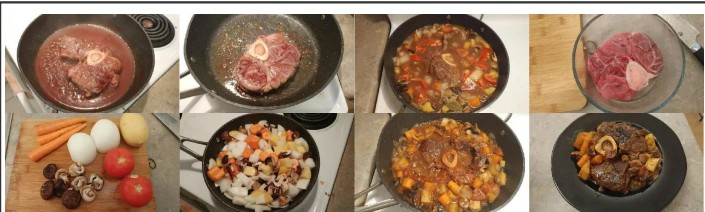

<input images>

**Gemini-2.5-Pro: :** Step 1: Ingredient Preparation. The first step in any recipe is to gather and prepare the ingredients. Image 4 shows the raw meat (a cut of beef shank, also known as osso buco) in a bowl. Image 5 shows the vegetables (carrots, onions, potato, tomatoes, mushrooms) prepped on a cutting board. The logical start is preparing both meat and vegetables. So the sequence begins with 4 and 5. Step 2: Searing the Meat. To build flavor, the meat is seared in a hot pan. Image 2 shows a piece of meat being browned in the pan. This step creates a flavorful crust on the meat and leaves browned bits (fond) in the pan. This must happen after the meat is prepped (4) and before other ingredients are cooked in the pan. So, the sequence is 4-5-2. Step 3: Sautéing the Vegetables. After the meat is seared and removed from the pan, the vegetables are typically sautéed in the remaining fat and fond. Image 6 shows the chopped carrots, onions, potatoes, and mushrooms being cooked in the same pan. This step follows the searing of the meat. The sequence is now 4-5-2-6. Step 4: Deglazing and Adding Liquid. After the vegetables are softened, a liquid (often wine in Boeuf Bourguignon) is added to deglaze the pan, lifting the flavorful fond from the bottom. The seared meat is then returned to the pan. Image 1 shows the meat back in the pan, simmering in a red liquid, which represents this stage. The sequence becomes 4-5-2-6-1. Step 5: Combining and Simmering. All the ingredients are combined to cook together. Image 3 shows the meat, all the sautéed vegetables, plus the chopped tomatoes and a bay leaf, simmering in the braising liquid. This is the beginning of the long, slow cooking process. The sequence is now 4-5-2-6-1-3. Step 6: Reducing and Finishing. The stew is simmered for an extended period, allowing the flavors to meld, the meat to become tender, and the sauce to reduce and thicken. Image 7 shows the dish after this process. The liquid level is lower than in Image 3, the colors are deeper, and the ingredients look fully cooked and tender. This is the final stage of cooking before serving. The sequence is 4-5-2-6-1-3-7. Step 7: Plating. The final step is to serve the finished dish. Image 8 shows the Boeuf Bourguignon plated and ready to eat. This is the last image in the sequence. Conclusion: The complete and logical order of the images to make the recipe is 4-5-2-6-1-3-7-8. This matches option D. # Answer: D"

**Error Category:** Other Error
**Error Reason:** Gemini-2.5-Pro sometimes overlook crucial information necessary to solve a problem. when analyzing a recipe, it fails to grasp the cues of variations in the appearance of the same side dish, such as a steak changing from rare to well-done

**Ground Truth: (C) 5-4-2-1-6-3-7-8**
**Explanation:** The reason for choosing C is that it follows the complete cooking process - ingredient preparation → frying meat → adding vegetables → stewing and reducing the sauce → serving.

Figure 42: An error example of Recipe Step Deduction task.

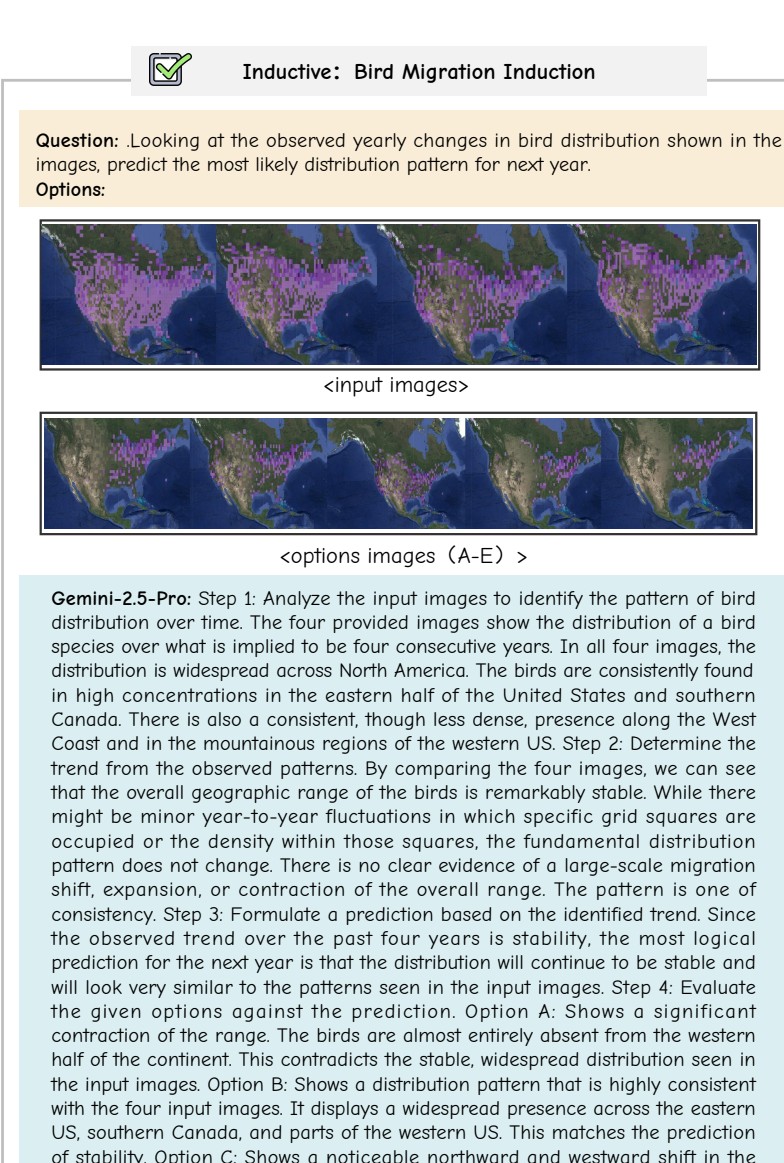

Figure 43: A correct example of Bird Migration Induction task.

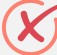 **Inductive: Bird Migration Induction**

**Question:** Looking at the observed yearly changes in bird distribution shown in the images, predict the most likely distribution pattern for next year.
**Options:**

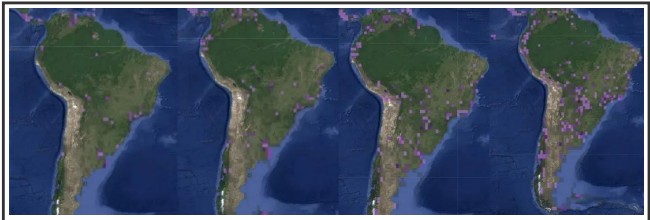

\<input images\>

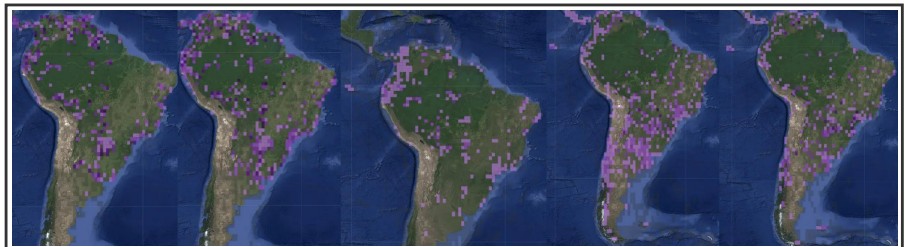

\<options images（A-E）\>

**Gemini-2.5-Pro:** First, compare the four yearly maps from earliest to latest (top-left to bottom-right). There is a clear monotonic increase in the number of purple grid cells (occupied/observed areas), with the strongest growth in the south (Argentina–Chile) and along coastlines. The fourth year shows the highest density, especially expanding into Patagonia and around the coasts, while the Amazon interior remains relatively less occupied. Extrapolating this trend to next year suggests further densification, with even more cells filled across the southern cone and coastal areas, while still keeping the Amazon comparatively lighter. Among the options, D best matches this continued expansion and densification pattern.#
Answer: D

**Error Category: Knowledge Error（Over-reliance on Internal Knowledge）**
**Error Reason:** Gemini-2.5-Pro does not perform analysis based on image information, but instead reasons based on internal geographical knowledge. And without any quantitative verification or comparison with the candidate images, D was finally selected based on the inertial cognition of "continuous expansion", while ignoring the actual inflection point changes in the image.

**Ground Truth: (E)**
**Explanation:** Because the four-year input images show that although the distribution of birds has gradually expanded, it has reached saturation by the fourth year, with only a slight increase in marginal areas such as the southern and eastern coasts; E just continues this "slight expansion" trend, rather than over-extrapolating it into a full-scale outbreak like D.

Figure 44: An error example of Bird Migration Induction task.

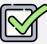 **Inductive: Plant Disease Induction**

**Question:** The leaves in the images are affected by a disease I already recognize. Which of the option leaves share the same infection?
**Options:**

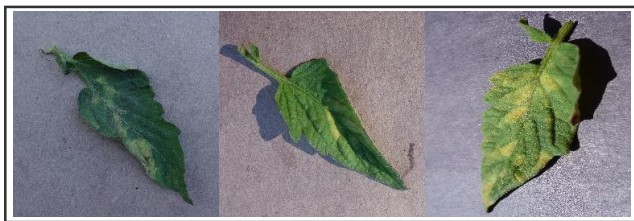

<input images>

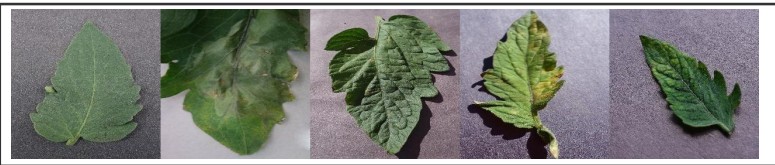

<options images (A-E) >

**Gemini-2.5-Pro:** Step 1: Analyze the initial set of images. The three images at the top show leaves, which appear to be from a tomato plant, exhibiting specific disease symptoms. The common symptoms across these three leaves are a mottled or mosaic pattern of yellow and green patches. The yellowing (chlorosis) is not uniform but appears as irregular blotches and spots across the leaf surface. In the third image, some of these yellow areas are starting to turn brown (necrosis). This pattern is characteristic of a viral disease, most likely Tomato Mosaic Virus (ToMV) or a similar virus. Step 2: Evaluate each option against the identified symptoms. Option A: This leaf is uniformly green and appears healthy. It does not show any mottling, yellowing, or spots. It also has a different shape from the tomato leaves in the initial images. This does not match. Option B: This leaf shows large, whitish-gray patches, which are typical of a fungal disease like powdery mildew. While there is some yellowing and necrosis, the primary symptom is different from the mosaic pattern in the initial images. This does not match. Option C: This leaf is dark green but has a severely puckered and bumpy texture. It does not exhibit the characteristic yellow and green mottling. This symptom is different from the one in the initial images. This does not match. Option D: This leaf is clearly a tomato leaf. It displays a distinct yellow and green mottled pattern, very similar to the leaves in the initial images. There are irregular yellow patches and some developing brown/necrotic spots. This is a strong match for the symptoms of the same disease. Option E: This leaf is mostly green with some slight discoloration and curling. The symptoms are not as pronounced or clear as in the initial images or in Option D. It does not clearly show the same mosaic pattern. Step 3: Conclude the best match. Comparing all the options, the leaf in Option D exhibits the most similar and characteristic symptoms to the leaves in the initial set of images. The yellow and green mosaic pattern is the key feature, and it is clearly present in both the initial images and Option D.# Answer: D

**Ground Truth: (D)**

Figure 45: A correct example of Plant Disease Induction task.

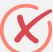 **Inductive: Plant Disease Induction**

**Question:** The images include leaves that show the same disease I've noticed earlier. From the options, which leaf is infected by the same disease?
**Options:**

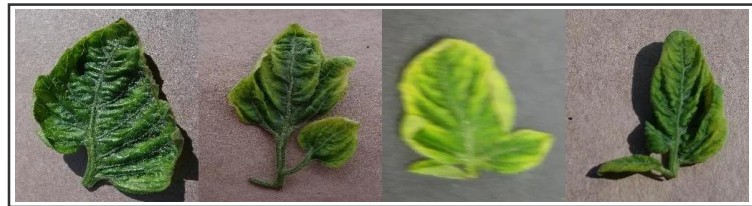

<input images>

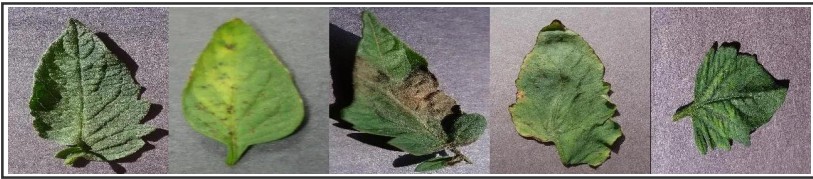

<options images（A-E）>

**Gemini-2.5-Pro:** Step 1: Analyze the symptoms in the initial set of four images. The leaves in the first four images share a very distinct primary symptom: they are severely wrinkled, crinkled, and have a bumpy or blistered texture. This is often referred to as \"savoying\". The leaves are also distorted in shape and appear curled or misshapen. Some of the leaves show discoloration, such as yellowing or a mosaic pattern of light and dark green (e.g., image 2 and the blurry image 3), but the most consistent and dramatic feature is the severe textural abnormality. Step 2: Evaluate each option against the identified symptoms. Option A: This leaf has a textured surface and serrated edges, but it lacks the severe, deep wrinkling and distortion seen in the initial images. Its overall shape is relatively normal. Option B: This leaf's primary symptoms are yellowing (chlorosis) and small, dark, necrotic spots. The leaf surface is mostly flat. This represents a spotting disease, which is different from the disease in the initial images. Option C: This leaf shows large patches of what appears to be fungal growth or blight (dark, fuzzy areas). This is a completely different symptom. Option D: This leaf is somewhat misshapen and has a slightly uneven, leathery texture, but it does not have the characteristic deep, blister-like crinkling of the initial set. Option E: This leaf perfectly matches the symptoms from the initial set. It is dark green, severely wrinkled, crinkled, and distorted. The texture is bumpy and blistered, identical to the disease pattern shown in the first four images. Step 3: Conclude based on the comparison. The disease shown in the initial images is characterized by severe leaf wrinkling and distortion. Option E is the only image that displays this same set of characteristic symptoms. # Answer: E"

**Error Category: Abstraction Error（Lack of Abstraction）**
**Error Reason:** Gemini-2.5-Pro failed to correctly generalize key features or patterns in an image and thus failed to identify key characteristics of a disease.

**Ground Truth: (B)**
**Explanation:** We can see that all four leaves in the input have yellowing symptoms, but only B has obvious yellowing.

Figure 46: An error example of Plant Disease Induction task.

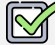 **Inductive: Sport Feature Induction**

**Question:** Given the sports displayed, determine which sport from the provided options should appear in the next position.
**Options:**

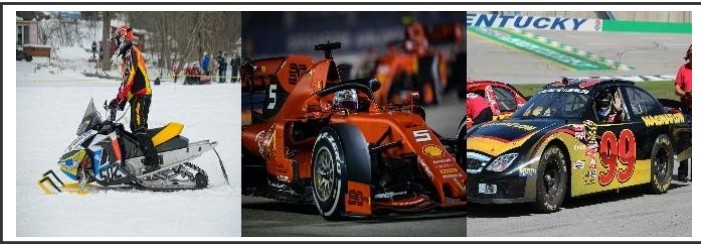

\<input images\>

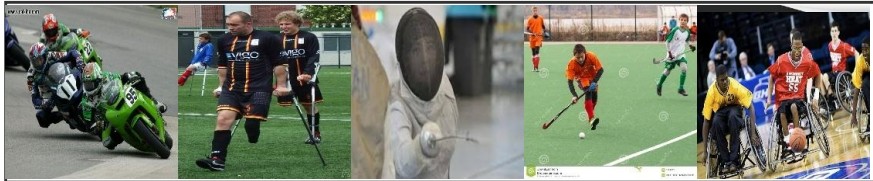

\<options images（A-E）\>

**Gemini-2.5-Pro:** Step 1: Analyze the initial set of images to identify a common theme. The first image shows a person on a snowmobile, which is a motorized vehicle used for racing on snow or ice. This is a form of motorsport. The second image shows a Formula 1 race car on a track. This is a high-speed, open-wheel motorsport. The third image shows a stock car, typical of NASCAR racing, on a racetrack. This is another form of motorsport. The unifying theme across these three images is **motorsports**, which are competitive sporting events involving motorized vehicles. Step 2: Evaluate each option to see which one fits the identified theme. **Option A:** This image shows several people racing on motorcycles. Motorcycle racing is a well-known category of motorsport. This option fits the theme. **Option B:** This image shows amputee football (soccer). This is a team sport and a parasport, but it does not involve motorized vehicles. It does not fit the theme. **Option C:** This image shows fencing. This is a combat sport and does not involve motorized vehicles. It does not fit the theme. **Option D:** This image shows field hockey. This is a team sport played with sticks and a ball, not motorized vehicles. It does not fit the theme. **Option E:** This image shows wheelchair basketball. This is a team sport and a parasport, but it does not involve motorized vehicles. It does not fit the theme. Step 3: Conclude which option is the correct choice. Based on the analysis, only Option A continues the theme of motorsports established by the first three images.# Answer: A

**Ground Truth: (A)**

Figure 47: A correct example of Sport Feature Induction task.

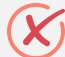 **Inductive: Sport Feature Induction**

**Question:** Considering the sports shown, which of the following should be chosen as the next figure?
**Options:**

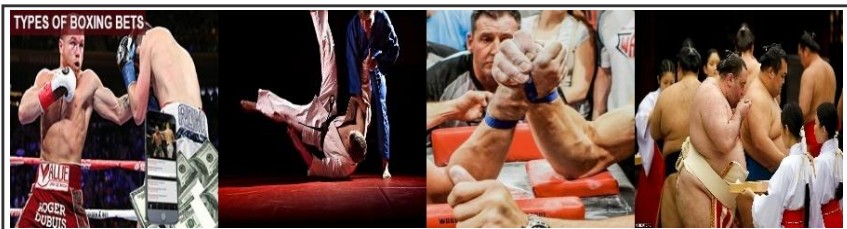

<input images>

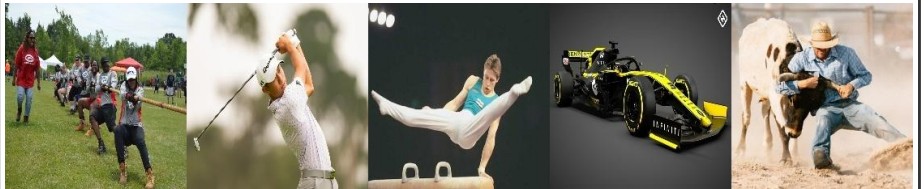

<options images (A-E) >

**GPT-5:** Look at the four given images: boxing, judo, arm wrestling, and sumo. These are all combat/strength-duel sports, typically one-on-one physical contests. Among the options, only steer wrestling (rodeo) is another direct physical combat/strength contest. The others (tug of war, golf, gymnastics, car racing) do not match this pattern.# Answer: E"

**Error Category: Abstraction Error (Incorrect Abstraction and Analogy)**
**Error Reason:** The four inputs (boxing, judo, arm wrestling, sumo) are all human-vs-human strength contests based on direct contact, leverage, and control. It generalized this to "any intense struggle" and chose steer wrestling because it looks like grappling. But steer wrestling changes a key element: it is human vs animal, usually with clear animal/rodeo cues, which are absent in the images. So the choice comes from surface resemblance ("looks like wrestling") rather than the shared structure ("human vs human").

**Ground Truth: (A)**
**Explanation:** Tug of war(A) matches the key pattern in the inputs: humans competing directly through strength and leverage. Even though it's usually team-based, it remains a purely human-vs-human physical contest, unlike sports that rely on vehicles, judged routines, or precision skills.

Figure 48: An error example of Sport Feature Induction task.

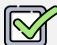 **Spatial: Relative Position Estimation**

**Question:** The washing machine is south of the microwave. In which direction is the carpet relative to the electric kettle?
**Options:**
(A) North
(B) Southeast
(C) East
(D) Southwest
(E) South

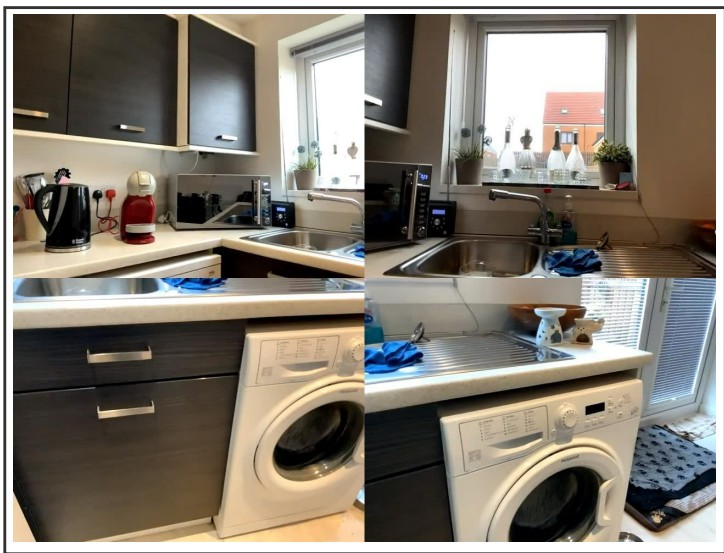

<input images>

**GPT-5:** From the statement, the washing machine is south of the microwave. In the images, the washing machine is below the microwave, so "down" corresponds to south and "right" to east.- The electric kettle is on the left side of the counter (northwest area of the scene).- The carpet is on the floor near the door, to the right of the washing machine and lower in the scene.- Relative to the kettle, the carpet is both to the right (east) and lower (south).# Answer: B

**Ground Truth: (B) Southeast**

Figure 49: A correct example of Relative Position Estimation task.

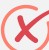 **Spatial: Relative Position Estimation**

**Question:** The chair is to the east of the Hello Kitty plush toy. Where is the Minion plush toy positioned relative to the pony plush toy?

**Options:**
(A) Southeast
(B) North
(C) Northwest
(D) Northeast
(E) West

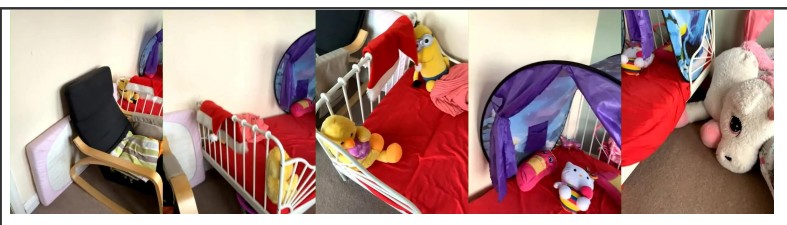

<input images>

**Gemini-2.5-Pro:** Step 1: Establish the cardinal directions. The problem states that \"The chair is to the east of the Hello Kitty plush toy.\" Image 1 shows a black chair with a wooden frame on the left side of the scene. Image 4 shows the Hello Kitty plush toy on the bed, inside a purple tent, which is to the right of the chair's initial position. The camera pans from left to right, moving from the chair towards the Hello Kitty toy. This means the chair is physically to the left of the Hello Kitty toy. Given that the chair is to the EAST of the Hello Kitty toy, we can establish our directional mapping: East = Left (in the direction of the chair from Hello Kitty) West = Right , North = Towards the back wall, South = Towards the foreground/camera Step 2: Locate the Minion plush toy and the pony plush toy.   The Minion plush toy is visible in Image 3. It is a yellow, one-eyed character sitting on the red bed, near the white headboard (which is against the back wall). The pony plush toy (a large white and pink unicorn) is visible in Image 5. It is on the floor, to the right of the bed. Step 3: Determine the relative position of the Minion to the pony.   We need to find the position of the Minion  relative to  the pony. The Minion is near the back wall, making it NORTH of the pony, which is further out on the floor. The Minion is on the bed, which is to the left of where the pony is on the floor. Since \"left\" is EAST, the Minion is to the EAST of the pony. Combining these two directions, the Minion is to the NORTH and to the EAST of the pony. Step 4: Conclude the final direction.   A position that is both North and East is Northeast. Therefore, the Minion plush toy is positioned to the Northeast of the pony plush toy.# Answer: D"

**Error Category: Knowledge Error（Directional and Positional Confusion）**
**Error Reason:** Gemini-2.5-Pro lacks common sense of orientation and reverses left to west and right to east.

**Ground Truth: (A) Southeast**
**Explanation:** Since the chair is east of Hello Kitty, the Minion plush toy are next to the chair, and pony plush toy is next to Hello Kitty, the Minion plush toy are southeast of pony plush toy.

Figure 50: An error example of Relative Position Estimation task.

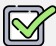 **Spatial: Camera Rotation Estimation**

**Question:** From the continuous sequence of images, what were the degrees of rotation the camera made?
**Options:**
(A) Rotation clockwise about 90 degrees.
(B) Rotate clockwise about 180 degrees.
(C) Rotate clockwise about 90 degrees.
(D) Rotate counterclockwise about 145 degrees.
(E) Rotate clockwise 120 degrees.

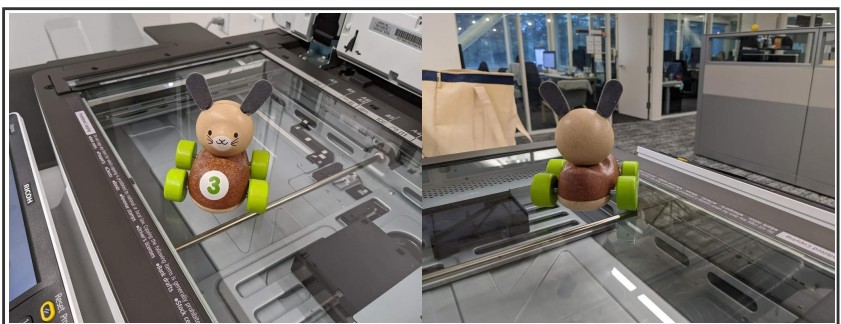

<input images>

**GPT-5:** Observe the toy bunny on the scanner bed. - In the first image, the bunny faces the camera (front with the "3" visible). - In the second image, the camera view shows the back of the bunny. - Seeing the front then the back implies the camera moved to the opposite side of the object, a rotation of about 180 degrees around it (direction doesn't matter for 180°; clockwise is equivalent). # Answer: B"

**Ground Truth: (B) Rotate clockwise about 180 degrees.**

Figure 51: A correct example of Camera Rotation Estimation task.

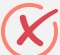 **Spatial: Camera Rotation Estimation**

**Question:** Given the continuous images, what were the successive rotation angles of the camera?

**Options:**
(A) Rotate clockwise about 30 degrees, then clockwise about 45 degrees
(B) Rotate clockwise about 30 degrees, then clockwise about 30 degrees
(C) Rotate clockwise about 30 degrees, then counterclockwise about 150 degrees
(D) Rotate counterclockwise about 30 degrees, then counterclockwise about 60 degrees
(E) Rotate clockwise about 45 degrees, then clockwise about 45 degrees

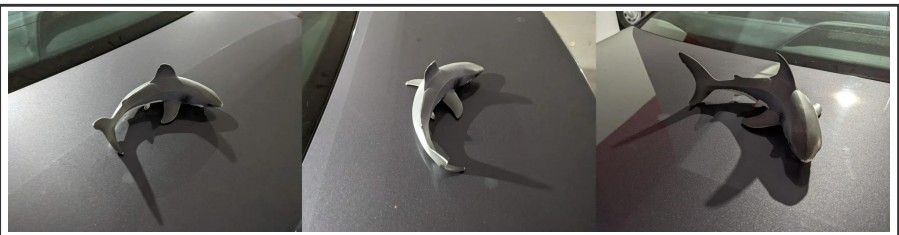

<input images>

**GPT-5:** We compare the shark toy's orientation relative to the frame and the car's edges.- From the first to the second image, the scene is rotated slightly clockwise: the shark's head points more downward-right and the window edge tilts a bit, indicating a small clockwise roll (~30°).- From the second to the third image, the rotation continues in the same (clockwise) direction and by a larger amount: the shark is now viewed closer to a frontal-right angle, roughly an additional ~45°.# Answer: A

**Error Category: Knowledge Error（Rotation and Viewpoint Confusion）**
**Error Reason:** GPT-5 misinterpreted a counterclockwise rotation of approximately 150° as a continued clockwise rotation of a small angle, ignoring the direction reversal evidence provided by the absolute reference.

**Ground Truth: (C) Rotate clockwise about 30 degrees, then counterclockwise about 150 degrees**
**Explanation:** First, from the right side of the shark to the shark's tail, you need to rotate clockwise about 30 degrees, and then to the shark's head, you need to rotate counterclockwise 150 degrees.

Figure 52: An error example of Camera Rotation Estimation task.

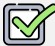 **Spatial: Navigation Route Planning**

**Question:** You are a robot beginning at the refrigerator and facing the refrigerator. You want to navigate to the recycling bin. You can only make 90-degree or 180-degree turns and go forward, and you have to go around obstacles. What actions should you perform?

**Options:**
(A) 1. Turn right 2. Go forward until the round table 3. Turn left 4. Go forward until the recycling bin
(B) 1. Turn left 2. Go forward until the round table with red chairs 3. Turn right 4. Go forward until the recycling bin
(C) 1. Turn back 2. Go forward until the round table 3. Turn left 4. Go forward until the recycling bin
(D) 1. Turn left 2. Go forward until the table 3. Turn left 4. Go forward until the recycling bin
(E) 1. Turn left 2. Go forward until the suitcase 3. Turn left 4. Go forward until the printer 5. Turn right 6. Go forward until the recycling bin

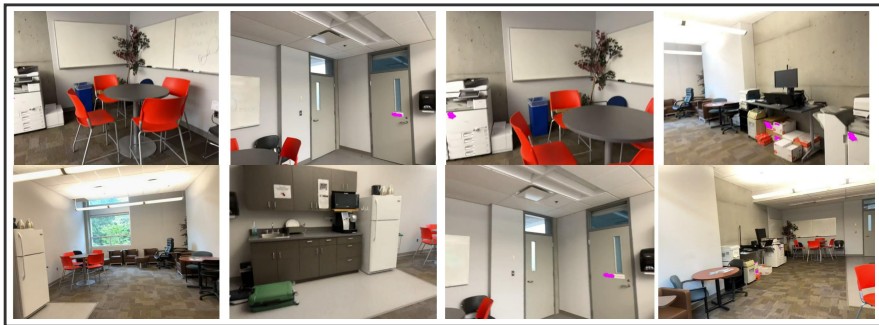

<input images>

**GPT-5:** Start at the refrigerator facing it; the recycling bin is across the room near the printers and the round table in the back corner.- First, turn left to align with the counter area and move forward along it until you reach the green suitcase on the floor.- Turn left again to head across the room toward the printers against the opposite wall.- Go forward until you reach the printer, then turn right to face the corner with the blue bin.- Move forward to the recycling bin.# Answer: E

**Ground Truth: (E) 1. Turn left 2. Go forward until the suitcase 3. Turn left 4. Go forward until the printer 5. Turn right 6. Go forward until the recycling bin**

Figure 53: A correct example of Navigation Route Planning task.

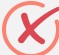 **Spatial: Navigation Route Planning**

**Question:** You are a robot beginning at the door, facing the window. You want to navigate to the refrigerator. You can only make 90-degree or 180-degree turns and go forward, and you have to go around obstacles. What actions should you perform?

**Options:**
(A)  1. Turn right 2. Go forward until the refrigerator
(B)  1. Turn left 2. Go forward until the stove 3. Turn right 4. Go forward until the refrigerator
(C)  1. Go forward until reaching the gas cooker. 2. Turn left. 3. Go forward until reaching the fridge.
(D)  1. Turn right 2. Go Forward until the counter 3. Turn right 4. Go Forward until the refrigerator
(E)  1. Go forward until the end of the counter. 2. Turn right. 3. Go forward until the refrigerator

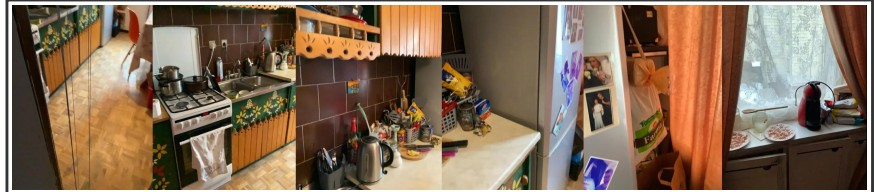

<input images>

**GPT-5:** From the doorway you're facing the window with the counter and stove along your left side.- Move straight ahead down the aisle beside the counter.- When you reach the end of the counter, the refrigerator is positioned to the right.- Turn right and go straight to the fridge.# Answer: E"

**Error Category: Knowledge Error (Directional and Positional Confusion)**
**Error Reason:** GPT-5 had a poor understanding of the kitchen's spatial perception in the image. It made the wrong choice because it ignored obstacles like the stove and counter, mistakenly assuming it could go straight and then turn right to the refrigerator, leading to an oversimplification of spatial reasoning.

**Ground Truth: (B) 1. Turn left 2. Go forward until the stove 3. Turn right 4. Go forward until the refrigerator**
**Explanation:** Since you are at the door and facing the window at the beginning, you need to turn left first instead of going directly to the kitchen counter.

Figure 54: An error example of Navigation Route Planning task.

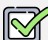 **Temporal: Crowd Timeline Reconstruction**

**Question:** In what order do these images appear chronologically?
**Options:**
(A) 3-1-4-2
(B) 1-3-2-4
(C) 4-1-2-3
(D) 3-1-2-4
(E) 4-3-1-2

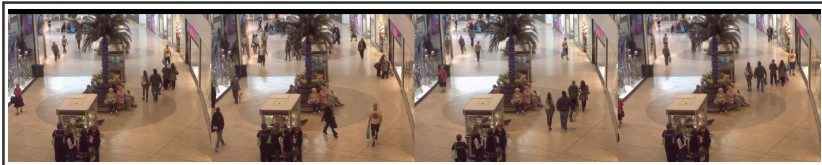

<input images>

**Gemini-2.5-Pro:** Step 1: Identify Key Moving Subjects. To determine the chronological order, we need to track the movement of people across the four images. Several individuals and groups can be used as reference points: 1. A group of four people (a man in a dark jacket, a woman in a light top, a man in a plaid shirt pushing a stroller, and a woman in a brown coat) walking away from the camera on the right side of the mall. 2. A man in a yellow t-shirt walking towards the camera on the right side. 3. A woman in a red coat walking away from the camera on the left side. 4. A woman in a dark outfit who walks from right to left in front of the people sitting around the palm tree. Step 2: Track the Group of Four with the Stroller. This group is walking away from the camera. Their position further down the hall indicates a later time. In Image 3, this group is the closest to the camera. The stroller is just beginning to pass the circular seating area. In Image 1, the group has moved slightly further away from the camera. In Image 4, the group has moved even further down the hall. In Image 2, the group is the furthest away from the camera among the four images. This tracking gives the order: 3 -> 1 -> 4 -> 2. Step 3: Track the Man in the Yellow T-shirt. This man is walking towards the camera. His position closer to the bottom of the frame indicates a later time. In Image 3, he is the furthest away, near the top right of the frame. In Image 1, he has walked closer to the camera. In Image 4, he is even closer than in Image 1. In Image 2, he is the closest to the camera, near the bottom right corner. This tracking also gives the order: 3 -> 1 -> 4 -> 2. Step 4: Track the Woman in the Red Coat. This woman is walking away from the camera on the left. Her position further down the hall indicates a later time. In Image 3, she is closest to the camera, near the bottom left. In Image 1, she has moved further up the walkway. In Image 4, she is further still. In Image 2, she is the furthest away of the four images. This tracking confirms the order: 3 -> 1 -> 4 -> 2. Step 5: Track the Woman in the Dark Outfit. This woman walks from right to left in front of the seating area. In Image 3, she is not yet visible in that area. In Image 1, she appears on the right side of the seating area. In Image 4, she is in the middle, directly in front of the seated people. In Image 2, she has passed the seated people and is on the left side. This tracking also supports the order: 3 -> 1 -> 4 -> 2. Step 6: Conclude the Order. All key subjects' movements consistently point to the same chronological sequence. The images, in order, show the progression of these people through the mall. The correct sequence is 3, then 1, then 4, and finally 2. Step 7: Match with the Options. The determined order is 3-1-4-2, which corresponds to option A.# Answer: A"

**Ground Truth: (A) 3-1-4-2**

Figure 55: A correct example of Crowd Timeline Reconstruction task.

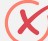 **Temporal: Crowd Timeline Reconstruction**

**Question:** What is the correct chronological sequence of these images?
**Options:**
(A) 5-3-2-4-1
(B) 5-3-1-4-2
(C) 2-4-1-3-5
(D) 1-2-3-4-5
(E) 4-1-2-3-5

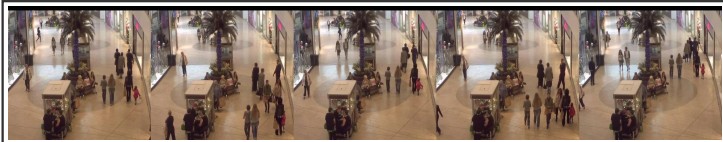

<input images>

**Gemini-2.5-Pro:** Here is a step-by-step analysis to determine the chronological order of the images:1. Identify Key Moving Elements: The images depict a busy shopping mall with many people moving around. To establish a timeline, we can track the positions of specific individuals or groups across the different frames. Let's focus on a few distinct groups Group A (Right Side): A group of four people, including a woman holding the hand of a small child in a red coat. They are walking away from the camera, down the right-hand corridor Group B (Center): A couple, a man in a dark top and a woman in a light top, walking towards the camera in the upper-middle part of the scene Person C (Left Side): A woman in a light-colored jacket and dark pants, walking from the upper left towards the bottom left of the frame. 2. Track the Movement of Group A (Child in Red): Image 5: This group is the furthest away from the camera, just entering the circular area around the central planter Image 3: They have moved forward. The woman holding the child's hand is now roughly level with the right side of the planter Image 1: They have moved further still. They are now past the planter and closer to the bottom right of the frame. Images 4 and 2: This group is no longer visible, indicating they have walked out of the frame. This tracking establishes the sequence: 5 -> 3 -> 1. Images 4 and 2 must come after these. 3. Track the Movement of Person C (Woman in Light Jacket): Image 5: This woman is not yet visible Image 3: She first appears in the upper left, beginning her walk Image 1: She has moved further along her path, towards the bottom left Image 4: She is even further along, closer to the bottom left corner Image 2: She is at the very edge of the frame, about to exit. This tracking confirms the sequence: (5) -> 3 -> 1 -> 4 -> 2 . 4. Track the Movement of Group B (Central Couple): Image 5: This couple is the furthest away, in the upper middle of the scene Image 3: They have walked closer to the camera Image 1: They are even closer Image 4: They have walked past the central planter and are now moving towards the kiosk on the left Image 2: They are even closer to the kiosk. This tracking also confirms the sequence: 5 -> 3 -> 1 -> 4 -> 2 . 5. Conclusion: By consistently tracking the movement of multiple people and groups, we can confidently determine the chronological order. The people in the scene are furthest away in Image 5 and progressively move closer or across the frame through Images 3, 1, 4, and finally 2. Therefore, the correct chronological sequence is 5-3-1-4-2. # Answer: B

**Error Category: Reasoning Error（Temporal and Sequential Confusion）**
**Error Reason:** Gemini-2.5-Pro can use a certain reference point to sort the events, but there are too many reference points. In this example, the model sorting result is completely opposite to the correct answer.

**Ground Truth: (C) 2-4-1-3-5**
**Explanation:** In these images, we can use the little girl in red and her surrounding behavior as a reference point. She is holding her parent's hand and can be seen moving further and further away from the camera.

Figure 56: An error example of Crowd Timeline Reconstruction task.

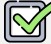 **Temporal: Driving Sequence Prediction**

**Question:** Which of the images in the options is most likely to be the next one to occur?
**Options:**

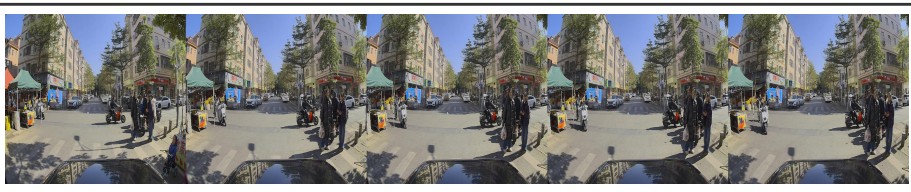

\<input images\>

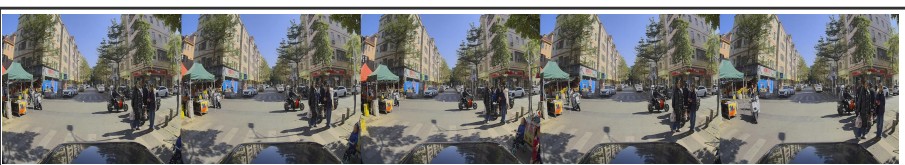

\<options images（A-E）\>

**GPT-5:** Observe the sequence: the dashcam car inches forward while the scooter on the left moves out from under the tent toward the street (rightwards each frame). - The pedestrians on the right shift slightly toward the sidewalk, and the small child at the lower-right remains visible. - The next frame should therefore show the left scooter further into the road and the right-side pedestrians a bit farther right.Option E best matches this continued motion: the left scooter advances further right and the pedestrians are slightly more to the side, consistent with the progression.# Answer: E

**Ground Truth: (E)**

Figure 57: A correct example of Driving Sequence Prediction task.

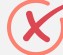 **Temporal: Driving Sequence Prediction**

**Question:** Which image in the options is most likely to appear in the next moment?
**Options:**

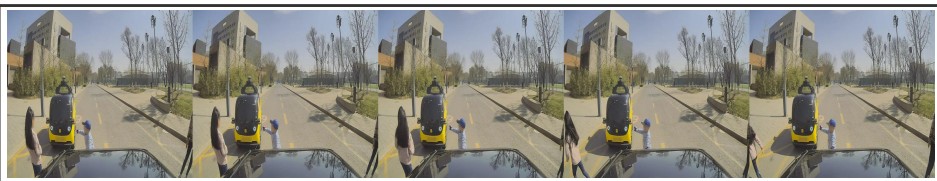
<input images>

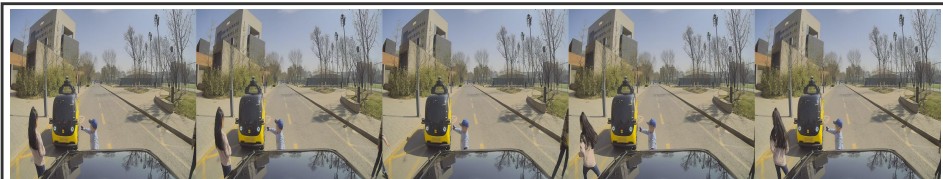
<options images（A-E）>

**GPT-5:** We have five consecutive frames showing a girl on the left starting to move away to the left, while the boy and the robot remain nearly stationary.- From frame to frame, the girl's position shifts progressively leftward; by the last provided frame she has begun running and is nearing the left edge.- The most likely next moment should show her even farther left (still partly in frame), with the rest of the scene largely unchanged.Among the options:- A and B show her still standing near the robot (earlier state).- C shows her already almost out of frame (likely a later moment).- E shows minor movement only.- D shows her continuing to run left, slightly farther than in the last frame, matching the immediate next step.# Answer: D

**Error Category: Reasoning Error（Temporal and Sequential Confusion）**
**Error Reason:** GPT-5 did not correctly analyze the little girl's movement trend. She was almost out of the frame.

**Ground Truth: (C)**
**Explanation:** We can see from the last image input that the little girl is almost running out of the picture, so only option C fits the sequence of events.

Figure 58: An error example of Driving Sequence Prediction task.

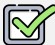 **Temporal: Human Activity Localization**

**Question:** Can you identify when the action 'a person walks through the doorway' happens in the sequence?
**Options:**
(A) Throughout the entire sequence.
(B) At the end of the sequence.
(C) In the middle of the sequence.
(D) At the beginning of the sequence.
(E) Not shown in the sequence.

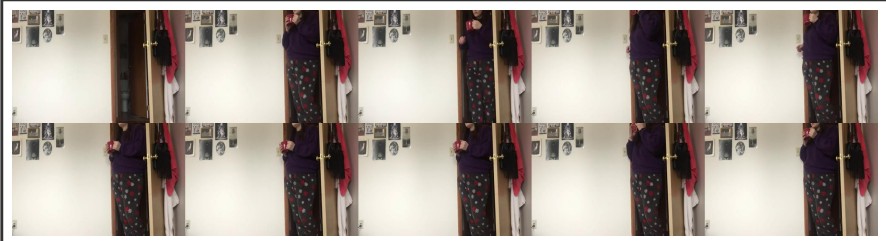

<input images>

**GPT-5:** First frame: door is open with no person present.- Second frame: a person appears, stepping out of the doorway with a mug.- Subsequent frames: the person stands near the doorway and drinks, not walking further.Therefore, the walking through the doorway occurs right at the beginning of the sequence. # Answer: D."

**Ground Truth: (D) At the beginning of the sequence.**

Figure 59: A correct example of Human Activity Localization task.

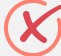 **Temporal: Human Activity Localization**

**Question:** During which part of the sequence does the action 'a person puts a coffee cup on a shelf' occur?
**Options:**
(A)  Throughout the entire sequence.
(B)  At the end of the sequence.
(C)  In the middle of the sequence.
(D)  At the beginning of the sequence.
(E)  Not shown in the sequence.

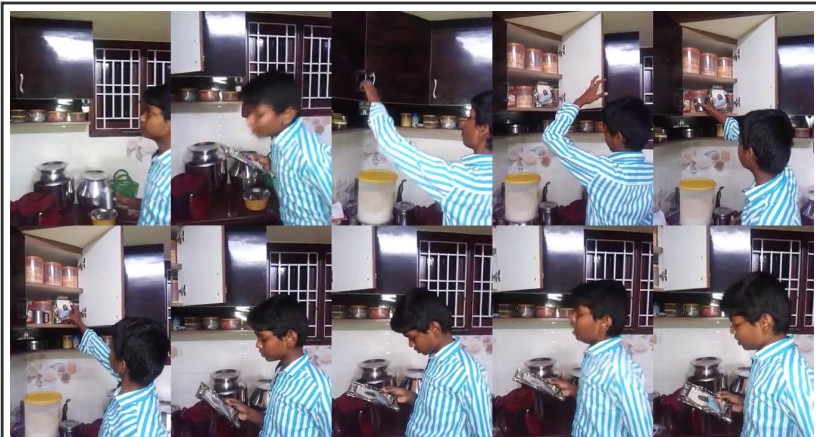
<input images>

**GPT-5:** Look at the sequence: the person opens a kitchen cabinet and reaches toward items inside.- The shelves contain jars and possibly packets; the person then holds a packet or tray-like item.- At no point is a coffee cup being placed onto a shelf; the actions are opening, reaching, and holding items, likely taking them out.# Answer: E.

**Error Category: Perceptual Error (Color and Visual Confusion)**
**Error Reason:** GPT-5 fails to capture key visual information, such as the coffee cup in the first few frames.

**Ground Truth: (D) At the beginning of the sequence.**
**Explanation:** We can see the first 5 frames, where the little boy puts the coffee cup on a shelf.

Figure 60: An error example of Human Activity Localization task.

