# OpenReview forum: "MMR-Life: Piecing Together Real-life Scenes for Multimodal Multi-image Reasoning"
_ICLR.cc/2026/Conference — ICLR 2026 Poster_

### Official Review · Reviewer_YUHP · 2025-10-31

**Soundness:** 2
**Presentation:** 4
**Contribution:** 3
**Rating:** 6
**Confidence:** 3

**Summary:**

This paper introduces MMR-Life, a multimodal multi-image reasoning benchmark focusing on real-life scenarios. The benchmark contains  2,676 multiple-choice questions covering seven reasoning categories. Evaluation of state-of-the-art models shows deficiency in performance, and current reasoning enhancement strategies do not work too well on the benchmark.

**Strengths:**

- The paper is well-written and easy to follow, with an abundance of examples.
- The benchmark contains interesting challenges and performs a pretty thorough evaluation of sota models, showing their deficiency.
- The exploration of reasoning enhancement strategies provides some interesting insights.

**Weaknesses:**

- I appreciate the rich appendix section and accompanying analysis. But these examples are most likely cherry-picked, which is totally fine for presentation. It would be good to include a link to an anonymously hosted repository containing the full dataset so reviewers can get a more unbiased, holistic view of data quality.
- Related to above: Some of the problems in the appendix are of low quality.
    - For example, on page 65, the ground truth explanation states that the answer should be "water sport/racing". Not all images in the question are water sports (e.g., there is horse racing), so it is perfectly fine to select E as the correct answer as well.
    - On page 45, the question states "My friend already owns this pair of shoes", but four pairs of shoes are shown. This is rather confusing. Also, I don't think this question makes any sense. The ground truth, C, looks similar to the first pair in terms of color and the second in terms of style.
    - On page 77, I do not even understand the question. It seems to contain two completely separate questions?
    - On page 37, even after zooming in, I do not see where there is milk on the table.

The point I want to make is that the selected questions probably represent some of the higher-quality ones, yet some of them are still not totally reasonable for models to answer, so I think giving reviewers access to the full dataset is essential for quality control.
- The paper mentions a 600-question validation set. But I think it only appears once in the paper in the context of human performance. Model performance is reported on the whole set, but human performance is only on the validation set. They may not be directly comparable. What is the process for constructing this validation set? What is model performance on this subset?
- The paper claims the questions cover real-life scenes, but many are in fact pretty artificial or unlikely to come across in real-life (e.g., pages 40, 41 -- synthetic images; pg 60, 61 -- scientific and domain-grounded; pg 48, 49 -- cartoon scenes). These still carry relevance and assess multimodal reasoning, but weakens the paper's focus on real-life scenes.

**Questions:**

The paper says that "We generate question-answer pairs using either automatic synthesis or manual annotation, depending on the task type". What is the percentage of machine- and human-generated problems? I see there is a task description section in the appendix, but I think it is also important to clearly outline sources of questions (i.e., human/LLM-generated) for each task.

---

> ### Author Response · Authors · 2025-11-14
> **Responses to Reviewer YUHP (part 1)**
>
> Thanks for your careful and insightful reviews.
>
> **Weakness 1**
>
> > It would be good to include a link to an anonymously hosted repository containing the full dataset so reviewers can get a more unbiased, holistic view of data quality.
>
> **Reply 1**
>
> Thank you for your valuable suggestions. Here, we have anonymously uploaded the entire dataset at the following link: https://drive.google.com/drive/folders/130sbsE4MZ8CtDBZxMZFKU3GaYtYJ_bh7?usp=sharing.
>
>
>
>
> **Weakness 2**
>
> > Some of the problems in the appendix are of low quality.
>
> **Reply 2**
>
> + For example on page 65,  although option E seems reasonable, most of the images in the input are related to water sports, so option B is a more reasonable answer. We apologize for the choice of explanation here, which was not optimal, and we will improve it in the subsequent version of the paper.
> + For example on page 45, we apologize for the typographical error, it should have been **"these pairs"** instead of "this pair." The task requires the model to select a style from the options that is different from all four input images, not just considering color. We can observe that the style in option C is different from that in the second image, as the former has a concave middle section, whereas the latter does not.
> + For example on page 77,  we apologize again for the incorrect phrasing here. The correct question should be: "During which part of the sequence does the action 'a person puts a coffee cup on a shelf' occur?" We will correct this in the subsequent version of the paper.
> + For example on page 37, in the first four images, there is a cup containing white milk placed on a black table in the center of the scene. Since the cup is transparent, this could indeed pose a significant challenge for the model's perception. However, this is also a realistic scenario that one might encounter in real life.

---

> > ### Author Response · Authors · 2025-11-14
> > **Responses to Reviewer YUHP (part 2)**
> >
> > **Weakness 3**
> >
> > > What is the process for constructing this validation set?  What is model performance on this subset?
> >
> > **Reply 3**
> >
> > + For the constructing process, we apologize for the confusion caused by the lack of this detail. Specifically, we extracted 10 questions from each task, forming a mini-test set of 210 questions. From this set, we randomly selected 50 questions at a time to be assigned to one of the 12 annotators, collecting a total of 600 valid annotations. This approach ensures that each specific question is annotated by multiple annotators, reducing randomness while controlling costs.
> >
> > + For the model performance on the subset, we have included additional experiments, and the results are shown in the table below:
> >
> >   | Model               |    Abd    |    Ana    |    Cau    |    Ded    |    Ind    |    Spa    |    Tem    |    Avg    |
> >   | :------------------ | :-------: | :-------: | :-------: | :-------: | :-------: | :-------: | :-------: | :-------: |
> >   | Human               | **79.76** |   57.65   | **75.00** |   70.59   |   63.41   | **79.76** | **79.76** | **72.28** |
> >   | gpt-5               |   63.33   |   66.67   |   53.33   |   73.33   | **66.67** |   30.00   |   50.00   |   57.62   |
> >   | gemini-2.5-pro      |   63.33   | **80.00** |   60.00   |   76.67   |   50.00   |   30.00   |   36.67   |   56.67   |
> >   | gemini-2.5-flash    |   53.33   |   63.33   |   50.00   |   53.33   |   50.00   |   50.00   |   53.33   |   53.33   |
> >   | o4-mini             |   46.67   |   73.33   |   43.33   |   73.33   |   50.00   |   23.33   |   43.33   |   50.48   |
> >   | gpt-5-mini          |   50.00   |   56.67   |   40.00   | **83.33** |   50.00   |   6.67    |   33.33   |   45.71   |
> >   | claude-sonnet-4     |   40.00   |   66.67   |   60.00   |   66.67   |   30.00   |   16.67   |   56.67   |   48.10   |
> >   | gpt-4.1             |   50.00   |   56.67   |   36.67   |   16.67   |   63.33   |   6.67    |   26.67   |   36.67   |
> >   | claude-3.7-sonnet   |   30.00   |   73.33   |   46.67   |   70.00   |   40.00   |   13.33   |   43.33   |   45.24   |
> >   | gpt-4o              |   43.33   |   50.00   |   40.00   |   16.67   |   46.67   |   20.00   |   26.67   |   34.76   |
> >   | gpt-4.1-mini        |   33.33   |   63.33   |   36.67   |   53.33   |   50.00   |   10.00   |   26.67   |   39.05   |
> >   | doubao-1.5-vision   |   50.00   |   50.00   |   36.67   |   63.33   |   40.00   |   13.33   |   23.33   |   39.52   |
> >   | VL-Rethinker-72B    |   26.67   |   50.00   |   40.00   |   63.33   |   33.33   |   13.33   |   23.33   |   35.71   |
> >   | QVQ-72B-Preview     |   26.67   |   53.33   |   50.0    |   36.67   |   26.67   |   3.33    |   16.67   |   30.48   |
> >   | MM-Eureka-Qwen-32B  |   20.00   |   46.67   |   43.33   |   50.00   |   33.33   |   16.67   |   36.67   |   35.24   |
> >   | MiMo-VL-7B-RL       |   36.67   |   16.67   |   40.00   |   70.00   |   16.67   |   13.33   |   26.67   |   31.43   |
> >   | VL-Rethinker-7B     |   23.33   |   36.67   |   13.33   |   13.33   |   33.33   |   0.00    |   16.67   |   22.78   |
> >   | Keye-VL-1.5-8B      |   13.33   |   23.33   |   33.33   |   20.00   |   26.67   |   13.33   |   26.67   |   22.38   |
> >   | Skywork-R1V-38B     |   30.00   |   6.67    |   33.33   |   16.67   |   30.00   |   16.67   |   13.33   |   20.95   |
> >   | Qwen2.5-VL-72B      |   36.67   |   40.00   |   40.00   |   20.00   |   36.67   |   10.00   |   30.00   |   30.48   |
> >   | Gemma3-27B          |   20.00   |   36.67   |   33.33   |   30.00   |   40.00   |   6.67    |   30.00   |   28.10   |
> >   | Gemma3-12B          |   30.00   |   50.00   |   16.67   |   33.33   |   20.00   |   6.67    |   23.33   |   25.71   |
> >   | Qwen2.5-VL-32B      |   23.33   |   30.00   |   30.00   |   20.00   |   20.00   |   6.67    |   23.33   |   21.90   |
> >   | Qwen2.5-VL-7B       |   13.33   |   0.00    |   23.33   |   20.00   |   0.00    |   6.67    |   10.00   |   14.67   |
> >   | InternVL3_5-30B-A3B |   33.33   |   13.33   |   40.00   |   26.67   |   16.67   |   10.00   |   16.67   |   22.38   |
> >   | InternVL3_5-8B      |   30.00   |   6.67    |   20.00   |   16.67   |   20.00   |   13.33   |   10.00   |   16.67   |
> >
> >   The results are consistent with those in Table 3, further highlighting the gap between humans and MLLMs.

---

> > > ### Author Response · Authors · 2025-11-14
> > > **Responses to Reviewer YUHP (part 3)**
> > >
> > > **Weakness 4**
> > >
> > > > The paper claims the questions cover real-life scenes, but many are in fact pretty artificial or unlikely to come across in real-life.
> > >
> > > **Reply 4**
> > >
> > > Thank you for your suggestion. In fact, we discuss this issue in detail in **Appendix B.1**. As defined in lines **1000–1001**, the "real-life" images in our work refer to those that "**exist in real life or are realistically simulated to resemble real-world conditions**." While we have included images such as cartoons and web pages, which are not directly sourced from nature, we believe these categories also **represent important visual information sources relevant to real life**. Additionally, since we rely solely on visual signals as input, it is challenging to capture the complex multi-hop causal relationships and counterfactual causality in real-world human activities. To ensure the dataset covers a broad range of reasoning types and tasks, we made a slight trade-off by including some images that simulate real-life situations.
> > >
> > >
> > >
> > > **Question 1**
> > >
> > > > What is the percentage of machine- and human-generated problems? I see there is a task description section in the appendix, but I think it is also important to clearly outline sources of questions (i.e., human/LLM-generated) for each task.
> > >
> > > **Reply 5**
> > >
> > > Thank you for your suggestion, it has been very helpful in improving our paper. Here, we provide the synthesis/manual classification for each task, as shown in the table:
> > >
> > > | Task                              | Synthesis/Manual | Count |
> > > | --------------------------------- | ---------------- | ----- |
> > > | Human Activity Attribution        | Manual           | 57    |
> > > | Character Interaction Attribution | Manual           | 70    |
> > > | Multi-Hop Collision Attribution   | Manual           | 181   |
> > > | Animal Relation Inference         | Synthesis        | 200   |
> > > | Product Similarity Inference      | Synthesis        | 191   |
> > > | Artwork Style Inference           | Synthesis        | 187   |
> > > | Character Interaction Prediction  | Manual           | 70    |
> > > | Multi-Hop Collision Prediction    | Manual           | 76    |
> > > | Counterfactual Fluid Prediction   | Synthesis        | 117   |
> > > | Material Composition Deduction    | Manual           | 99    |
> > > | Card Winner Deduction             | Synthesis        | 94    |
> > > | Recipe Step Deduction             | Synthesis        | 90    |
> > > | Bird Migration Induction          | Manual           | 60    |
> > > | Plant Disease Induction           | Synthesis        | 186   |
> > > | Sport Feature Induction           | Synthesis        | 198   |
> > > | Relative Position Estimation      | Manual           | 97    |
> > > | Camera Rotation Estimation        | Manual           | 98    |
> > > | Navigation Route Planning         | Manual           | 60    |
> > > | Crowd Timeline Reconstruction     | Synthesis        | 168   |
> > > | Driving Sequence Prediction       | Synthesis        | 195   |
> > > | Human Activity Localization       | Synthesis        | 182   |
> > >
> > > Overall, the proportion of manually labeled questions is 868 / 2676 =  **32.43%**, and the proportion of synthetic questions is 1808 / 2676 = **67.57%**.
> > >
> > >
> > >
> > > We commit to adding all of the above experiments and providing more complete results in the subsequent version of the paper.

---

> > > > ### Author Response · Authors · 2025-11-27
> > > > **Official Comment by Authors**
> > > >
> > > > Dear Reviewer YUHP,
> > > >
> > > > We sincerely appreciate the time and effort you have dedicated to reviewing our work. In response to your valuable feedback, we have provided detailed clarifications to the questions raised and included a revised version of the paper. As we are nearing the end of the discussion period, we would love to hear your thoughts on our response, including whether it sufficiently addresses your concerns. If our revisions and discussions indicate potential for a score adjustment, we would be deeply grateful for your consideration. We remain committed to incorporating all your suggestions to further enhance the quality of our manuscript. We hope we have addressed all your concerns and look forward to your further comments and discussions.
> > > >
> > > > Best regards,
> > > >
> > > > Authors

---

### Official Review · Reviewer_Bdkt · 2025-11-01

**Soundness:** 3
**Presentation:** 3
**Contribution:** 2
**Rating:** 6
**Confidence:** 5

**Summary:**

The paper introduces MMR-Life, a large-scale benchmark designed to evaluate multi-image reasoning abilities of multimodal large language models (MLLMs) in real-world scenarios. It includes 2,676 multiple-choice questions derived from 19,367 real-life images, spanning seven reasoning types. Unlike previous benchmarks that rely on domain-specific knowledge or synthetic geometric objects, MMR-Life emphasizes integrating information across multiple real-world images. Experiments with a broad range of open and proprietary models reveal certain insights and gaps relative to human performance, underscoring the need for further progress in multi-image reasoning.

**Strengths:**

1. The paper tackles an important and timely problem - evaluating whether MLLMs can effectively handle multi-image, vision-based reasoning tasks, a topic of clear relevance to the ICLR community.
2. The MMR-Life benchmark covers a broad range of reasoning types across diverse real-world image contexts, enabling fine-grained analysis of model strengths and weaknesses and offering a well-structured evaluation framework.
3. The evaluation is thorough and well-rounded, including a diverse set of both open and proprietary models. Importantly, the authors also examine the impact of different training and inference methods, such as Chain-of-Thought, Self-Consistency, Best-of-N, and GRPO, adding an additional analytical dimension.
4. The paper provides insightful analyses of performance patterns across reasoning categories and includes an error taxonomy that helps characterize model limitations and failure modes.
5. The paper is clearly written, logically structured, and comprehensive in its literature review, making it accessible and informative to readers.

**Weaknesses:**

1. The benchmark largely repurposes existing datasets (Appendix C.1), which limits its novelty. The primary contribution lies in the reorganization and categorization of these datasets by reasoning type rather than in introducing new data or task formulations.
2. The evaluation omits several important baselines, including representative MLLMs (e.g., LLaVA, InstructBLIP) and non-MLLM approaches such as supervised CNNs or few-shot/meta-learning methods (Prototypical Networks, SNAIL, MetaBaseline).
3. The human performance evaluation (Section 3.1) appears shallow and exhibits high variance. Each of the 600 validation questions seems to have been solved by only one annotator, making it unclear how results would generalize across individuals. Moreover, the comparison to models could be more balanced if model performance were reported on the same subset of questions used for human evaluation.
4. The claims of a substantial human–model performance gap (e.g., Figure 1) seem overstated. As shown in Table 3, GPT-5 actually outperforms humans on three reasoning categories (Analogical, Deductive, Inductive), with humans leading in the remaining four.
5. The Spatial reasoning category, identified as the most challenging, has already been explored in prior work - particularly in the SPACE benchmark [Ramakrishnan, 2025]. This overlap somewhat limits the unique contribution and potential utility of MMR-Life for advancing spatial reasoning evaluation.

Ramakrishnan, Santhosh Kumar, et al. "Does spatial cognition emerge in frontier models?" International conference on learning representations. 2025.

**Questions:**

1. Table 2: Why is the source of MMR-Life marked solely as A (Annotated)? Given the benchmark heavily draws from existing datasets, shouldn’t it also be labeled as E (Existing datasets)?
2. Can the authors clarify why human performance falls below GPT-5 in the Analogical, Deductive, and Inductive categories? Does this reflect genuine model advantages, human weaknesses, or potential biases in the evaluation setup?
3. How can the insights derived from the benchmark inform future model architecture design or prompting strategies to enhance multi-image reasoning?
4. Were the authors able to verify whether the benchmark’s source datasets were not included in the training data of the evaluated open models?
5. As noted, Keye-VL-1.5-8B and InternVL3.5-8B perform worse than random guessing. Can the authors explain this behavior? Does it stem from output formatting issues, context-length limitations, or other factors?

---

> ### Author Response · Authors · 2025-11-18
> **Responses to Reviewer Bdkt (part 1)**
>
> Thanks for your careful and insightful reviews.
>
> **Weakness 1**
>
> > The benchmark largely repurposes existing datasets (Appendix C.1), which limits its novelty. The primary contribution lies in the reorganization and categorization of these datasets by reasoning type rather than in introducing new data or task formulations.
>
> **Reply 1**
>
> We apologize for the confusion caused by the insufficient explanation in this section. In fact, the data sources listed in Appendix C.1 are solely the **image sources** for our dataset (see **lines 137-138**).  All the **questions** in this work are not directly reused from existing studies, but are **newly annotated or generated** (as described in the **"Question-Answer Generation"** section of Section 2.2). We believe that the reuse of images in this work does not limit its novelty. For instance, images in well-known works such as MMMU [1] are also collected from existing resources.
>
>
>
> **Weakness 2**
>
> > The evaluation omits several important baselines.
>
> **Reply 2**
>
> Thank you for your valuable suggestions, which have greatly contributed to improving our paper. We did not choose these baselines for our experiments due to the following considerations:
>
> + For traditional MLLMs like Llava, they support relatively short context lengths and cannot handle a large number of input images (e.g., >10) without **exceeding the maximum context length**, which leads to the inability to answer.
> + For training-based methods, such as supervised CNNs and meta-learning, a comparison is not feasible because **our benchmark does not have a training set**.
>
>
>
> **Weakness 3**
>
> > The human performance evaluation (Section 3.1) appears shallow and exhibits high variance. Moreover, the comparison to models could be more balanced if model performance were reported on the same subset of questions used for human evaluation.
>
> **Reply 3**
>
> + We apologize for the confusion caused by the lack of this detail. In fact, the 600 questions in this part are not all distinct.  We extracted 10 questions from each task to form a mini test set of 210 unique questions. From this pool, we repeatedly sampled 50 questions at a time and assigned them to one of 12 annotators, yielding a total of 600 valid annotations. Therefore, in practice, each individual question is annotated by multiple annotators to reduce randomness in the results.

---

> > ### Author Response · Authors · 2025-11-18
> > **Responses to Reviewer Bdkt (part 2)**
> >
> > + For the model performance on the subset, we have included additional experiments, and the results are shown in the table below:
> >
> >   | Model               |    Abd    |    Ana    |    Cau    |    Ded    |    Ind    |    Spa    |    Tem    |    Avg    |
> >   | :------------------ | :-------: | :-------: | :-------: | :-------: | :-------: | :-------: | :-------: | :-------: |
> >   | Human               | **79.76** |   57.65   | **75.00** |   70.59   |   63.41   | **79.76** | **79.76** | **72.28** |
> >   | gpt-5               |   63.33   |   66.67   |   53.33   |   73.33   | **66.67** |   30.00   |   50.00   |   57.62   |
> >   | gemini-2.5-pro      |   63.33   | **80.00** |   60.00   |   76.67   |   50.00   |   30.00   |   36.67   |   56.67   |
> >   | gemini-2.5-flash    |   53.33   |   63.33   |   50.00   |   53.33   |   50.00   |   50.00   |   53.33   |   53.33   |
> >   | o4-mini             |   46.67   |   73.33   |   43.33   |   73.33   |   50.00   |   23.33   |   43.33   |   50.48   |
> >   | gpt-5-mini          |   50.00   |   56.67   |   40.00   | **83.33** |   50.00   |   6.67    |   33.33   |   45.71   |
> >   | claude-sonnet-4     |   40.00   |   66.67   |   60.00   |   66.67   |   30.00   |   16.67   |   56.67   |   48.10   |
> >   | gpt-4.1             |   50.00   |   56.67   |   36.67   |   16.67   |   63.33   |   6.67    |   26.67   |   36.67   |
> >   | claude-3.7-sonnet   |   30.00   |   73.33   |   46.67   |   70.00   |   40.00   |   13.33   |   43.33   |   45.24   |
> >   | gpt-4o              |   43.33   |   50.00   |   40.00   |   16.67   |   46.67   |   20.00   |   26.67   |   34.76   |
> >   | gpt-4.1-mini        |   33.33   |   63.33   |   36.67   |   53.33   |   50.00   |   10.00   |   26.67   |   39.05   |
> >   | doubao-1.5-vision   |   50.00   |   50.00   |   36.67   |   63.33   |   40.00   |   13.33   |   23.33   |   39.52   |
> >   | VL-Rethinker-72B    |   26.67   |   50.00   |   40.00   |   63.33   |   33.33   |   13.33   |   23.33   |   35.71   |
> >   | QVQ-72B-Preview     |   26.67   |   53.33   |   50.0    |   36.67   |   26.67   |   3.33    |   16.67   |   30.48   |
> >   | MM-Eureka-Qwen-32B  |   20.00   |   46.67   |   43.33   |   50.00   |   33.33   |   16.67   |   36.67   |   35.24   |
> >   | MiMo-VL-7B-RL       |   36.67   |   16.67   |   40.00   |   70.00   |   16.67   |   13.33   |   26.67   |   31.43   |
> >   | VL-Rethinker-7B     |   23.33   |   36.67   |   13.33   |   13.33   |   33.33   |   0.00    |   16.67   |   22.78   |
> >   | Keye-VL-1.5-8B      |   13.33   |   23.33   |   33.33   |   20.00   |   26.67   |   13.33   |   26.67   |   22.38   |
> >   | Skywork-R1V-38B     |   30.00   |   6.67    |   33.33   |   16.67   |   30.00   |   16.67   |   13.33   |   20.95   |
> >   | Qwen2.5-VL-72B      |   36.67   |   40.00   |   40.00   |   20.00   |   36.67   |   10.00   |   30.00   |   30.48   |
> >   | Gemma3-27B          |   20.00   |   36.67   |   33.33   |   30.00   |   40.00   |   6.67    |   30.00   |   28.10   |
> >   | Gemma3-12B          |   30.00   |   50.00   |   16.67   |   33.33   |   20.00   |   6.67    |   23.33   |   25.71   |
> >   | Qwen2.5-VL-32B      |   23.33   |   30.00   |   30.00   |   20.00   |   20.00   |   6.67    |   23.33   |   21.90   |
> >   | Qwen2.5-VL-7B       |   13.33   |   0.00    |   23.33   |   20.00   |   0.00    |   6.67    |   10.00   |   14.67   |
> >   | InternVL3_5-30B-A3B |   33.33   |   13.33   |   40.00   |   26.67   |   16.67   |   10.00   |   16.67   |   22.38   |
> >   | InternVL3_5-8B      |   30.00   |   6.67    |   20.00   |   16.67   |   20.00   |   13.33   |   10.00   |   16.67   |
> >
> >   The results are consistent with those in Table 3, further highlighting the gap between humans and MLLMs.
> >
> >
> >
> > **Weakness 4**
> >
> > > The claims of a substantial human–model performance gap (e.g., Figure 1) seem overstated. As shown in Table 3, GPT-5 actually outperforms humans on three reasoning categories (Analogical, Deductive, Inductive), with humans leading in the remaining four.
> >
> > **Reply 4**
> >
> > Thank you for the suggestion. In the revised version of the paper, we will change the relevant claim to: “*There remains a gap between humans and SOTA MLLMs on some real-life reasoning tasks*.”

---

> > > ### Author Response · Authors · 2025-11-18
> > > **Responses to Reviewer Bdkt (part 3)**
> > >
> > > **Weakness 5**
> > >
> > > > The Spatial reasoning category, identified as the most challenging, has already been explored in prior work - particularly in the SPACE benchmark [Ramakrishnan, 2025]. This overlap somewhat limits the unique contribution and potential utility of MMR-Life for advancing spatial reasoning evaluation.
> > >
> > > **Reply 5**
> > >
> > > In fact, in the **Related Work** section, **lines 473 to 476**, we mention existing spatial reasoning benchmark studies. We acknowledge that the spatial reasoning component of our benchmark substantially overlaps with prior work. However, the primary contribution of our study is **not to design a stronger spatial reasoning benchmark**, but to provide a **general evaluation** of multi-image reasoning ability for real-world scenarios and to systematically assess the **different types of reasoning capabilities** of current MLLMs. From this perspective, overlap in certain subtasks does not diminish the overall contribution of our work.
> > >
> > >
> > >
> > >
> > >
> > > **Question 1**
> > >
> > > > Table 2: Why is the source of MMR-Life marked solely as A (Annotated)? Given the benchmark heavily draws from existing datasets, shouldn’t it also be labeled as E (Existing datasets)?
> > >
> > > **Reply 6**
> > >
> > > Here, A and E in Table 2 denote the source of the dataset questions. As stated in Reply 1, although the images in our dataset are largely collected from existing resources, **the questions in MMR-Life are newly annotated or generated**. Therefore, we label this as A rather than E.
> > >
> > >
> > >
> > > **Question 2**
> > >
> > > > Can the authors clarify why human performance falls below GPT-5 in the Analogical, Deductive, and Inductive categories? Does this reflect genuine model advantages, human weaknesses, or potential biases in the evaluation setup?
> > >
> > > **Reply 7**
> > >
> > > We believe the reasons for this gap are as follows:
> > >
> > > + **In these three task categories, the model can more readily obtain the key information from the images.** Compared with the other four categories, the images in these categories have **simpler information representations**. For example, in analogical tasks, the model only needs to recognize the basic style of items such as sneakers and paintings, and in deductive tasks, it only needs to identify information such as the number of objects. By contrast, achieving strong causal reasoning typically requires the model to capture differences in the behavior of the same entity before and after an event and to infer the underlying connections. Likewise, spatial tasks often require the model to reconstruct the relative positions of different objects across the entire scene. Humans are adept at extracting such complex information from multiple images.
> > > + **These three tasks rely more heavily on prior knowledge, and such knowledge is readily acquired during pretraining.** In these three task categories, **if the model has already acquired the relevant knowledge or rules, the reasoning becomes straightforward**. For example, in the Card Winner Deduction task, more capable models have likely encountered the rules of Texas Hold'em, and in the Artwork Style Inference task, they may already know the correspondence between artworks and their creators. In such cases, the model can complete the task through simple matching and recall. By contrast, the other tasks offer far less prior knowledge to leverage, **making it difficult for the model to find shortcuts**, which gives humans a greater advantage.
> > >
> > >
> > >
> > > **Question 3**
> > >
> > > > How can the insights derived from the benchmark inform future model architecture design or prompting strategies to enhance multi-image reasoning?
> > >
> > > **Reply 8**
> > >
> > > Based on our benchmark results, models perform worse when there are complex relations across images, such as temporal and spatial dependencies. Therefore, future model architectures or prompting strategies should **strengthen the ability to capture relationships among objects and events across multiple images**. For example, constructing scene graphs over the image sequence can improve an MLLM’s understanding of the sequence as a whole.
> > >
> > >
> > >
> > > **Question 4**
> > >
> > > > Were the authors able to verify whether the benchmark’s source datasets were not included in the training data of the evaluated open models?
> > >
> > > **Reply 9**
> > >
> > > Unfortunately, because the training datasets of most open source models are not publicly available, we cannot verify this potential issue. However, all questions and answers in our dataset are newly generated. Therefore, the likelihood of data leakage is very low.

---

> > > > ### Author Response · Authors · 2025-11-18
> > > > **Responses to Reviewer Bdkt (part 4)**
> > > >
> > > > **Question 5**
> > > >
> > > > > As noted, Keye-VL-1.5-8B and InternVL3.5-8B perform worse than random guessing. Can the authors explain this behavior? Does it stem from output formatting issues, context-length limitations, or other factors?
> > > >
> > > > **Reply 10**
> > > >
> > > > Based on our observations, the suboptimal performance of these two models is not due to limitations of the output format or the context length. We observed that when presented with the real-world scenario reasoning tasks in our benchmark, these two models exhibit markedly weak perception and reasoning performance. For example:
> > > >
> > > > + **Perception Error**
> > > >   + **Task:** Camera Rotation Estimation
> > > >   + **Model (InternVL3.5-8B):** *To determine the successive rotation angles of the camera, let's analyze the changes in perspective between the images:\n\n1. First to Second Image:\n  - The object appears to rotate slightly to the right.\n  - This suggests a clockwise rotation.\n\n2. Second to Third Image:\n  - The object appears to rotate further to the right...*
> > > >   + **Analysis:**  In this case, the model shows no ability to perceive the rotation angle, it can only perceive basic rotations directions.
> > > > + **Reasoning Error**
> > > >   + **Task:** Human Activity Attribution
> > > >   + **Model (Keye-VL-1.5-8B):** *<analysis>This question is straightforward and asks for a specific reason based on the images provided. Therefore, /no_think mode is more appropriate.</analysis># Answer: D*
> > > >   + **Analysis:**  For the Keye-VL-1.5-8B model, a substantial portion of its errors arises because it mistakenly switches to a fast thinking mode and produces a final answer without any intermediate CoT, a strategy that is often error-prone.
> > > >
> > > > Overall, the poor performance of these smaller models primarily stems from **insufficient perceptual or reasoning capabilities**, which is consistent with our analysis in Section 5. We will add further error analyses in the revised version of the paper.
> > > >
> > > >
> > > >
> > > > [1] Yue X, Ni Y, Zhang K, et al. Mmmu: A massive multi-discipline multimodal understanding and reasoning benchmark for expert agi[C]//Proceedings of the IEEE/CVF Conference on Computer Vision and Pattern Recognition. 2024: 9556-9567.

---

> > > > > ### Comment · Reviewer_Bdkt · 2025-11-25
> > > > > **Follow-up questions**
> > > > >
> > > > > Thank you for addressing my earlier questions and providing detailed responses.
> > > > >
> > > > > > Reply 2
> > > > >
> > > > > Thank you for the clarification. You mentioned that training-based baselines are infeasible due to the absence of a training set. However, since the task is essentially selecting the correct answer from several textual options based on visual information, would it be possible to construct a CLIP-style zero-shot baseline? Specifically, one could compute image embeddings (averaged or max-pooled across the multiple images) and compare them via cosine similarity against the text embeddings of each candidate answer. Such a baseline might help detect dataset-level biases or shortcut solutions.
> > > > >
> > > > > > Reply 3
> > > > >
> > > > > Thank you for elaborating, though the explanation still feels somewhat unclear. In the paper, “annotator” usually refers to dataset annotators who produce candidate answers, whereas here it seems to refer to the 12 student participants mentioned in Section 3.1 who took the test. Could you clarify the terminology to avoid confusion? Second, your explanation differs from what the paper initially implied – my understanding was that 600 unique questions from the dataset were solved by humans, not 210. I strongly suggest revising paragraph “Human Level Performance” in Section 3.1 by including these details, because they are important for interpreting human performance. In addition, it would be valuable to report standard deviation, minimum, and maximum scores across participants.
> > > > >
> > > > > > Reply 6
> > > > >
> > > > > Thank you for the clarification. Could you further explain what “generated” specifically means in this context? The paper provides an example for the temporal reasoning task, but it is not entirely clear how question generation works for the other reasoning categories. For instance, were some of the questions or candidate answers produced by an LLM, or created entirely through a rule-based or template-based process?

---

> > > > > > ### Author Response · Authors · 2025-11-26
> > > > > > **Responses to Reviewer Bdkt (part 1)**
> > > > > >
> > > > > > Thank you for your prompt reply and valuable suggestions.
> > > > > >
> > > > > > **Question 1**
> > > > > >
> > > > > > > Thank you for the clarification. You mentioned that training-based baselines are infeasible due to the absence of a training set. However, since the task is essentially selecting the correct answer from several textual options based on visual information, would it be possible to construct a CLIP-style zero-shot baseline? Specifically, one could compute image embeddings (averaged or max-pooled across the multiple images) and compare them via cosine similarity against the text embeddings of each candidate answer. Such a baseline might help detect dataset-level biases or shortcut solutions.
> > > > > >
> > > > > > **Reply 1**
> > > > > >
> > > > > > Here, we conduct experiments on MMR-Life using two CLIP models in an image-text matching paradigm. Specifically, we use the average of the input image embeddings as the query. For text options, we concatenate the question with each option before computing their embeddings. For image options, we directly embed each image. The option with the highest similarity is ultimately selected as the predicted answer. The results are shown in the table:
> > > > > >
> > > > > > |               | Abd   | Ana   | Cau   | Ded   | Ind   | Spa   | Tem   | Avg   |
> > > > > > | ------------- | ----- | ----- | ----- | ----- | ----- | ----- | ----- | ----- |
> > > > > > | clip-ViT-B-32 | 24.03 | 41.52 | 28.14 | 19.79 | 57.66 | 20.39 | 21.47 | 32.47 |
> > > > > > | clip-ViT-L-14 | 25.32 | 44.46 | 28.14 | 16.96 | 62.39 | 24.31 | 15.23 | 32.85 |
> > > > > >
> > > > > > The results show that, for reasoning types that rely on pattern matching between images (such as Ana and Ind), the CLIP models achieve satisfactory performance. However, for other inference types, the CLIP models only reach chance-level performance. This suggests that **a single CLIP model can only handle a very limited range of reasoning tasks**.
> > > > > >
> > > > > > **Question 2**
> > > > > >
> > > > > > > Thank you for elaborating, though the explanation still feels somewhat unclear. In the paper, “annotator” usually refers to dataset annotators who produce candidate answers, whereas here it seems to refer to the 12 student participants mentioned in Section 3.1 who took the test. Could you clarify the terminology to avoid confusion? Second, your explanation differs from what the paper initially implied – my understanding was that 600 unique questions from the dataset were solved by humans, not 210. I strongly suggest revising paragraph “Human Level Performance” in Section 3.1 by including these details, because they are important for interpreting human performance. In addition, it would be valuable to report standard deviation, minimum, and maximum scores across participants.
> > > > > >
> > > > > > **Reply 2**
> > > > > >
> > > > > > Thank you for your suggestion. We apologize for the misstatement regarding the term 'annotator' in our previous response. In our paper, we only used the term 'annotator' **when discussing data annotation (Appendix C)**, whereas we solely used 'students' to refer to the participants in the human performance section. We have supplemented the relevant details in the latest version of the paper (**marked in blue**). Besides, we report the standard deviation, minimum, and maximum scores as follows:
> > > > > >
> > > > > > |               | Abd    | Ana   | Cau    | Ded   | Ind   | Spa   | Tem   | Avg   |
> > > > > > | :------------ | :----- | :---- | :----- | :---- | :---- | :---- | :---- | :---- |
> > > > > > | $\sigma$  | 14.99  | 9.22  | 23.57  | 13.41 | 11.26 | 15.89 | 9.32  | 10.73 |
> > > > > > | min       | 66.67  | 48.00 | 50.00  | 56.00 | 50.00 | 58.33 | 70.00 | 60.00 |
> > > > > > | max       | 100.00 | 66.67 | 100.00 | 83.33 | 73.33 | 90.00 | 86.67 | 80.00 |

---

> > > > > > > ### Author Response · Authors · 2025-11-26
> > > > > > > **Responses to Reviewer Bdkt (part 2)**
> > > > > > >
> > > > > > > **Question 3**
> > > > > > >
> > > > > > > > Thank you for the clarification. Could you further explain what “generated” specifically means in this context? The paper provides an example for the temporal reasoning task, but it is not entirely clear how question generation works for the other reasoning categories. For instance, were some of the questions or candidate answers produced by an LLM, or created entirely through a rule-based or template-based process?
> > > > > > >
> > > > > > > **Reply 3**
> > > > > > >
> > > > > > > We apologize for not explaining this in detail in our previous response. As emphasized in **Lines 181–183** of the manuscript, 'generate' refers to constructing the questions and candidate options **using a set of heuristic rules**, rather than employing an LLM for generation. We present additional examples below:
> > > > > > >
> > > > > > > + **Artwork Style Inference (Appendix E.2.3)**
> > > > > > >
> > > > > > >   This task provides several input artworks and requires the model to select the most similar artwork from the candidate options. For data generation, we **randomly select works from the same artist** to serve as the **inputs** and the **correct option**. Subsequently, we **randomly sample works from different artists** to serve as the **incorrect options**.
> > > > > > >
> > > > > > > + **Card Winner Deduction (Appendix E.4.2)**
> > > > > > >
> > > > > > >   This task requires the model to select the final winner from the candidate options based on the rules of Texas Hold'em. When generating the problems, we perform **sampling without replacement** from a standard deck of cards to create the input images and sample the options based on a specific set of rules. Finally, the correct answer is determined by **rules implemented in Python code**.
> > > > > > >
> > > > > > > + **Plant Disease Induction (Appendix E.5.2)**
> > > > > > >
> > > > > > >   Similar to the ASI task, we randomly sample the input images and positive examples from images **corresponding to the same disease**. We then sample negative examples from **images corresponding to other diseases**. The disease categorization used here is inherently provided by the original dataset.
> > > > > > >
> > > > > > > In fact, the generation code for each synthesis task is included in our **uploaded supplementary material**. We will supplement the corresponding generation details in the subsequent version of the paper.

---

### Official Review · Reviewer_5uk2 · 2025-11-01

**Soundness:** 3
**Presentation:** 3
**Contribution:** 3
**Rating:** 6
**Confidence:** 5

**Summary:**

The paper introduces MMR-Life, a benchmark for evaluating multi-image and multimodal reasoning in realistic scenarios. It builds 2,676 multiple-choice questions from 19,367 real-world images, covering seven reasoning types and 21 tasks. The authors construct the dataset through automatic question generation plus human filtering, then evaluate 37 vision–language models to compare with human performance. Results show a clear gap between models and humans, especially on spatial and temporal reasoning, so the benchmark is intended to serve as a diagnostic testbed for future multimodal reasoning model

**Strengths:**

1. The benchmark uses 2,676 MCQ questions built from 19,367 real-world images, with many questions requiring information integration across several images, so models cannot rely on single-image shortcuts.
2. Reasoning tasks are organized into seven well-defined types (abductive, analogical, causal, deductive, inductive, spatial, temporal) and 21 tasks, providing a structured view of where current models fail.
3. The analysis across reasoning types shows differentiated difficulty profiles, which makes the benchmark useful as a diagnostic tool rather than only a leaderboard.

**Weaknesses:**

1. The paper analyzes the effect of longer thinking traces only at the overall level, lack discussion by break down reasoning type, it is unclear whether “longer thinking ≠ better” holds uniformly across tasks.
2. Human-level performance is based on a subset (validation questions), whereas model results are reported on the full benchmark, which makes the human–model comparison not fully aligned and may overstate the human gap.
3. Error analysis stays relatively shallow and mostly descriptive, could reveal deeper insights, guiding future model improvements.

### **Minor comments**
1. Figure 6 font size is a bit small
2. In Figure 48 the chosen option seems defensible from the question phrasing.
3. Table 2 lists MMMU as a comparison point, but MMMU is not a single-image benchmark.

**Questions:**

Please refer to weakness section.

---

> ### Author Response · Authors · 2025-11-14
> **Responses to Reviewer 5uk2 (part 1)**
>
> Thanks for your careful and insightful reviews.
>
> **Weakness 1**
>
> > The paper analyzes the effect of longer thinking traces only at the overall level, lack discussion by break down reasoning type, it is unclear whether “longer thinking ≠ better” holds uniformly across tasks.
>
> **Reply 1**
>
> We apologize for the confusion caused by the unclear presentation of this part of the experiment. In fact, we have already discussed this in detail in **lines 320–355 of Section 4.1** (see **Figure 4** and **Figure 5**). Our conclusion is that for reasoning types that require rapid thinking (such as inductive reasoning), longer thinking does not lead to better performance. However, for reasoning types that require step-by-step thinking (such as deductive reasoning), longer thinking does improve performance.
>
>
>
> **Weakness 2**
>
> > Human-level performance is based on a subset (validation questions), whereas model results are reported on the full benchmark, which makes the human–model comparison not fully aligned and may overstate the human gap.
>
> **Reply 2**
>
> For the model performance comparison on the subset, we have included additional experiments, and the results are shown in the table below:
>
> | Model               |    Abd    |    Ana    |    Cau    |    Ded    |    Ind    |    Spa    |    Tem    |    Avg    |
> | :------------------ | :-------: | :-------: | :-------: | :-------: | :-------: | :-------: | :-------: | :-------: |
> | Human               | **79.76** |   57.65   | **75.00** |   70.59   |   63.41   | **79.76** | **79.76** | **72.28** |
> | gpt-5               |   63.33   |   66.67   |   53.33   |   73.33   | **66.67** |   30.00   |   50.00   |   57.62   |
> | gemini-2.5-pro      |   63.33   | **80.00** |   60.00   |   76.67   |   50.00   |   30.00   |   36.67   |   56.67   |
> | gemini-2.5-flash    |   53.33   |   63.33   |   50.00   |   53.33   |   50.00   |   50.00   |   53.33   |   53.33   |
> | o4-mini             |   46.67   |   73.33   |   43.33   |   73.33   |   50.00   |   23.33   |   43.33   |   50.48   |
> | gpt-5-mini          |   50.00   |   56.67   |   40.00   | **83.33** |   50.00   |   6.67    |   33.33   |   45.71   |
> | claude-sonnet-4     |   40.00   |   66.67   |   60.00   |   66.67   |   30.00   |   16.67   |   56.67   |   48.10   |
> | gpt-4.1             |   50.00   |   56.67   |   36.67   |   16.67   |   63.33   |   6.67    |   26.67   |   36.67   |
> | claude-3.7-sonnet   |   30.00   |   73.33   |   46.67   |   70.00   |   40.00   |   13.33   |   43.33   |   45.24   |
> | gpt-4o              |   43.33   |   50.00   |   40.00   |   16.67   |   46.67   |   20.00   |   26.67   |   34.76   |
> | gpt-4.1-mini        |   33.33   |   63.33   |   36.67   |   53.33   |   50.00   |   10.00   |   26.67   |   39.05   |
> | doubao-1.5-vision   |   50.00   |   50.00   |   36.67   |   63.33   |   40.00   |   13.33   |   23.33   |   39.52   |
> | VL-Rethinker-72B    |   26.67   |   50.00   |   40.00   |   63.33   |   33.33   |   13.33   |   23.33   |   35.71   |
> | QVQ-72B-Preview     |   26.67   |   53.33   |   50.0    |   36.67   |   26.67   |   3.33    |   16.67   |   30.48   |
> | MM-Eureka-Qwen-32B  |   20.00   |   46.67   |   43.33   |   50.00   |   33.33   |   16.67   |   36.67   |   35.24   |
> | MiMo-VL-7B-RL       |   36.67   |   16.67   |   40.00   |   70.00   |   16.67   |   13.33   |   26.67   |   31.43   |
> | VL-Rethinker-7B     |   23.33   |   36.67   |   13.33   |   13.33   |   33.33   |   0.00    |   16.67   |   22.78   |
> | Keye-VL-1.5-8B      |   13.33   |   23.33   |   33.33   |   20.00   |   26.67   |   13.33   |   26.67   |   22.38   |
> | Skywork-R1V-38B     |   30.00   |   6.67    |   33.33   |   16.67   |   30.00   |   16.67   |   13.33   |   20.95   |
> | Qwen2.5-VL-72B      |   36.67   |   40.00   |   40.00   |   20.00   |   36.67   |   10.00   |   30.00   |   30.48   |
> | Gemma3-27B          |   20.00   |   36.67   |   33.33   |   30.00   |   40.00   |   6.67    |   30.00   |   28.10   |
> | Gemma3-12B          |   30.00   |   50.00   |   16.67   |   33.33   |   20.00   |   6.67    |   23.33   |   25.71   |
> | Qwen2.5-VL-32B      |   23.33   |   30.00   |   30.00   |   20.00   |   20.00   |   6.67    |   23.33   |   21.90   |
> | Qwen2.5-VL-7B       |   13.33   |   0.00    |   23.33   |   20.00   |   0.00    |   6.67    |   10.00   |   14.67   |
> | InternVL3_5-30B-A3B |   33.33   |   13.33   |   40.00   |   26.67   |   16.67   |   10.00   |   16.67   |   22.38   |
> | InternVL3_5-8B      |   30.00   |   6.67    |   20.00   |   16.67   |   20.00   |   13.33   |   10.00   |   16.67   |
>
> The results are consistent with those in Table 3, further highlighting the gap between humans and MLLMs.

---

> > ### Author Response · Authors · 2025-11-14
> > **Responses to Reviewer 5uk2 (part 2)**
> >
> > **Weakness 3**
> >
> > > Error analysis stays relatively shallow and mostly descriptive, could reveal deeper insights, guiding future model improvements.
> >
> > **Reply 3**
> >
> > Thank you for your valuable suggestions. In fact, in addition to providing qualitative analyses of MLLM errors with concrete examples in Section 5, we also conducted an in depth quantitative analysis in Section 4 from three perspectives: thinking length, training methodology, and reasoning type, to investigate the factors contributing to the suboptimal performance of MLLMs. In future work, we will provide deeper mechanistic explanations of these errors by analyzing the training dataset, the internal information flow of the model, and other related factors.
> >
> >
> >
> > **Minor Comments**
> >
> > > Figure 6 font size is a bit small. In Figure 48 the chosen option seems defensible from the question phrasing. Table 2 lists MMMU as a comparison point, but MMMU is not a single-image benchmark.
> >
> > **Reply 4**
> >
> > + We apologize for the inconvenience caused by the small font size. We will enlarge it in the subsequent version of the paper.
> > + For this example,  although option E seems reasonable, most of the images in the input are related to water sports, so option B is a more reasonable answer. We apologize for the choice of explanation here, which was not optimal, and we will improve it in the subsequent version of the paper.
> > + We apologize for the ambiguity in our description that led to this misunderstanding. MMMU indeed contains a small number of multi-image input questions. The average number of images in Table 2 was computed using the total number of images divided by the total number of questions provided by the original paper [1].
> >
> >
> >
> > We will include the above experiments and related analyses in the revised version of the paper, and we will refine several insufficient descriptions accordingly.
> >
> >
> >
> > [1] Yue X, Ni Y, Zhang K, et al. Mmmu: A massive multi-discipline multimodal understanding and reasoning benchmark for expert agi[C]//Proceedings of the IEEE/CVF Conference on Computer Vision and Pattern Recognition. 2024: 9556-9567.

---

> ### Comment · Reviewer_5uk2 · 2025-11-21
>
> Thank you for your detailed response. I think the paper is acceptable, most of my concerns have been addressed, and I will raise my score.

---

> > ### Author Response · Authors · 2025-11-23
> > **Response to Reviewer 5uk2**
> >
> > Thank you for your response and for raising the score. If you have any further questions or concerns, please feel free to contact us at any time.

---

### Author Response · Authors · 2025-12-02
**Author Final Remarks (part 1)**

Dear Chairs,

We deeply regret the premature termination of the discussion period and sincerely appreciate your dedication throughout the review phase. To assist you in better evaluating our work, we hereby provide a summary of the entire discussion process.

**1. The Main Weaknesses and Corresponding Responses**

We summarize the main weaknesses of our work as identified by the reviewer and provide corresponding responses as follows:

+ **W1: The details of some experiments are not clear**

  + **Description of the Weakness**

    Some reviewers have raised questions regarding the specific experimental settings of our benchmark (**Reviewer Bdkt's W3, Reviewer YUHP's W1, W3, and Q1**). For instance, queries include how the human evaluation experiment was specifically conducted, the specific generation process for each task, and other related details.

  + **Our Responses**

    We have provided detailed supplements for all the above experimental details and made all benchmark data available via an anonymous link. Please refer to the corresponding replies for specifics.

+ **W2: The performance of the model on the human evaluation subset is missing**

  + **Description of the Weakness**

    All reviewers pointed out the need to provide the model's performance on the human evaluation subset to enable a fairer comparison (**Reviewer 5uk2's W2, Reviewer Bdkt's W3, Reviewer YUHP's W3**).

  + **Our Responses**

    We have supplemented the results for all models in the main experiment on the subset (see specific replies), which further demonstrates the gap between human performance and SOTA MLLMs.

+ **W3: Some descriptions in the paper caused confusion for the reviewers**

  + **Description of the Weakness**

    Some reviewers believe that certain descriptions in the paper are inappropriate, leading to confusion (**Reviewer 5uk2's Minor Comments, Reviewer Bdkt's W1, W4, Q1, Reviewer YUHP's W2, W4**). For example, issues include the labeling of benchmark data sources, the description of examples in the appendix, and the summary of related work.

  + **Our Responses**

    Regarding some misunderstandings on the part of the reviewers (e.g., incorrectly interpreting our data source), we provided detailed clarification in our subsequent replies. For some objectively existing statement issues (e.g., conclusions being stated too absolutely), we revised the subsequent version of the paper according to the reviewers' suggestions.

We believe that these responses **effectively resolved the reviewers' main concerns**, and **Reviewers 5uk2 and Bdkt confirmed this point** in their subsequent replies.



**2. Main Strengths of Our Work**

All reviewers gave us a **positive rating** and highly affirmed the following strengths of the work:

+ **Diagnostic Power and Structured Evaluation Framework**

  Our work's primary strength is the creation of a highly structured and fine-grained evaluation framework that serves as a diagnostic tool rather than just a leaderboard, clearly outlining model deficiencies (**Reviewers 5uk2 and Bdkt**).

+ **Thoroughness and Breadth of Evaluation**

  The paper excels in its comprehensive evaluation, covering various state-of-the-art models and specific reasoning enhancement strategies (**Reviewer Bdkt and YUHP**).

+ **High Quality and Clarity of Writing**

  The paper is commended for being well-written, making the complex topic and findings accessible to the reader (**Reviewer Bdkt and YUHP**).

These strengths indicate that the reviewers are uniformly affirmative regarding the main contributions of the work. In contrast, we believe that the aforementioned weaknesses do not affect these contributions.

---

> ### Author Response · Authors · 2025-12-02
> **Author Final Remarks (part 2)**
>
> **3. Discussion of Potential Changes in Reviewer Impressions**
>
> Unfortunately, despite sending reminders to the reviewers, kindly asking them to respond to our rebuttal, we were still unable to engage in a complete discussion with all reviewers by the time of the leak incident. However, we believe **the reviewers’ evaluation of our paper would have seen a positive increase** had the discussion proceeded normally, for the following reasons:
>
> + **For Reviewer 5uk2**
>
>   Before the leak incident occurred, Reviewer 5uk2 had explicitly stated, ***"I think the paper is acceptable, most of my concerns have been addressed, and I will raise my score"*** and subsequently raised the rating of the paper (**6 -> 8**).
>
> + **For Reviewer Bdkt**
>
>   First, the reviewer explicitly stated, ***"Thank you for addressing my earlier questions,"*** indicating that **most of his/her initial concerns had been resolved**. Second, regarding the follow-up questions, we promptly supplemented our response with relevant experimental results and statements. Although the reviewer was unable to reply to these answers in a timely manner, given that these questions **were not based on subjective disagreements**, we have reason to believe that **these objective experimental results will convince the reviewer**.
>
> + **For Reviewer YUHP**
>
>   We regret that the reviewer was unable to provide a timely response to our initial rebuttal, but we observed that some of the weaknesses raised are **highly overlapping with those of Reviewer 5uk2** (W2 part 1 and W3). Since Reviewer 5uk2 affirmed the corresponding replies, we have reason to believe that **these replies would also resolve Reviewer YUHP's concerns**. Furthermore, some of the reviewer's questions required us to provide more data (W1 and Q1), and our supplementation of this data should **objectively resolve these issues directly**.
>
> **In summary**, we comprehensively addressed the weaknesses raised by the reviewers and updated the relevant content in the manuscript (**marked in blue**). Although we were unable to receive further responses from most reviewers due to the leak incident, we have provided evidence to demonstrate the potential for a positive increase in their evaluation.
>
> Once again, we wish to express our profound appreciation for your invaluable role and the considerable time devoted to this process. We further extend our sincere thanks to the reviewers for their meticulous feedback and valuable suggestions.
>
> Best regards,
>
> Authors

---

### Meta-Review · Area_Chair_gzMZ · 2026-01-07

**Summary:**

Strengths:

- Broad set of models and systems evaluated (Bdkt, YUHP)
- Clearly written paper (Bdkt, YUHP)

Weaknesses

- Shallow error analysis (5uk2)
- "Only" repurposes existing datasets (Bdkt)
- Low quality cherry picked examples (YUHP)

**Reviewer Concerns:**

Overall, I think the paper is strong, with no outstanding concerns.

Both the error analysis and the cherry-picked examples seem to have been resolved during the rebuttal.

**Reviewer Scores:**

I agree with the authors and I think all reviewers would have raised their scores.

---

### Decision · Program_Chairs · 2026-01-26

Accept (Poster)